# Inter-bacterial mutualism promoted by public goods in a system characterized by deterministic temperature variation

Yuxiang Zhao[1], Zishu Liu [1], Baofeng Zhang[2], Jingjie Cai[1], Xiangwu Yao[1], Meng Zhang[1], Ye Deng [3,4] & Baolan Hu [1,5,6] ✉

Mutualism is commonly observed in nature but not often reported for bacterial communities. Although abiotic stress is thought to promote microbial mutualism, there is a paucity of research in this area. Here, we monitor microbial communities in a quasi-natural composting system, where temperature variation (20 °C–70 °C) is the main abiotic stress. Genomic analyses and culturing experiments provide evidence that temperature selects for slow-growing and stress-tolerant strains (i.e., *Thermobifida fusca* and *Saccharomonospora viridis*), and mutualistic interactions emerge between them and the remaining strains through the sharing of cobalamin. Comparison of 3000 bacterial pairings reveals that mutualism is common (~39.1%) and competition is rare (~13.9%) in pairs involving *T. fusca* and *S. viridis*. Overall, our work provides insights into how high temperature can favour mutualism and reduce competition at both the community and species levels.

Typically, bacteria interact with other organisms to form a complex network through the exchange of information, energy, and materials[1]. Bacterial interactions are the primary driver for the compositional and functional stability of communities[2]. Bacterial interactions can be classified according to the different combinations of neutral, win, and loss outcomes for the two interacting partners[3].

Mutualism is a win-win situation and is represented by cross-feeding, a well-established and common type of mutualistic interaction[4]. However, microbes have evolved numerous methods to harm other strains, including competition for nutrients and altering the local environment. The gold standard (comparison of monoculture and co-culture) has always shown a dominance of competition[5]. In particular, experiments on culturable bacteria isolated from tree holes[6] and gut[7] have confirmed that positive interactions were found only rarely in bacterial pairings and were even nonexistent for some environmental systems[8]. Kehe et al. reported a counterexample that positive interactions (primarily as parasitisms) are commonly

observed between cultivable strains isolated from soil[9]. However, mutualism is still rare as it would change to commensalism or parasitism depending on the environmental conditions.

However, some controversy has emerged. Piccardi et al. reported that mutualism was the dominant relationship among 4 species cultured in metal working fluids[10], implying that abiotic stress might favor positive interactions[11]. On paper, recent investigations have provided clues to this phenomenon, including studies on different altitudes[12], water supply gradients[13], nutrients[14], and temperatures[15]. However, it is hard to repeatedly observe how bacterial interactions respond to changing abiotic stress in that it involves a continuous dynamic process that encompasses changes in the environment, microbial communities, and interactions within the same habitat. To address the challenges of repeatability and changing stress, artificial communities have thus been studied. These communities feature high tractability, but they have low generality and are thus poorly reflect the natural behavior of bacteria[16]. Moreover, artificial abiotic stress might lead to

[1]College of Environmental and Resource Sciences, Zhejiang University, Hangzhou, China. [2]Hangzhou Ecological and Environmental Monitoring Center, Hangzhou, China. [3]CAS Key Laboratory for Environmental Biotechnology, Research Center for Eco-Environmental Sciences, Chinese Academy of Sciences, Beijing, China. [4]College of Resources and Environment, University of Chinese Academy of Sciences, Beijing, China. [5]Zhejiang Province Key Laboratory for Water Pollution Control and Environmental Safety, Hangzhou, China. [6]Key Laboratory of Environment Remediation and Ecological Health, Ministry of Education, College of Environmental Resource Sciences, Zhejiang University, Hangzhou, China. ✉e-mail: blhu@zju.edu.cn

the collapse of a community due to rapid or inappropriate changes. Thus, it is essential to explore how microbial interactions change in a quasi-natural habitat under abiotic stress.

In this study, we explored a quasi-natural habitat (composting) which literally satisfied the conditions required to tackle the above-mentioned problems. Composting is an efficient treatment for solid waste, and a great deal of attention has been paid to its efficiency and quality[17]. With its unique ecological characteristics, composting could help us to solve the problems related to microbial interactions and abiotic stress. Specifically, (i) repeatability and continuation: composting undergoes a heating-cooling process. The whole bioprocess can occur within 30 days, and the dynamics and environmental changes of similar microbial communities can be obtained with similar initial substrates[18]; (ii) High temperature stress: the biodegradation of organic matter is accompanied by the variation in temperature, which could rise from 20 °C to 70 °C and then fall back to 20 °C[19]. The bacterial community can acclimatize to high temperature without collapsing, because the temperature changes are caused by their own metabolism; (iii) A semi-sealed environment: once the initial material is composted, the composting process will be completed spontaneously without any intervention. Although ventilation is the only possible way to introduce external microbes, it has little impact[20]. Given these conditions, we considered composting as an ideal biosystem to demonstrate how bacterial interactions change when subjected to dynamic temperature stress.

Herein, we continuously monitored how microbial communities respond to abiotic stress with multiple metagenomic approaches and analyses. Moreover, we isolated and co-cultured 44 strains to confirm the genomic findings of communities and offer a new perspective at the species level. In particular, we aimed to answer two questions: (i) How did bacterial interactions change in a biosystem characterized by deterministic temperature variation? (ii) Are there any species that can regulate inter-bacterial mutualism? Overall, our results indicated that (i) high temperature stress could enhance mutualism and (ii) slow-growing, but stress-tolerant, species cooperate with other bacteria by leaking metabolites.

## Results
### Temperature effects on mutualism
Full-scale (about 8.0 tons of food waste) composting was repeated 10 times with similar initial material (Fig. 1). As predicted, the temperature rose from 18.9 °C to 68.8 °C and remained above 50 °C for 19 days, on average (Fig. 1a). To reveal changes of potential bacterial interactions under these temperature variations, a total of 60 bacterial communities (including 6 sampled points for each pile) were determined with high-throughput 16 S rRNA gene sequencing (V4 region) (Note 1 in SI). Meanwhile, we calculated the positive and negative cohesion to characterize the potential bacterial interactions, which is the sum of the significant positive (negative) correlations between taxa weighted by the relative abundance of taxa in each sample (Fig. 1b and Fig. S5 in SI)[21]. Given its importance for both community assembly (calculated by iCAMP analysis) and the functional potential (abundance of 52 functional genes quantified by quantitative PCR (qPCR)), the focus was on positive cohesion (i.e. potential interactions) rather than negative cohesion (Note 2 in SI and Supplementary Data 1). As with temperature, positive cohesion also peaked on day 5 (0.28), which was 1.1–1.6 times higher than other periods (Fig. 1b). A significant correlation could be observed between positive cohesion and temperature ($r = 0.71$, $p < 0.001$) (Fig. 1c). Meanwhile, various statistical methods also indicated that temperature was the most decisive factor in determining positive cohesion (Note 3 in SI). Thus, high temperature was the main factor enhancing the potential mutualism.

Four kinds of ASVs were defined, including overall abundant ASVs (average relative abundance > 0.1%)[22], ubiquitous ASVs (detected in over 80% of samples)[23], frequently abundant ASVs (the relative abundance being top 80% in >50% of samples)[24], and PosCoh related ASVs (ASVs with significant impact on the changes of positive cohesion) (Supplementary Data 2). In detail, the impact was assessed by the percentage of increase of mean square error (increase of MSE), and significance was assessed by the significance of each ASV on positive cohesion (performed through a random forest model[25]) (Fig. S18 in SI). Only 0.0002% (3 ASVs) fulfilled all four criteria, and these accounted for 12.3% of the total sequences (Fig. S19-S21 and Text S4.1 in SI). These three ASVs belonged to the Actinobacteria, with two ASVs (ASV2 and ASV4) classified as *Thermobifida fusca* and one ASV (ASV1) classified as *Saccharomonospora viridis*. To distinguish them from other Actinobacterial ASVs, we named these 3 ASVs as ACT ASVs. Based on their within-module connectivity (*Zi*) and among-module connectivity (*Pi*) in the molecular ecological network (Fig. 1e and Supplementary Data 3), all ACT ASVs were also identified as keystone nodes (Fig. S22 and Text S4.2 in SI). The sub-network between ACT ASVs and neighboring ASVs (those directly connected to the ACT ASVs) indicated that 23 neighboring ASVs (71.9%), were positively correlated with the ACT ASVs (Fig. 1f and Supplementary Data 4). More than 68.8% of them were affiliated with Firmicutes (Fig. 1f and Fig. S23 in SI), which is the dominant phylum in composting systems[18].

The relative abundance of all ACT ASVs peaked at day 5 and then decreased as temperature decreased. Moreover, the significant positive correlations between temperature and the relative abundance of the ACT ASVs (Fig. S21 in SI) also confirmed that high temperature favored these species. To further determine the ability of species to survive under high temperature (fast-growing copiotrophs or slow-growing but efficient oligotrophs), maximum growth rate was assessed[26]. Considering that the ribosomal RNA operon (rrn) copy number is a phylogenetically conserved trait and a proxy for maximum growth rate[27], we matched Ribosomal RNA Operon Copy Number Database (rrndb) to infer maximum growth rates of neighboring ASVs and ACT ASVs (Text S5 in SI). The significantly higher copy number of rrn in neighbouring ASVs (6.4) (Fig. S24 in SI), and the high explanation of changes in functional potential (55.0%) (Fig. S25 in SI) emphasized that the neighboring ASVs were fast-growing bacteria, and were highly associated with composting function. In contrast, the lower rrn copy number in ACT ASVs indicated their slow-growing characteristics. The significant positive correlations between ACT ASVs' abundance (with lower rrn copy number) and temperature (Fig. S21 in SI) also implied that high temperature might favor slow-growing bacterial species. To confirm this, we calculated the mean copy number (MCN), which is the abundance-weighted rrn copy number for each sample[26], to summarize how temperature affected the distribution of fast and slow growing organisms in the composting bacterial community. The minimum value of MCN was observed on day 5 (Fig. S24 in SI). Moreover, correlation analysis (Fig. S24c in SI) and multiple linear regression (MLR) analysis (Fig. S26 and Table S6 in SI) supported that temperature was the main factor regulating MCN (Text S5 in SI). Therefore, high temperature favors slower-growing bacterial species, and these species (i.e., ACT ASVs) might cooperate with other species that grow rapidly and affect composting functions. Finally, the global distribution of the ACT ASVs is supported by their widespread co-occurrence in both natural and engineered habitats, with a relative abundance of >1% (Fig. S27 in SI). Overall, ACT ASVs, affiliated with *Thermobifida fusca* and *Saccharomonospora viridis*, might promote functional potential via mutualism with Firmicutes.

### MAGs' composition and transcriptional activity
The 60 samples were combined according to the different composting phases (10 samples mixed into 1 depending on sampling time) to obtain 6 combined samples, for metagenomic and metatranscriptomic sequencing. After being co-assembled and binned, 467 metagenome assembled genomes (MAGs) were obtained. Following published

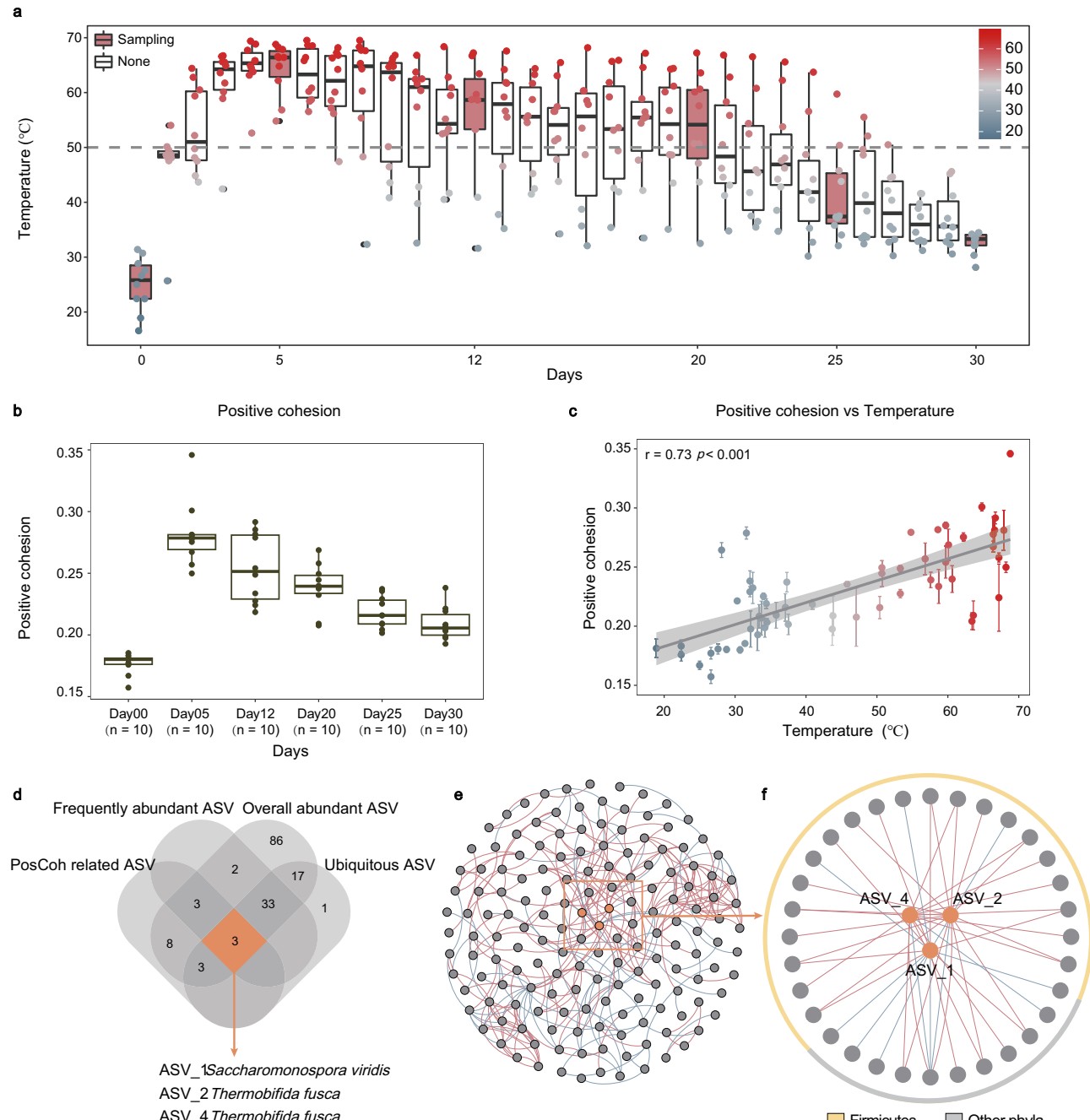

**Fig. 1 | Changes of temperature and positive cohesion. a** Temperature variations in the 10 composting piles. 50 °C is generally considered to be the dividing line for composting in terms of distinguishing the high temperature phase from other phases in this system. In the boxplots of panels, hinges indicate the 25th, 50th, and 75th percentiles, whiskers indicate 1.5 × interquartile ranges, and dots indicate values of individual samples. **b** Changes in the positive cohesion of different phases. In the boxplots of panels, hinges indicate the 25th, 50th, and 75th percentiles, whiskers indicate 1.5 × interquartile ranges, and dots indicate values of individual samples ($n = 10$ biologically independent samples for each day). **c** Relationship between positive cohesion and temperature. Data are presented as mean values +/- SEM. Pearson correlation coefficients is shown ($n = 60$ biologically independent samples). **d** Screening for the ACT ASVs (detail of the four criteria are described in Methods section). **e** Whole composting network constructed by the random matrix theory (RMT) model. The red line indicates a positive correlation and the blue lines a negative correlation. **f** The sub-network containing ACT ASVs and their neighbors. Different colors of the outer ring are used to indicate taxonomic affiliation.

standards[28], 159 MAGs were regarded as high-quality and these were selected for further study (Supplementary Data 5 and 6). According to the phylogenetic analysis based on the Genome Taxonomy Database (GTDB), only bacterial MAGs belonging to 10 different phyla were obtained (Fig. 2). Of these, 48.4% were affiliated with Firmicutes, followed by Proteobacteria (25.2%) and Actinobacteriota (10.7%). Meanwhile, 30 nearly complete MAGs (>98% complete) were obtained and two of these (*i.e.* MAG73 *Thermobifida fusca* and MAG406

*Saccharomonospora viridis*) were affiliated with the same species as the ACT ASVs.

To reveal the relative abundance and transcriptional activity profile of each MAG, the average genome coverage of each MAG and the average gene expression level (reads per kb per million mapped reads, RPKM) of all protein-coding genes in each MAG were calculated. Both MAG73 and MAG406 were <0.0001% in relative abundance and transcriptional activity at Day 00. In contrast, their relative abundance

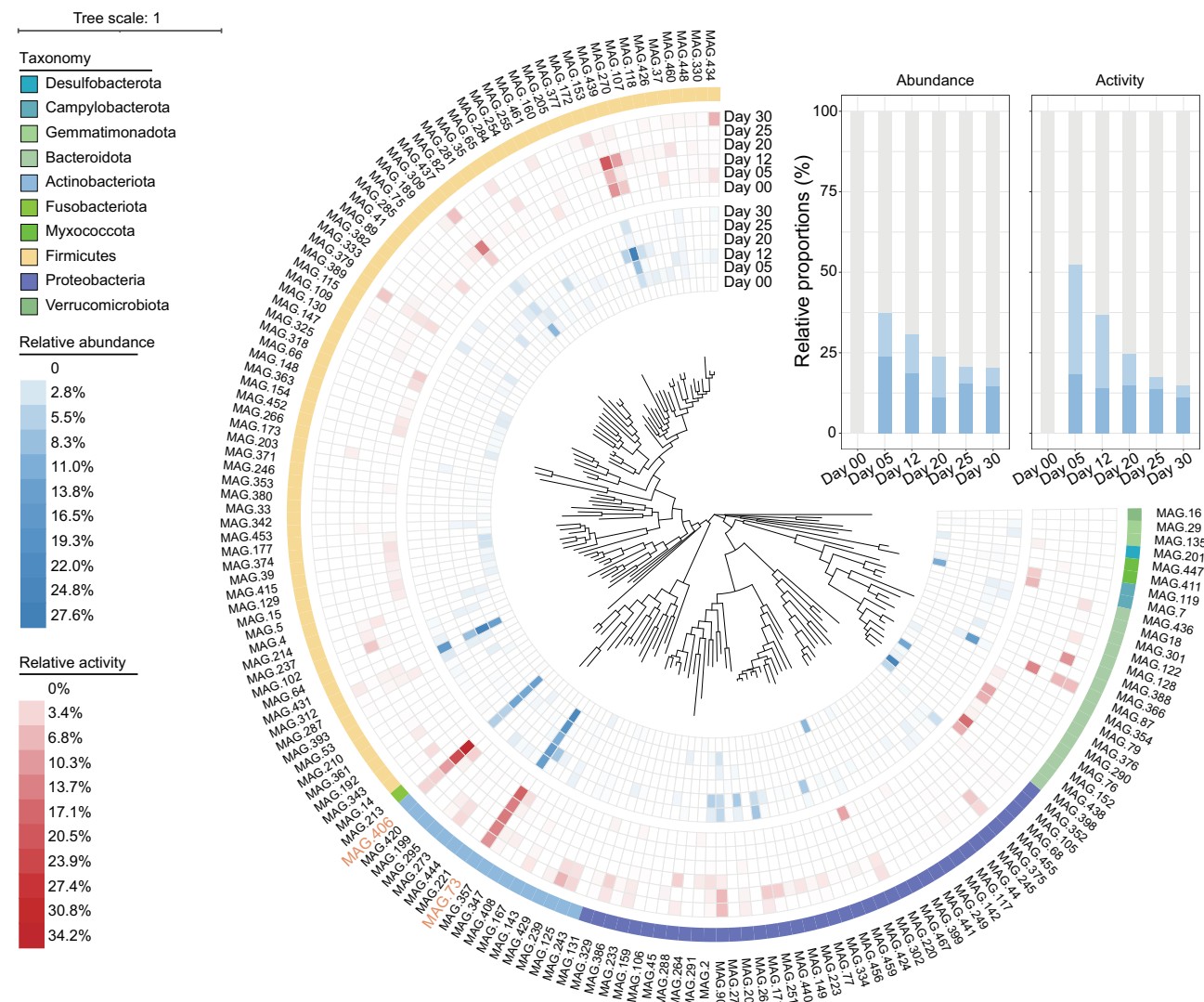

**Fig. 2 | Phylogenetic analysis of 159 high quality MAGs.** MAGs belonging to the same species as the ACT ASVs are marked in orange (i,e. MAG73 and MAG406). The blue and red shading of the inner arcs represent the relative abundance and relative activity of each MAG during each phase of composting, respectively. The details of the relative abundance and activity of MAG73 and MAG406 are highlighted in the histogram.

and transcriptional activity peaked at Day 05 (37.4%, 51.4%), followed by Day 12 (30.8%, 36.8%), and Day 20 (23.9%, 24.9%). Overall, these results indicated the importance of *Thermobifida fusca* and *Saccharomonospora viridis* on days 05, 12 and 20, when the average temperature exceeded 50 °C.

**Analysis of metabolic selection**

Genes in each MAG were annotated by KEGG Orthology (KO). Meanwhile, each KEGG module was checked for completeness according to its definition and 7 groups were obtained depending on the module completeness (Fig. S28 and Fig. S29 in SI). The grouping also showed some clear taxonomic divisions (Fig. S30 in SI). Groups I, II and VII were composed primarily of Firmicutes. Group III was composed solely of Gram (-) bacteria, primarily Proteobacteria. Bacteroidota and Firmicutes contributed equally to Group IV. Group V was also composed entirely of Gram (-) bacteria but from a wide range of phyla. Group VI consisted of MAGs solely of Actinobacteria affiliated organisms. Remarkably, both MAG73 and MAG406 belonged to this group. The similarity between functional grouping and bacterial taxonomy implied the unique metabolic potential of the various groups.

For synthesis, we focused on the KEGG modules related to amino acids (24 modules), vitamins and cofactors (18 modules), and lipids

and fatty acids synthesis (28 modules). A module was considered complete if over half of the group members had a module completeness above 67% (Fig. S31 in SI). A complete module indicated that the bacteria possessed this functional potential. To reveal the range of synthesizing abilities in different groups, we focused on the shared and unique modules between each group (Fig. 3a). A total of 21 modules were shared between >6 groups, with 7 modules being shared between all groups. Only Group VI had unique metabolic modules (*n* = 2), and both unique modules were associated with the aerobic biosynthesis of cobalamin (vitamin B12) (Fig. 3b). Thus, these results implied the uniqueness and importance of cobalamin biosynthesis in composting communities.

The relationship between supply and demand for cobalamin was further explored at the genomic and transcriptional levels (Fig. 3c, d). We used genes encoding cobalamin-dependent enzymes (*methionine synthase, metH, methylmalonyl-CoA mutase, mutA*, and *ribosomal small subunit methyltransferase, rsmB*) as indicators of cobalamin dependence (Text S6 in SI). Results showed that >96.2% of MAGs encoded cobalamin-dependent genes, indicating a high demand for cobalamin by composting bacterial communities. However, in contrast to the high demand for cobalamin, only 10.4% of the MAGs encoded the complete cobalamin biosynthetic pathway, with only two MAGs

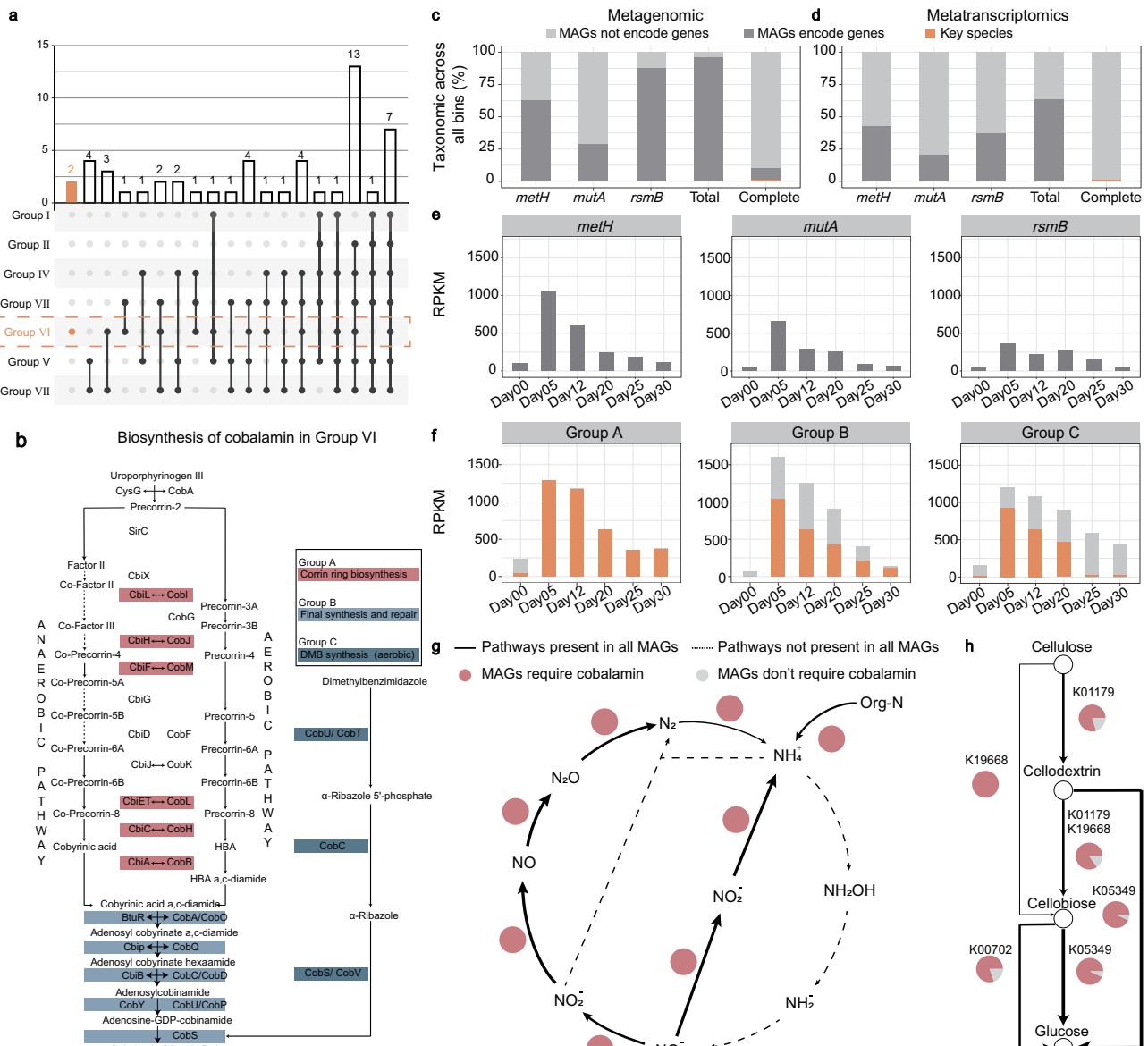

**Fig. 3 | The biosynthesis and demand for cobalamin during composting based on metagenomic and metatranscriptomic sequencing. a** The number of unique and shared modules related to the complete biosynthesis of amino acids, vitamins/co factors, and lipids/fatty acids between groups I–VII. Group VI, to which MAG73 and MAG406 belong, is marked in orange. Different sets are represented in different columns, with bars representing the number of sets and dots representing the composition (black: include, gray: exclude). For example, the second column refers that 4 modules (bar) are completed in both Groups V and VII (black point), but not in the other groups (gray point). **b** Three-step synthesis of cobalamin. **c** Proportion of MAGs encoding three cobalamin-dependent genes (i.e. *mutA*, *metH*,

and *rsmB*) and those encoding the complete cobalamin biosynthesis pathway. Total refers to the number of MAGs containing each of these three genes. **d** Proportion of MAGs transcribing the three cobalamin-dependent genes and those transcribing the complete cobalamin biosynthesis pathway. **e** RPKM of the three cobalamin-dependent genes at different sampling points. **f** RPKM values for the three-step synthesis of cobalamin at different sampling points. **g** Cobalamin requirement of MAGs transcribing nitrogen cycle-related functional genes. **h** Cobalamin requirement of MAGs transcribing cellulose degradation-related functional genes. The line thickness indicates the number of MAGs transcribing the functional gene.

(MAG73 and MAG406) transcribing the complete synthetic pathway. The sum of reads per kilobase of exon model per million mapped reads (RPKM) for genes encoding cobalamin-dependent enzymes peaked on day 05 (2074.9), followed by day 12 (1122.8) and day 20 (774.1), when the average temperature exceeded 50 °C (Fig. 3e). Meanwhile, the sum of RPKM for genes associated with cobalamin biosynthesis also peak on these days (Day 05, Day 12 and Day 20), and were 1.8-8.8 times higher than others. These results suggested that the demand for cobalamin might rise with abiotic stress (high temperature). *Thermobifida fusca* and *Saccharomonospora viridis* were the main suppliers under high temperature (>50 °C) as they contributed nearly 72.3% (on

average) to the biosynthesis of cobalamin on days 05, 12, and 30 (Fig. 3f). Therefore, these results pointed to a higher demand for cobalamin at high temperature, and highlighted the importance of *Thermobifida fusca* and *Saccharomonospora viridis* to the bacterial community due to their unique ability to biosynthesize cobalamin. To confirm the effect of cobalamin on functional potential, MAGs encoding genes related to the nitrogen cycle and cellulose degradation were further screened and tested for their dependence on cobalamin (Fig. 3g, h). Over 83.3% of these MAGs encoded cobalamin-dependent genes. Meanwhile, more than 57.1% of these functional species were affiliated with Firmicutes, supporting the results based on the

correlation and best multiple regression models (Fig. S11 and Fig. S25 in SI and Supplementary Data 8–9). Thus, given the higher abundance and transcriptional activity of *Thermobifida fusca* and *Saccharomonospora viridis* under high temperature (>50 °C) and their unique biosynthetic capacity, these species might share cobalamin as public goods to cooperate with Firmicutes.

## Validation of microbial interactions with pure culture

The effects of *Thermobifida fusca* and *Saccharomonospora viridis* (Key group; type strains purchased to be more generality) on Firmicutes (40 strains) were tested, with *Sphingomonas paucimobilis, Lactobacillus brevis* and *Idiomarina andamanensis* as controls (General group) (Fig. S32 in SI). Firstly, we collected the supernatants from the General and Key groups and tested their effect on Firmicutes at 37 °C and 50 °C, respectively. We also set up a Cobalamin group by adding 1.5ppm cobalamin to confirm its effect on Firmicutes. We focused our analysis on the 72-h time point by which cultures typically reach saturation[9]. These tests showed that the Key group and Cobalamin group could significantly enhance the abundance of Firmicutes strains

by 1.6 and 3.5 times, respectively (Fig. 4a). By contrast, no significant difference was observed between the General group and the blank control group. Secondly, to further confirm whether these differences were caused by cobalamin, we tested the uptake capacity of Firmicutes and the synthesis capacity of the Key group (Fig. S33 in SI). Results showed a remarkable reduction in the concentration of cobalamin in the supernatants of the tested strains (nearly 90% reduction, Fig. S33a in SI). Moreover, the accumulation of cobalamin could be detected in the supernatants of the Key group after 72 h of incubation (-750.6 ppb, Fig. S33b in SI). In contrast, we could only detect minimal signals (<10 ppb) in the General group, which may have been introduced by the yeast extract of the medium. To confirm whether other metabolites showed the patterns observed for cobalamin, we utilized riboflavin as a control, as it is a biosynthetic vitamin like cobalamin (Fig. S34 in SI). The results presented the exact opposite of those for cobalamin, that riboflavin could be secreted by all species tested (Fig. S34 in SI). Overall, these results indicated that *Thermobifida fusca* and *Saccharomonospora viridis* secreted cobalamin, promoting the growth of Firmicutes.

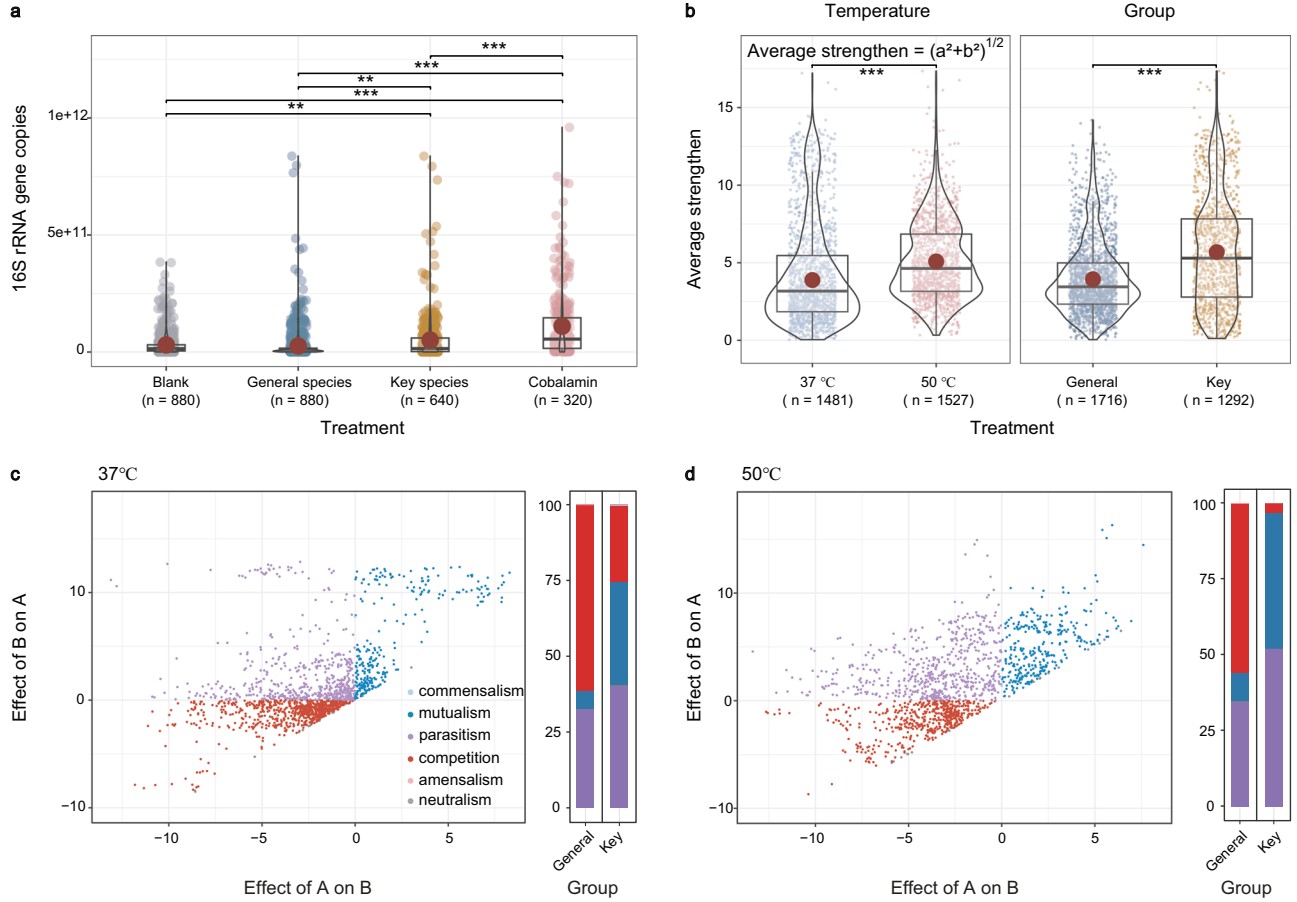

**Fig. 4 | Microbial interactions between pure culture strains. a** The growth yield of 40 Firmicutes strains after the addition of supernatants and cobalamin. The blank group received only LB medium (*n* = 880 biologically independent samples). The General group contained supernatants collected from *Sphingomonas paucimobilis, Lactobacillus brevis*, and *Idiomarina andamanensis* (10% (v/v)) after 72 h culture, respectively (*n* = 880 biologically independent samples). The key group contained supernatants collected from *Thermobifida fusca* and *Saccharomonospora viridis* (10% (v/v)), respectively (*n* = 640 biologically independent samples). The cobalamin group received additional LB medium (10% (v/v)) which contained 1.5 ppm cobalamin (*n* = 320 biologically independent samples). In the boxplots of panels, hinges indicate the 25th, 50th, and 75th percentiles, whiskers indicate 1.5 × interquartile ranges, and dots indicate values of individual samples. Unadjusted *p*-values of the one-way ANOVA are labeled as ***$p$ < 0.001, **$p$ < 0.01.

**b** The average strength of interactions for different temperatures and groups (***$p$ < 0.001). In the boxplots of panels, hinges indicate the 25th, 50th, and 75th percentiles, whiskers indicate 1.5 × interquartile ranges, dots indicate values of individual samples, and red points indicate mean values. A total of 3008 biologically independent samples were included and divided into different treatment, based on temperature (37 °C and 50 °C) and groups (General and Key). Significant comparisons (two-sided *t*-test) between different temperature and different groups are indicated by ***$p$ < 0.001. **c** The detail of interaction types under 37 °C. The stacked bar indicates the interaction types between the Key group and the General group. Bar colors indicate the type of interaction. **d** The detail of interaction types under 37 °C. Bar colors indicate the type of interaction. The legends are the same as for Fig. 4c.

Further, we co-cultured Firmicutes with Key group and General group under different conditions (medium: LB medium and compost leachate (CL); temperature: 37 °C and 50 °C) to confirm the variations in interaction types (Supplementary Data 10). Each bidirectional interaction was described quantitatively (average strength) and qualitatively (interaction classification) using Kehe's method[9]. Results showed that the average strength peaked at 50 °C, which was 1.2 times higher than at 37 °C (Fig. 4b). The average strength in the key group was also 1.4 times higher than the general group. Mutualistic relationships in the bacterial pairs in the key groups were 33.8% and 44.3% at 37 °C and 50 °C, respectively, which was 4.7–5.5 times higher than that in the general group (Fig. 4c, d). Meanwhile, the key group had significantly lower competitive relationship at 37 °C (24.9%) and 50 °C (3.2%) than the general group (60.6%, 54.7%), decreasing by 35.7% and 51.6% respectively. Overall, mutualism peaked at high temperatures and could be enhanced through the secretion of cobalamin by *Thermobifida fusca* and *Saccharomonospora viridis*.

## Discussion

Based on 10 repeated observations in a continuous and semi-sealed habitat, our study provided a body of evidence that stress induced by high temperature would strengthen positive co-occurrence (potential mutualism) in an overlooked biosystem characterized by deterministic temperature variation. Moreover, high temperature stress would select the species that could promote mutualism through cobalamin sharing. To overcome the limitations of co-occurrence analysis, we also verified this conclusion by pure culture.

The positive correlation observed between taxa might be driven by two mechanisms, microbial interactions and environmental filtering. When influenced by microbial interactions, the positive correlation was caused by mutualism[29,30]. In terms of environmental filtering, the positive correlation indicated ecological similarity[31]. Based on various statistical approaches (Figs. S8–S10, and Text S2.3 in SI), the results consistently showed that the observed positive correlations were mainly caused by the potential mutualism rather than ecological similarity. The results of statistical approaches (Note 3 in SI) and in vitro work (Fig. 4) emphasized that the increased temperature could also promote the cooperative behaviour. Although the stress gradient hypothesis (SGH) might be a probable explanation for the increase of cooperative behaviour under environmental stress. However, the SGH focuses on systems in which all species can survive within a range of variations in the 'focal' environmental stress and the rapid temperature rise to 70 °C was undoubtedly a selection pressure for bacteria. Thus, we suggested that the increase of cooperative behaviour may have been caused by the temperature selection on the bacterial community.

The distribution of fast and slow growers is a powerful trait-based description of the structure of a bacterial community[26,32]. We found a significantly negative correlation between MCN and temperature range in the tested composting piles, suggesting that higher temperature would select for slower growing bacteria. However, this negative correlation may have depended on nutrients. Therefore, we constructed a multivariate regression model (Fig. S26 and Table S6 in SI) to test whether the inclusion of nutrients would influence the temperature coefficient, the results of which showed that temperature was the most important factor in explaining the variation in fast and slow growing bacteria. These results were consistent with previous studies from soils[33] and oceans[26]. Due to strong temperature selection, the slow-growing and stress-tolerant species would be selected[34], and these species might provide direct benefits for other community members[35].

The importance of *Thermobifida fusca* and *Saccharomonospora viridis* in mutualism was also supported by the different effects of the Key group and the General species on Firmicutes (Fig. 4). Mutualism occurred more commonly between the Key group (*Thermobifida fusca* and *Saccharomonospora viridis)* and Firmicutes (33.8–44.3%), which

was 4.7–5.5 times higher than that of the General group (3.2–35.7%). The rate of mutualism between *Thermobifida fusca* and *Saccharomonospora viridis* and *Firmicutes* spp. was higher than that between pairs of bacteria in tree holes (10.0%)[36], soil (5.2%)[9], and the guts of *Caenorhabditis elegans* (2.5%)[8], humans (1.4%)[7], and mice (0%)[37]. This result was consistent with that obtained from metal-working-fluid bacteria (48.9%)[10]. *Thermobifida fusca* and *Saccharomonospora viridis* are thermophilic microbes with a lower rrn copy number (Fig. S24 in SI) and defined by slow growth and stress tolerance[14]. In contrast to microbes with a high rrn number, those with a low rrn number might be more likely to cooperate with others[14]. The most probable mechanism was that *Thermobifida fusca* and *Saccharomonospora viridis* could secrete public goods which Firmicutes strains would then use to increase their growth yield. This notion could be supported by theoretical predictions such as the Black Queen Hypothesis (BQH) which states that losing the production of "expensive" public goods is evolutionarily selected for and can result in obligate cross-feeding[38,39]. Moreover, the metabolism of public goods does occur under nutritional conditions[40]. It is undeniable that *Thermobifida fusca* and *Saccharomonospora viridis* may secrete carbon degrading enzymes to promote the growth of facilitated strains. It is more common during the degradation of polysaccharides, but it might not be appropriate in more nutrient rich medium, such as those that contains yeast extract and simple carbon sources, as these simple carbon sources would be preferred. Overall, the sharing of public goods may well be the optimal strategy.

Our results suggested that cobalamin might be the public good shared by *Thermobifida fusca* and *Saccharomonospora viridis* (Fig. 4 and Fig. S33 in SI). The supply and demand of cobalamin in the composting system was imbalanced, as only these two species could completely transcribe entire biosynthetic pathway. Extensive microbial dependence on cobalamin has also been observed in soil[41] and the marine enviornment[42], where >80% of the microbes require cobalamin but are unable to synthesize it. The biosynthesis of cobalamin requires >30 enzymatic steps, constituting a high metabolic burden for microbes[43]. The imbalance in soil and marine systems could be supported by the BQH that most microbes would live more efficiently by reducing the number of burdensome genes they carry and benefit from the products of other microbes[39]. Although the composting community experienced the heating-cooling cycle only one time, in addition to selecting target species, high temperatures might also act as a selective force to promote cooperation evolution among species. *Actinobacteria* were the main cobalamin producers in our composting system, as compared to the main producers in soil and the marine environment being Proteobacteria[41] and Thaumarchaeota[42], respectively. Different producers endow cobalamin with a specificity that sets them apart from other vitamins. The cobalamin secreted by *Thermobifida fusca* and *Saccharomonospora viridis* was benzimidazolyl cobalamin (Fig. S34 in SI), which could be consumed by the tested Firmicutes strains (Fig. S33 in SI). Consistent with our results, almost all previously tested Firmicutes genera prefer benzimidazolyl cobalamin[44]. As a shared nutrient, cobalamin is notable because it could substantially affect microbial growth at low external concentrations, even at the picomolar level[45].

Cobalamin has also been shown to govern a variety of microbial processes, including the regulation of gene expression[46], replication and repair of DNA[47], tricarboxylic acid (TCA) cycle[48], amino acid synthesis[43] and $CO_2$ fixation[49]. The exchange of cobalamin forms a potential sociomicrobiology, implying a two-way physiological feeding pathway between *Thermobifida fusca* and *Saccharomonospora viridis* and others organisms[50]. Firstly, these two species might reduce intracellular pressure via over-synthesis of cobalamin, as *Nitrosopumilus* over-synthesize $NO_2$-cobalamin to reduce the stress caused by the accumulation of NO and nitrite in cells[51]. Secondly, *Thermobifida fusca* and *Saccharomonospora viridis* might depend on Firmicutes to shape

the environment, given that these species require a high temperature for growth. The fasting-growing and resource-wasting microbes (*i.e.* Firmicutes) would release additional energy into the compost to raise the temperature, which may benefit these species, as previously described in a biofilm[52]. Thirdly, intact pathways related to the biosynthesis of glutamate and methionine were not detected (<10% module completeness) in *Thermobifida fusca* and *Saccharomonospora viridis*. In contrast, most of the MAGs that contained complete biosynthesis pathways for these two amino acids were affiliated with the Firmicutes. Meanwhile, the biosynthesis of glutamate and methionine is dependent on cobalamin. Thus, the exchange of cobalamin and amino acid(s) (*e.g.*, glutamate and methionine) might be the strategy for sharing cobalamin in *Thermobifida fusca* and *Saccharomonospora viridis*.

Virtually all engineered habitats are exposed to self-produced or natural stress, such as pH, chemical oxygen demand (COD), antibiotics, heavy metals, etc. The rapid adaptation of microbial communities to their environment is a matter of concern. It is far from feasible to culture and regulate thousands of species, but by focusing on regulating slow-growing and stress-tolerant species it may be possible to promote mutualism to appreciate the unique metabolic pathways benefiting their partners. However, due to the limitation of sample numbers, the constructed network was built based on only 10 input samples, which represented the average of all 6 time points for a given pile. There is a need yet to further verify these findings in other chronosequences or composting processes as well.

## Methods

### Location and sample collection

Samples were taken from 10 composting piles (between May and August 2020) at a food waste (FW) composting facility (Zhejiang, China; 30°52′51.50″N; 119°46′12.87″E). A consistent pre-treatment method was used for each pile. Specifically, about 8 tons of FW was collected every 3 days and subsequently crushed into 30 ~ 50 mm pieces. The moisture content was controlled at around 70% with a press and the FW was mixed with straw (bulking agent) at a ratio of 1:3. The volume of each pile was controlled at $3 \times 3 \times 1.2$ m. Air was supplied by air blowers with the aeration rate maintained at 0.9 L/kg$^{-1}$ dry matter (DM) $_{initial}$ min$^{-1}$. The physicochemical properties of the raw materials are shown in Supplementary Data 11. Sampling was carried out on days 0, 5, 12, 20, 25, 30 depending on the changes of temperature (Text S1.1 in SI). To eliminate heterogeneity, each sample consisted of five sub-samples, including the four corners and the center of the pile. Overall, a total of 60 samples were collected.

### DNA extraction and high-throughput sequencing

60 samples of the composting microbiome (including 10 different piles, 6 different sample times) were extracted with the DNeasy PowerSoil Kit (12888-50, Qiagen, Germany) following the manufacturer's instructions, in triplicate. The purity and concentration of the DNA were tested using a Qubit 2.0 (ThermoFisher Scientific, USA) and Nanodrop One (ThermoFisher Scientific, USA), respectively. The V4 region (515 F′ GTGCCAGCMGCCGCGGTAA, 806 R′ GGACTACHVGGGTWTCTAAT) of the 16 S rRNA gene was amplified, and triplicate PCR products were mixed and purified with ENZA Gel Extraction Kit (Omega Bio-Tek Inc., Norcross, GA, USA). We used the NEBNext® Ultra™ DNA Library Prep Kit for Illumina (New England Biolabs, Ipswich, MA, USA) to generate the sequencing library, following manufacturer recommendations. Finally, the library was sequenced by Illumina MiSeq (MAGIGENE Biological Technology Co. Ltd Guangzhou, China) to generate 250 bp paired-end reads[14]. The details of the reaction system of PCR before sequencing are described in Table S7. Divisive Amplicon Denoising Algorithm 2 (*DADA2*) was used to differentiate the 16 S rRNA gene ASVs by de-duplication or clustering at 100% similarity only. Afterwards, QIIME2 (v2020.11.0) was used for downstream data analysis. 110,406 ASVs were identified and resampled to create a standardized ASV table with 12,507 reads per sample. In conclusion, a total of 180 high-throughput sequencing samples were measured for 60 samples (3 replicates per sample), and the average of the parallel samples was representative of the sampling time point.

### High-throughput qPCR

A total of 60 DNA samples were analyzed by high-throughput qPCR for 54 functional genes (including carbon degradation, carbon fixation and nitrogen cycling) on a Wafergen Smart Chip Real-Time qPCR platform as previously described[47], with a threshold cycle of 31 as the detection limit[53], in triplicate. Details of tested genes and corresponding primers are provided in the supplementary material (Supplementary Data 12). The same gene was considered a single unique functional gene when targeted by multiple primer sets. The abundance of functional genes for each DNA sample was calculated based on the Pfaffl method[49].

### Metagenomic sequencing and data analysis

Since bacterial communities were clustered according to different sampling time points, 180 DNA samples were mixed according to the phase (10 to 1 per composting phase). Overall, 18 DNA samples (6 different sampling time points with 3 replicates each) were obtained for further metagenomic sequencing. Sequencing libraries were constructed according to the product instructions of NEB Next® Ultra™ DNA Library Prep Kit for Illumina® (New England Biolabs, MA, USA) and sequenced on the Illumina NovaSeq 6000 platform with a 150 bp paired-end reads strategy by a sequencing server (MAGIGENE Biological Technology Co. Ltd Guangzhou, China). On average, 10 GB raw data and $1.23 \times 10^{10}$ bases were obtained per sample ($n = 18$). The raw data were processed using Trimmomatic (v.0.36) to discard bases with lengths <50 and mass fractions <20 base pairs[54]. Approximately, $8.0 \times 10^9$ clean bases (Q20 = 100%, Q30 = 99.7%) were obtained. All clean data was co-assembled by MEGAHIT (v1.0.6, parameters: k-min 35, k-max 115, k-step 20) to obtained Scaffold[55]. To remove chimeric sequences, the Scaffold was rigorously separated at N (unknown bases). Only those contiguous sequences that were >2500 bases in length and free of N could be extracted as a Scaftig. Scaftigs were automatically assembled using MetaBAT2 with a sensitive parameter[56]. Quality of the obtained MAGs was assessed by CheckM[57], and evaluated using the MAG quality standard developed by the Genomic Standards Consortium[26]. In detail, MAGs could be classified into 3 quality groups based on their completeness and contamination, including high-quality MAGs (completeness > 90%, contamination < 5%), medium-quality MAGs (completeness ≥ 50%, contamination < 5%), and low-quality MAGs (completeness < 50%, contamination < 10%). The taxonomic and phylogenetic analyses were carried out using the Genome Taxonomy Database Toolkit (GTDB-Tk; v 1.7.0)[58] and visualized in iTOL v3[59]. The relative abundance of MAGs was calculated by the proportion of the average coverage of each MAG in a metagenomic sequencing sample. In conclusion, a total of 18 metagenomic sequencing samples were measured for 6 sampling time points (3 replicates each), and the average of the replicate samples was representative of a sampling time point.

### Metatranscriptomics sequencing and data analysis

RNA of the composting microbiome was extracted with RNeasy Powersoil (12888-25, Qiagen, Germany), according to the manufacturer's instructions. The quality and quantity of RNA was measured by Qubit 3.0 (ThermoFisher Scientific, USA) and Nanodrop One (ThermoFisher Scientific, USA) at the same time. RNA integrity was checked using the Agilent 4200 system (Agilent Technologies, Waldbron, Germany). The mRNAseq libraries were generated using NEB Next® Ultra™ Non-directional RNA Library Prep Kit for Illumina® (New England Biolabs,

MA, USA) following the manufacturer's instructions. The constructed library was sequenced on an Illumina Novaseq6000 platform (MAGIGENE Biological Technology Co. Ltd Guangzhou, China), and 150 bp paired-end reads were generated. The transcript data were aligned to the assembled MAGs with Bowtie2 (v2.33), and quantified with corset (v1.06)[60]. Thus, the metatranscriptomes inherited the corresponding MAGs' annotations, without de novo assembly of the RNA sequences. Moreover, to allow for the comparability of expression levels between various genes, RPKM was calculated[61]. The relative activity of the MAGs was similarly calculated, except that the relative activity was based on RPKM. In conclusion, a total of 18 metatranscriptomic sequencing samples were measured for 6 sampling time points (3 replicates each), and the average of the replicate samples was representative of a sampling time point.

### Molecular ecological network analysis

A molecular ecological network (MEN) was constructed based on 10 sample inputs that represented the average of all 6 time points for a given pile. To ensure the reliability of the constructed network[62], only ASVs present in at least 5 out of the total 10 samples were included for further correlation calculations. The network was constructed based on the Pearson correlations of log-transformed OTU abundances. Afterward, an RMT-based approach was further employed to determine the correlation cut-off threshold, as it could minimize the uncertainty in network construction and comparison[63]. To assess the significance of the constructed network, 100 random networks were constructed by randomly reconnecting links between nodes, following the Maslov–Sneppen procedure[64]. The topological indices of the constructed network and random networks were calculated on the MENAP interface (http://ieg4.rccc.ou.edu/MENA/). For each node included in the constructed network, its $Z_i$ value (i.e., within-module connectivity) and $P_i$ value (i.e., among-module connectivity) were calculated[65]. The sub-network between ACT ASVs and their neighboring ASVs was extracted using Cytoscape (v3.8.2). Whole networks were visualized through Gephi (v0.9.2)[66].

### Cohesion index calculation

To reveal both positive and negative co-occurrences, the cohesion index was calculated, which could characterize the potential interactions by the similarities and differences in the niches of various taxa[12]. Specifically, the null model-corrected correlations for each taxon were used for further analysis, and were calculated by the differences between pairwise correlations based on the relative abundance and pairwise correlations based on the multiple null model iterations. The recommended 'taxa shuffle' null model was used. Connnectedness for each taxon referred to the average of positive and negative correlations. The positive and negative cohesion of each sample were the respective sum of the product of positive and negative connnectedness and the relative abundance for each species. Overall, the cohesion index was calculated from the null model-corrected correlations and abundance-weighted, and, as such, was considered as a promising method to measure the strength of biological interactions[12,63].

### Screening for ACT ASVs

Four criteria were used to screen the ACT ASVs[67], including overall abundance (being among the top 0.1% ASVs in mean relative abundance)[22], ubiquity (detected in over 80% samples)[23], and being frequently abundant (the relative abundance being top 80% in >50% samples)[24], and PosCoh related ASVs (ASVs with significant impact on the changes of positive cohesion). PosCoh related ASVs were identified by random forest model. In this random forest model, the relative abundance of each ASV served as predictor for positive cohesion. To estimate the importance of these indexes, we used percentage increases in the MSE of variables: higher MSE% values imply variables with greater importance[25]. The significance of each predictor on the

response variables was assessed with the "*rfPermute*" R package. 500 decision trees were set, followed by 1000 random permutations and 5 repetitions of 10-fold cross-validation.

### The isolation of strains and co-culture

To isolate composting bacteria, samples from different phases were collected and mixed with PBS buffer at a ratio of 1:10. All mixed samples were shaken for 1 h at 150 rpm on a rotary shaker in the dark at 30 °C and 50 °C, respectively. Dilution series ($10^{-6}$) were plated on LB medium solidified with 1.5% agar. After a 48-h incubation at 30 °C and 50 °C in the dark, 118 colonies were picked and phylogenetically characterized by colony PCR using a universal primer set (27 F and 1492 R) for the 16 S rRNA gene[68]. 41 strains, including 40 Firmicutes strains and *Sphingomonas paucimobilis* were isolated. Moreover, we used De Man Rogosa Sharpe (MRS) medium for the isolation of *Lactobacillus brevis* and 2216E medium for the isolation of *Idiomarina andamanensis* at 25 °C and 30 °C, respectively (Text S7.3 in SI). Finally, the 43 isolated strains were stored at −80 °C in 15% glycerol. The type strains of *Thermobifida fusca* and *Saccharomonospora viridis* were purchased from the Deutsche Sammlung von Mikroorganismen und Zellkulturen (DSMZ, German) to confirm and demonstrate the generality of the ability to synthesize cobalamin within these species and not unique to the strains isolated from composting. Results of phylogenetic analysis (whole length of 16 S) and comparative genomics showed the consistency of the purchased species with MAGs in the compost (Fig. S36-S37, Text S7.2 and Table S8-S9 in SI). We divided the 45 strains into 3 groups, including the Key group (*Thermobifida fusca* and *Saccharomonospora viridis*), the General group (*Sphingomonas paucimobilis*, *Lactobacillus brevis*, and *Idiomarina andamanensis*), and the Firmicutes group (40 Firmicutes strains).

Firstly, we tested the effect of the supernatants and cobalamin on Firmicutes in LB-Lennox medium (peptones 10 g, yeast extract, 5 g, NaCl 5 g per litre). The supernatants were collected from all strains in Key and General groups, after 72 h of culturing in LB medium with 1ppm cobalt (by adding 2.188 mg / L CoCl$_2$), which is key to the production of cobalamin[43]. Benzimidazolyl cobalamin (Adamas, >98%) was used as the standard due to its structural similarity to the cobalamin secreted by *Thermobifida fusca* and *Saccharomonospora viridis* (Fig. S34 in SI). We setup four groups in this part, including (1) the Blank group: only LB medium (with additional 10% of LB medium (v/v)), (2) the General group: LB medium + supernatants collected from *Sphingomonas paucimobilis*, *Lactobacillus brevis*, and *Idiomarina andamanensis* (General group), respectively (10% (v/v)), (3) the Key group: LB medium + supernatants collected from *Thermobifida fusca* and *Saccharomonospora viridis* (Key group), respectively (10% (v/v)), (4) the Cobalamin group: LB medium + cobalamin (10% (v/v), 1.5ppm). Each group was tested at both 37 °C and 50 °C. Overall, 8 groups were tested (Table S10 in SI). All samples were uniformly collected at 72 h and the 16 S rRNA gene abundance was measured by qPCR to characterize the overall growth yield for all groups. The details of the specific primers and the reaction system are described in Table S11.

Secondly, we tested pairwise effects between different groups. 5 co-culture systems were set up under both 37 °C and 50 °C, including (1) 40 Firmicutes strains + *Thermobifida fusca*, (2) 40 Firmicutes strains + *Saccharomonospora viridis*, (3) 40 Firmicutes strains + *Sphingomonas paucimobili*, and (4) 40 Firmicutes strains + *Lactobacillus brevis*, (5) 40 Firmicutes strains + *Idiomarina andamanensis*. Pure cultures of all strains were used as controls. Although the carbon sources had less effect on mutualism[9], we tested both the LB medium and the composting leachate (CL, collected from composting) to confirm our results. Overall, we set up 5 (strains) * 2 (temperature) *2 (medium) = 20 groups for this experiment (Table S12 in SI). All treatments were tested with 3 biological replicates and 3 technical replicates. Specifically, as *Lactobacillus brevis* is classified as a Firmicutes, the abundance of Firmicutes in bacterial pairs related to *Lactobacillus brevis* were

calculated by the difference between Firmicutes and *Lactobacillus brevis*.

Thirdly, we selected 10 Firmicutes, which had positive interactions with *Thermobifida fusca* and *Saccharomonospora viridis*, and cultured them in LB medium containing 1.5 ppm cobalamin. Cobalamin is used in a wide range of concentrations, ranging from 0.0005 to 20 ppm when culturing bacteria[58,60,69]. We used 1.5 ppm because we found that cobalamin residues only at concentration greater than or equal to 1.5 ppm (Fig. S33 in SI), which means that it meets the requirements for Firmicutes growth. Meanwhile, we cultured Key and General group species in LB medium containing 1.0 ppm Co to reveal the accumulation of cobalamin (Table S13 in SI). Samples were taken at 0, 0.5, 1, 2, 3, 4, 5, 6, 12, 24, 48, 72 h and centrifuged to obtain supernatants. Before detection, the supernatants were filtrated though a polyvinylidene fluoride membrane (0.22 μm) and diluted 50x to avoid signal suppression due to the matrix effect. After that, the supernatants were analyzed with an ultra-performance liquid chromatography-triple quadrupole mass spectrometer (XEVO TQ MS, Waters, UK). In detail, the ACQUITY UPLC HSS T3 column (1.7 μm, 100 mm × 2.0 mm) (Waters, Manchester, UK) was used with a column temperature of 30 °C. Formic acid (0.1% (v/v)) was used as solvent A and acetonitrile was used as solvent B. Gradient elution was achieved by solvent A: 0 ~ 3.0 min (80% ~ 65%), 3.0 ~ 4.0 min (65% ~ 20%), 4.0 ~ 5.0 min (20% ~ 65%), and 5.0 ~ 6.0 min (65% ~ 80%). The injection volume was 10.0 μL and the flow rate was 0.2 mL min$^{-1}$. The tandem triple quadrupole mass spectrometry was set for electron spray ionization (ESI) and multiple reaction monitoring (MRM). The positive electrospray ionization mode was used and the ion source temperature was 150 °C. Only characteristic ions satisfying all the setting parameters were considered as the test cobalamin (Table S14 in SI).

### The classification of interactions

The type of interaction was calculated by the difference in growth yield (i.e., fluorescence intensity obtained from qPCR) between co-culture and monoculture. The x-axis was the difference between the yield of A in coculture and monoculture, and the y-axis was the difference between the yield of B in coculture and monoculture. Further, this point would be plotted on a polar coordinate. The angle (Θ) and magnitude (m) of this point quantified the interaction type and the interaction strength, respectively[9]. In detail, the line $y = -x$ was assigned to 0°. Values $-90° < Θ < -45°$ indicated competition, $Θ = -90°$ indicated equally inhibited, $Θ = -45°$ indicated amensalism, $-45° < Θ < 0°$ indicated parasitism, $0° < Θ < 45°$ indicated parasitism, $Θ = 45°$ indicated commensalism, $45° < Θ < 90°$ indicated mutualism, and $Θ = 90°$ indicated equally facilitated. Interaction strengthen was calculated by the formula listed in Fig. 4e.

### Reporting summary

Further information on research design is available in the Nature Portfolio Reporting Summary linked to this article.

## Data availability

The 16 S rRNA gene high-throughput sequencing, metagenomic sequencing, and metatranscriptomics sequencing data generated in this study have been deposited in the National Center for Biotechnology Information (NCBI) Sequence Read Archive (SRA) database under under accession number PRJNA877822, PRJNA878660, and PRJNA897831. The database used for rrn copy number estimation is available online (https://rrndb.umms.med.umich.edu/). Source data are provided as a Source Data file. Source data are provided with this paper.

## Code availability

The R script for screening for screening overall abundant ASVs, ubiquitous ASVs, frequently abundant ASVs and PosCoh related ASVs is

publicly available on GitHub at https://github.com/Yuxiang-Zhao/Composting or under Zenodo at https://doi.org/10.5281/zenodo.8260129.

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

## Acknowledgements

This work is supported by the National K&D Plan of China [2019YFC1905003 to B.H., 2019YFC1905001 to Y.D.]. We thank the Magigene Biotechnology. Thanks to James Voordeckers for his contribution to the language. Thanks to Miaolian Hua for her contribution.

## Author contributions

Y.Z.: Conceptualization, methodology, visualization, writing—original draft; Z.L.: writing—review and editing; B. Z., J.C., X. Y.: methodology, investigation; M.Z: writing—review and editing; Y.D.: writing—review and editing; B. H.: funding acquisition, conceptualization, writing—review and editing.

## Competing interests

The authors declare no competing interests.
