## [Peer Review File · Nature Communications]

Inter-bacterial mutualism promoted by public goods in a system characterized by deterministic temperature variationReviewer #1 (Remarks to the Author):

In the manuscript "Inter-bacterial mutualism promoted by public goods in a poikilothermic biosystem", Ning and coauthors described compositional turnover in compost communities as temperature varies over the course of 30 days. The authors measured positive cohesion, indicative of positively correlated abundances between the most abundant taxa in the community, and found that it increases with temperature, therefore peaking during the thermophilic phase of the composting process. The authors ascribe increases in positive cohesion to the prevalence of mutualistic interactions among two key Actinobacteria taxa supplying cobalamin and several Firmicutes taxa, possibly ameliorating growth conditions for the Actinobacteria.

I think the presented results are of interest for microbial ecologists and scientists working at the interface with applied microbiology. However, I would like the authors to clarify and expand on the following points:

1. While increasing temperatures can affect the outcome of species interactions and favor slower-growing taxa (see Lax, Abreu & Gore Nat Ecol Evol 2020 and Abreu, Dal Bello et al. bioRxiv 2022), the switch from 20 to 70C can actually filter out several species due to their thermal ranges. Do the authors know which % of the species in the initial community does not deterministically survive when temperature is above 50C? The claim reported in the discussion starting at line 278 confounds these two scenarios and needs to be backed by data.
2. The compositional turnover observed in this compost communities is consistent with a *succession*, which is one of the scenarios in which SGH might operate. However, I believe that the SGH applies more to systems in which the range of variation in the 'focal' environmental stressor is within the range ensuring the survival of all species. This might true for the depletion of nutrients during the composting process, but not for temperature, for the reasons highlighted in the previous point. This, however, does not detract from the finding that mutualistic interactions are prevalent in the thermophilic compost communities.
3. I am very intrigued by the fact that, during the thermophilic and cooling phases, slower-growers like Actinobacteria become are more abundant at higher temperatures, despite the general idea that slower-growing taxa usually benefit from a nutrient-poor environment (possibly like the one in the cooling phase). This is consistent with results from the two studies I reported above (Lax, Abreu & Gore Nat Ecol Evol 2020 and Abreu, Dal Bello et al. bioRxiv 2022). I think it would be interesting if the authors could plot the weighted mean copy number (see Abreu, Dal Bello et al. bioRxiv 2022, Wu et al ISME 2017, Nemergut et al. ISME 2016) of their communities in the thermophilic and cooling phases as a function of temperature and nutrient concentration.
4. How is community diversity changing during the composting process? Do the positive cohesion enhance diversity?

While results appear interesting, the manuscript quite difficult to read because crucial details about why and how a certain analysis is performed are missing. I believe the reader should be able to the get a sense about these details without going to the Methods section. As a general suggestion, starting each results section explaining why an analysis is performed, briefly describing how, and then presenting the results would improve a lot the readability of this manuscript.

Detailed comments:

Introduction:

- Lines 48-55: It might be useful to talk about positive interactions in more general terms to include also commensalism, which has been found to be quite common among soil bacteria (Kehe et al. Sci Adv. 2021)

- Line 69 "proper" might not the right word to convey the need for a system that is closer to natural conditions

- Lines 70-88: In describing compost as a model system, the authors never mention the term that

is used in the title and abstract (poikilothermic): I suggest to not use this term, because usually it is referred to animals that cannot control their internal temperature. I'd rather say that that compost is system characterized by deterministic temperature variation or something along these lines.

Results:

- I find intriguing that different communities sampled in the same day during the thermophilic phase can display a wide range of temperatures (between 30 to 70C). It would be interesting to look at temperature trajectories of individual communities to assess the variability in the time it takes to switch between phases, and whether interactions or community composition can explain such variability.
- Given the variability of the temperature of the different communities within the different phase, especially the thermophilic one, I find panel B of Fig. 1 quite misleading.
- It would be also curious to see variations in positive (and negative) cohesion in the 10 composting piles (basically the same plot of Fig. 1a but with cohesion on the y axis).
- Line 105-110: This paragraph is quite vague and assumes all reader are familiar with iCAMP analyses. I strongly recommend that the authors briefly explain in the results section what tools of the iCAMP analysis have been used, the reason why they were used, and what is their relevance for the present study. I also strongly suggest to link the main take-aways of this analysis to the supplementary figures. Same for the qPCR on for functional genes. The claim that these analyses prompted the authors to focus on the relationship between cohesion and environmental factors is not well motivated.
- From main text it is unclear how the authors estimated the association between each ASVs and positive cohesion. The authors might add a couple of sentences in the paragraph starting at line 119 about this analysis.
- Line 134: is rrn copy number important for function because is it a proxy for maximum growth rate? If this is the rationale it should be explained in the results, not only in the discussion.
- Validation of microbial interactions with pure culture section:
 - o The description of these experiments is too vague and makes it hard to understand whether the results support any conclusion. In the main text it should be clear in which medium species were cultured.
 - o I am also confused about the way the growth yield is estimated: In Fig 4 panel a the y axis is the 16S gene operon copy number, but this is a proxy for maximum growth rate.
 - o Fig. 4 panel b: I might suggest to present interactions type data from experiments at the two temperatures in separate panels in Fig. 4.
 - o Fig. 4 panel d seems redundant with c
 - o I think an interesting control is to perform experiments with the Lactobacillus that dominates the communities at day 0.

There are several typos in the figures and figure legends that need to be fixed. In addition, I encourage the authors to define all the acronyms the first time they are used in the text (or in the Figures).

Reviewer #2 (Remarks to the Author):

The authors explore how self-generated temperature changes affect a composting community composition over time. By combining 16S sequencing, metagenome-assembled genomes (MAGs), meta-transcriptomics and in vitro interactions, they demonstrate that:

- Temperature is the main driver of community structure
- Heat stress, in accordance with the stress gradient hypothesis, favors facilitative behaviors instead of competition between community members
- Cobalamin (Vitamin B12) leakage by Actinobacteria towards Firmicutes contributes to cooperation at high temperature

We found the manuscript clear and well-structured. We find the use of the semi-natural poikilothermic system of compost well-suited for the in-situ study of community dynamics under temperature variation. We also appreciate that the authors try to go to the molecular details driving the ecological dynamics observed in their poikilothermic biosystem.

Major comments

- 1) Cobalamin cross-feeding is suggested as the mechanism of facilitation happening at high temperature. While the authors demonstrate i) the accumulation of cobalamin in the supernatant after in vitro growth of Actinobacteria, ii) the uptake of cobalamin by Firmicutes during in vitro growth and 3) that mutualistic interactions peak at 50°C in vitro, they don't really show that cobalamin cross-feeding is the driving mechanism. This brings two points:
 - a. The wording of the conclusions must be careful. The authors did well by writing that cobalamin "promotes" mutualism and "benefits" the community. However, we feel that it is important to note early on in the manuscript that it is not shown here that cobalamin is the main factor driving cooperation at high temperature (as they mention in the discussion: "It is undeniable that key species may secrete carbon degrading enzymes to promote the growth of facilitated strains."). We therefore suggest to tone down the conclusion and mention it at the end of the introduction to avoid misinterpretation of the results by the reader.
 - b. The authors justified their targeted analysis of cobalamin, although it was unclear whether doing this analysis differently might have highlighted a different metabolite. Is cobalamin usage "high"? Relative to what? The dynamics they show in vitro (secretion by Actinobacteria and uptake by Firmicutes) could in principle also occur for other metabolites. It might be worth testing one or two other metabolites as baselines, to check whether they show the patterns seen for cobalamin.
- 2) Many statistical tests are performed to disentangle ecological effects underlying the community dynamics observed. Please clarify the assumptions and expectations of each model in the text when they are mentioned as it is hard to follow for non-experts. It is unclear to us, for example, how you can prove that increased co-occurrence at high temperature is not just due to a few species tolerating those harsh environmental conditions. Please explain these points, as they are key to the conclusions of the paper.

Minor comments

- 1) Of course, it would have been optimal to isolate the Actinobacteria from the compost community rather than purchase them. However, we understand that if these species could not be isolated at the time of the experiment, purchasing similar strains is appropriate.
 - a. Can the authors comment on why the Actinobacteria strains could not be isolated in the first place while Firmicutes were?
 - b. Please provide comparative genomics information showing that the strains purchased carry the same metabolic capacities (especially for the cobalamin pathways) as the strains identified in the compost community with the MAGs.
 - c. Mention clearly in the main text that the strains used in vitro were different from the ones in the community.
- 2) Our understanding of the study is that it is mostly about the ecological dynamics of the community throughout the temperature changes of the composting process. Yet, when the idea of the Black Queen Hypothesis is mentioned, we are now talking about evolution of the community. As the heating-cooling cycle of the composting process is not cyclically experienced by the same community (in the sense that this process would typically happen only once), do you think there are enough selection forces to promote the evolution of cooperation among species growing at high temperatures? Please consider adding a few sentences about this in the discussion.
- 3) Throughout the manuscript, authors use the term "key" species, even once mentioning "keystone" species. This "key" term is a bit confusing. Why not use "Actinobacteria"?
- 4) Introduction:
 - a. "Bacteria can adapt to high temperature without collapsing". We suggest changing "adapt" for "acclimatize" as in this context it is about ecology, not evolution.
 - b. Please define "poikilothermic".

5) Results:

a. Section 2.1: Please explain why the rrn copy number is a good indicator (you can use the same explanation used in the discussion)

6) Discussion:

a. "This notion could be supported by theoretical predictions such as the Black Queen Hypothesis (BQH) which states that the cross-feeding of "expensive" public goods is evolutionarily selected for". This is not exactly the BQH, as it is the loss of costly traits that is selected for, not the cross-feeding. We suggest changing the text to: "This notion could be supported by theoretical predictions such as the Black Queen Hypothesis (BQH) which states that losing the production of "expensive" public goods is evolutionarily selected for and can result in obligate cross-feeding."

b. "slow-growing, but stress-tolerant, species preferred to cooperate with others though the sharing of public goods". We suggest changing the wording as "preferred" is anthropomorphic and poorly reflects that "sharing public goods" is generally a passive mechanism in the form of involuntary leakage of metabolites to the environment.

7) Fig. 1a:

a. The red line doesn't seem to add value to the plot and doesn't fit the data so well. We suggest removing it.

b. The thermophilic phase is described as starting when the temperature reaches 50°C. At the bottom of the plot, the initial phase is shown to last 5 days while the 50°C threshold is reached on day 2. Please adjust it or explain why it is this way.

8) Fig 3a: The legend is not detailed enough to understand the figure, please detail what dots and bars are representing.

9) Fig 3c: It is unclear what the 'total' category represents. Please add description in the legend.

10) Fig 3c-f: the legends are too small and hard to read, please consider increasing the font size.

11) Extended Fig. 1a: typo in the plot labels: 'Hemicellose/Cellose' should be 'Hemicellulose/Cellulose'?

12) Extended Fig. 1b-c: Please add in the legend what the arrows color stand for

13) Extended Fig. 1d-f: We understand the legend as "T1/T2/T3 groups" on the x-axis to be the "Initial/Thermophilic/Cooling groups" as stated in the box with the colors. Please consider changing the x-axis labels for simplicity.

14) Extended Fig. 2: Please be more specific in the legend: "based on correlation" between what and what?

15) Extended Fig. 3: Please be more specific in the legend: How did you obtain the red rectangles that differentiate the groups

16) Fig. S8b: What are all these indexes representing? Is the point to show that indexes values change over time? or that some are better than other to describe the system? Please explain in the legend.

17) Fig. S14: Please be more specific in the legend to describe the colors, square size and line size.

18) Fig. S17: Please be more specific in the legend: What does Eligible / Ineligible mean?

19) Please check the figures referencing in the text (especially in supplementary text), as there seems to be many errors.

Reviewer #3 (Remarks to the Author):

Zhao et al has done a thorough study of microbiomes in compost which has employed many complementary approaches including networks, coculturing, functional gene analysis, and chemical testing. I strongly applaud the authors for their efforts to provide a synthetic story with a proposed mechanism for beneficial interactions within the microbiome. This topic of understanding interactions within the microbiomes – especially outside of gut microbiomes – is at the cutting edge of the microbiome field right now and thus is very timely.

While there is much to like about this work, I do have serious reservations that I describe below.

1. Pseudoreplication and experimental design/analysis:

It is very hard to tell exactly what the authors have done in their statistical analyses based on what is in the manuscript and supplements, but I believe there are some substantial issues with the analyses based on what I can glean from the text. In general, I suggest the authors be more explicit about how the data are being analyzed in any revision regardless of whether it goes back to Nature Comm. I understand that it may need to be in the supplement rather than the main text, but I do not see anywhere that specifies the structure of statistical models.

Here are a couple of examples of issues with the design and statistics. What is an independent replicate in the sampling design? There are 180 samples sequenced coming from 10 piles that the authors measure at 6 time points with 3 technical reps/subsamples. There are only 10 piles within each time point and that would be the independent replicate for this experimental design.

-- If "time point" is the variable of interest, then there would only be 10 independent replicates per time (since subsamples are not independent replicates). In some places time point appears to be the variable of interest; for instance, in Figure S8 the authors show a trajectory of 6 networks. I don't know if the authors used their subsamples/technical reps as separate input into this network analysis to get up to 30 samples but that would be incorrect since it would inflate replication with non-independent samples (pseudoreplication). Alternatively, they may have used only 10 replicates (1 from each pile by averaging or randomly selecting 1 subsample) which would avoid pseudoreplication. However, that would also be an issue since this would be low sample number for network construction.

-- If the authors are grouping timepoints into three phases (as in figure 1b) to make "phase" the variable of interest, then pseudoreplication is likely an even bigger issue since community 1 at time point 2 is not independent of community 1 at time point 3 or 4, but it seems like data from community 1 at timepoints 2, 3, and 4 are being treated as independent replicates and subsamples within these timepoints are being treated as independent replicates. Both of these would be serious pseudoreplication issues.

2. Missing information/difficult to find information -- Another issue is there are a lot of places where not enough information is given to understand why the authors are doing something or what it means. I understand space is limited in the main text and short format papers are really hard to write, but I found I had a lot of questions about basic things reading the paper that impeded my understanding and it took a lot of work on my part to try to get answers. Even after looking at all materials in the supplement etc, I cannot tell what the authors did in many cases (e.g., how many samples went into some analyses, what was the structure of their statistic models, etc).

To help the authors improve their manuscript, I will highlight a few examples where the manuscript is unclear and not enough information is provided. In these cases, I spent a lot of time trying to piece information together to understand what is going on, but that level of time investment is not realistic for the average reader and did not answer my questions in some cases. This was a repeated problem throughout the manuscript and the paper could be improved by a careful evaluation of the entire text for areas where this happened. I think this is partially due to the breadth of analyses and approaches included but still needs to be resolved.

Table S1 – what is this a test of? The table title is "Tab. S1 Significance test for differences between 6 groups and 3 groups" but that doesn't tell me what the response variable is or why this analysis is being done. I looked in the main text for where these analyses are being called and it says "Thermophilic phase (days 5, 12, and 20) and Cooling phase (days 25 and 30) (Fig 1a), according to the composting phase (Table S1-S3 and Text S1 in SI file1)." However, this doesn't clarify what the response variable is either. When I look at Fig 1a, it is a graph of temperature. So maybe the response variable is temperature? However, that graph shows many box plots not grouping on 6 vs 3 which is what the table professes to be about. Also, some of the other supplementary material is hard to understand due to lack of information (e.g. Fig S5 doesn't define Hes, DL, Hos, etc in caption).

Line 97 – The first sentence of the paragraph says there are 10 compositing setups and the second sentence says there are 180 bacterial communities evaluated. I immediately wondered how you could get 180 samples from 10 replicates. This really isn't clear until line 367. This would not be a big deal to move up in the text, but it does highlight a bigger problem that most of the data is not independent of one another (see comments on pseudoreplication above).

Line 119 – Text reads “We selected four measures based on relative abundance, occurrence frequency, and significance to positive cohesion to reveal the key amplicon sequence variants (ASVs) influencing potential mutualism. Only those ASVs that fulfilled all four criteria could be defined as key ASVs (Fig. 1d and SI file2: Sheet 2).” This is really hard to follow. As written, I only see 3 not 4 measures listed (abundance, frequency, and positive cohesion) and what it takes to fulfill these criteria is not clear since nothing says whether high or low values are important. In looking later in the manuscript, I see that the authors give four criteria on line 468, but more information at least on the direction of effects needs to come sooner.

3. The way the authors talk about mutualisms is a bit odd in places (but could easily be corrected). For instance, in the abstract the authors write “Mutualism is supposed to be an ideal interactive paradigm of multi-species coexistence, but it is rarely observed. Abiotic stress is usually considered to promote mutualism, while there is still a paucity of research regarding how mutualism responds to it”.

First, mutualism is commonly observed in nature and there is a ton of research on how abiotic stress shapes mutualisms. The authors likely mean mutualisms in microbial communities, but as written the first two sentences of the abstract are factually incorrect.

Second, in this writing and in other places in the text, there is a connotation of mutualisms as idealized beneficial interactions where everything is altruistic (e.g., calling it an “ideal interactive paradigm of multi-species coexistence”). Realistically, it has been well established for decades that mutualisms are complex, context-dependent interactions that often shift from beneficial to commensal or parasitic depending on the environmental conditions and there is a lot of evidence of conflict among partners in mutualisms. I would just say the wording needs reworking.

Third, I think cross-feeding needs to be acknowledged in the intro because this is a well-established, common type of mutualistic interaction that occurs in natural microbiomes.

4. Keystone ASVs

Why are ASVs required to be in 90 out of 180 samples (line 440) to be included in the network? That is a very high threshold and may be excluding keystone ASVs which are not always dominant taxa. The four criteria around abundance and frequency for identifying key ASV (line 486) also excludes nondominant keystone ASVs. Outside of the microbial world keystone species are typically low biomass/abundance taxa that have a disproportionately large effect on their communities. Those taxa are being filtered out here. In microbiome work, researchers usually do not require taxa be rare to count as a keystone but have shown that rare and specialist taxa can be important keystones. Explicitly excluding those microbes via filtering and criteria maybe giving the false impression that there are only a few keystone ASVs.

Reviewer #1:

In the manuscript "Inter-bacterial mutualism promoted by public goods in a poikilothermic biosystem", Ning and coauthors described compositional turnover in compost communities as temperature varies over the course of 30 days. The authors measured positive cohesion, indicative of positively correlated abundances between the most abundant taxa in the community, and found that it increases with temperature, therefore peaking during the thermophilic phase of the composting process. The authors ascribe increases in positive cohesion to the prevalence of mutualistic interactions among two key Actinobacteria taxa supplying cobalamin and several Firmicutes taxa, possibly ameliorating growth conditions for the Actinobacteria.

I think the presented results are of interest for microbial ecologists and scientists working at the interface with applied microbiology. However, I would like the authors to clarify and expand on the following points.

Response: We are grateful for these comments. According to your comments and suggestions, we have carefully revised the manuscript. The response text including: a. Black italic type: the exact comment. b. Indent and in normal font: response to the comment. c. Indent and in blue: revisions in manuscript. Below we provide our point-by-point responses to your comments and hope the revised manuscript will meet with your approval.

1 While increasing temperatures can affect the outcome of species interactions and favor slower-growing taxa (see Lax, Abreu & Gore Nat Ecol Evol 2020 and Abreu, Dal Bello et al. bioRxiv 2022), the switch from 20 to 70C can actually filter out several species due to their thermal ranges. Do the authors know which % of the species in the initial community does not deterministically survive when temperature is above 50C? The claim reported in the discussion starting at line 278 confounds these two scenarios and needs to be backed by data.

Response: Thank the reviewer for this comment. It is meaningful to determine what percentage of species in the initial community do not deterministically survive when temperature is above 50°C. However, as the temperature tolerance should be identified at the species level, and over 90% (relative abundance) could not be classified to species level at day 0 (Fig R1), it was hard to determine the percentage of species that could not survive when temperature is above 50°C. Actually, we had intended to highlight that the increasing temperature would select for slow growing bacteria. We confused the environmental selection and environmental filtering in the original manuscript.

We revised the text to clarify it in manuscript “Due to strong temperature selection, the slow-growing and stress-tolerant species would be selected³⁴, and these species might provide direct benefits for other community members³⁵.” Please see lines 313-315 on page 15.

Fig. R1 The relative abundance of unclassified and classified taxa in samples collected from day 0.

To support the selection of slow-growing bacteria by high temperatures, we carefully referred to Lax, Abreu & Gore Nat Ecol Evol 2020 and Abreu, Dal Bello et al. Sci Adv 2023 and calculated the mean copy number (MCN) (Fig. S23b and c). Results showed that the minimum of MCN was observed at Day 05 (MCN = 3.8). Further correlation analysis showed a significantly negative correlation between temperature and MCN ($r=0.75$, $p < 0.001$). The results were consistent with those obtained by Abreu et al (2020) and Abreu et al (2023), where an increase in temperature would select for the slow-growing bacteria.

We added the following text in manuscript: “The ribosomal RNA operon (*rrn*) copy number was calculated by matching Ribosomal RNA Operon Copy Number Database (*rrndb*), which is a phylogenetically conserved trait and a proxy for maximum growth rate²⁶, to inferred maximum growth rates of neighboring ASVs and ACT ASVs (Text S5 in SI file1). The significantly higher copy number (6.9) of *rrn* in neighbouring ASVs (6.9) (Fig. S23 in SI file1), and the high explanation of changes in functional potential (51.0%) (Extended Data

Fig. 2) emphasized that the neighboring ASVs were fast-growing bacteria, and are highly associated with composting function. In contrast, the lower rrn copy number in ACT ASVs indicated their slow-growing characteristics. Due to the significantly positive correlations between ACT ASVs' abundance (with lower rrn copy number), and temperature (Fig. S20 in SI file1), it also implied that high temperature might favor slow-growing bacterial species. To confirm this, we calculated the mean copy number (MCN), which is the abundance-weighted rrn copy number for each sample²⁷, to summarize how temperature affected the distribution of fast and slow growers in the composting bacterial community. The minimum value of MCN was observed on day 5 (Fig. S23b in SI file1). Moreover, the correlation analysis (Fig. S23c in SI file1) and multiple linear regression (MLR) analysis (Fig. S24 and Table S6 in SI file1) supported that temperature was the main factor regulating MCN (Text S5 in SI file1). Therefore, high temperature favour slower-growing bacterial species, and these species (i.e., ACT ASVs) might cooperate with other species that grow rapidly and affect composting functions.” Please see lines 137-158 on pages 7-8.

We added text in Section E of Supplementary Information I: “**The ribosomal RNA operon copy number and mean copy number** Since the maximum growth rate of a bacterium is proportional to the number of ribosomal RNA operon (rrn) copies it has, we calculated the rRNA copy number of different composting bacteria by matching ASVs to the Ribosomal RNA Operon Copy Number Database (rrndb) to infer the maximum growth rate of different composting bacteria¹⁶. The rrn copy

number was matched the lowest available rank (e.g., if the species and genus levels are unclassified, then the family level would be the representative). Mean copy number (MCN) was calculated as the average rRNA copy number for each sample as described by Abreu et al. (2022). In detail, the MCN was calculated by weighting the rRNA copy number of each ASV and its relative abundance¹⁷. MCN for each sample is the average of three replicates.” Please see lines 425-435 on page 40 and lines 456-461 on page 43 in Supplementary Information I - Section E.

Fig. S23. Mean copy number (MCN) during the composting process. a) Ribosomal RNA operon copy number between neighbouring ASVs and ACT ASVs. In the boxplots of panels, hinges indicate the 25th, 50th, and 75th percentiles, whiskers indicate $1.5 \times$ interquartile ranges, and dots indicate values of individual samples. b) MCN for different sampling time points. c) The relationship between MCN and temperature.

Reference:

1. Lax, S., Abreu, C.I. & Gore, J. Higher temperatures generically favour slower-growing bacterial species in multispecies communities. *Nat. Ecol. Evol.* 4, 560–567 (2020).
2. Abreu, C. I., Dal Bello, M., Bunse, C., Pinhassi, J. & Gore, J. Warmer temperatures favor slower-growing bacteria in natural marine communities. *Sci Adv.* 9, e8352 (2023).

2 The compositional turnover observed in this compost communities is consistent with a succession, which is one of the scenarios in which SGH might operate. However, I believe that the SGH applies more to systems in which the range of variation in the 'focal' environmental stressor is within the range ensuring the survival of all species. This might true for the depletion of nutrients during the compositing process, but not for temperature, for the reasons highlighted in the previous point. This, however, does not detract from the finding that mutualistic interactions are prevalent in the thermophilic compost communities.

Response: We thank the reviewer for this comment. Indeed, SGH is more applicable to systems where all species can survive within a range of variations in the 'focal' environmental stress.

We revised the text and checked it throughout the manuscript “The results of statistical approaches (Section C in SI file1) and *in vitro* work (Fig. 4) emphasized that the increased temperature could also promote the cooperative behaviour. Although the stress gradient hypothesis (SGH) might be a probable explanation for the increase of cooperative behaviour under environmental stress. However, the SGH focuses on systems in which all species can survive within a range of variations in the 'focal' environmental stress and the rapid temperature rise to 70°C was undoubtedly a selecting pressure for bacteria. Thus, we suggested that the increase of cooperative behaviour may have been caused by the temperature selection on the bacterial community.” Please see lines 296-305 on pages 14-15.

3. *I am very intrigued by the fact that, during the thermophilic and cooling phases, slower-growers like Actinobacteria become are more abundant at higher temperatures, despite the general idea that slower-growing taxa usually benefit from a nutrient-poor environment (possibly like the one in the cooling phase). This is consistent with results from the two studies I reported above (Lax, Abreu & Gore Nat Ecol Evol 2020 and Abreu, Dal Bello et al. bioRxiv 2022). I think it would be interesting if the authors could plot the weighted mean copy number (see Abreu, Dal Bello et al. bioRxiv 2022, Wu et al ISME 2017, Nemergut et al. ISME 2016) of their communities in the thermophilic and cooling phases as a function of temperature and nutrient concentration.*

Response: Thank the reviewer for this suggestion. We carefully referred to the recommended manuscript. The mean copy number (MCN) is indeed an appropriate indexes to characterize how high temperature selects the growth rate of bacterial communities. We calculated the MCN as Abreu, Dal Bello et al (2023) described (response to question 1). Multiple linear regression was used to plot the MCN of the communities in the thermophilic and cooling phases as a function of temperature and nutrient concentration. Results showed that the combination of temperature, cellulose, lignin and total organic carbon (TOC) had the highest R^2 (0.56). Meanwhile, all of these factors had significant impacts on the changes in MCN, of which temperature was the most important factor, followed by cellulose. Our results were consistent with those of Lax, Abreu & Gore Nat Ecol Evol. 2020 and Abreu, Dal Bello et al. Sci. Adv. 2023, showing that temperature is a determined factor in regulating the bacterial

maximum growth rates and high temperature favors slower-growing bacteria in composting.

We added the following text in manuscript: “Moreover, the correlation analysis (Fig. S23c in SI file1) and multiple linear regression (MLR) analysis (Fig. S24 and Table S6 in SI file1) supported that temperature was the main factor regulating MCN (Text S5 in SI file1). Therefore, high temperature favour slower-growing bacterial species, and these species (i.e., ACT ASVs) might cooperate with other species that grow rapidly and affect composting functions.” and “The significantly negative correlation between MCN and temperature range in the tested composting piles suggested that higher temperature would select for slower growing bacteria. However, this negative correlation may have depended on nutrients. Therefore we constructed a multivariate regression model (Fig. S24 and Table S6 in SI file1) to test whether the inclusion of nutrients would influence the temperature coefficient, the results of which showed that temperature was the most important factor in explaining the variation in fast and slow growing bacteria. These results were consist with previous studies from soils³³ and oceans²⁷. Due to strong temperature selection, the slow-growing and stress-tolerant species would be selected³⁴, and these species might provide direct benefits for other community members³⁵.” Please see lines 153-158 on page 8 and lines 306-315 on page 15.

We added following text in Section E of the Supplementary Information I: “**Multiple linear regression model** We conducted 2 multiple linear regression (MLG) models to reveal how temperature and nutrients affected MCN: (1) with all 60

samples, (2) with 50 samples (excluding samples collected on Day 00) to confirm the results. In both MLG models, temperature and nutrients were independent variables, and MCN was dependent variable. We used R packages “*tidyverse*” and “*leaps*” to construct the MLR and obtain R^2 for each model (Fig. S25a). Given the equal importance of optimal model and minimal model in the construction of MLG models, we further calculated the Bayesian Information Criterion (BIC) for the different models with a lower BIC indicating a better R^2 .

Results of MLG model showed that although the R^2 for the combination of "temperature" and "TC" was not the largest among the 60 sample group, they had the smallest BIC and was the simplest model, therefore this MLR was the most appropriate for the 60 sample group. Also, the combination of "temperature", "TC", “Cellulose”, and “Lignin” was both the simplest and most optimal model for the 50 samples group (Fig. S24 and Table S6). Temperature was the most important factor in both models, therefore, temperature was the key factor influencing the MCN. Considering the significantly negative correlation between temperature and MCN (Fig. S23), high temperature might therefore favor slow-growing bacteria.” Please see lines 436-453 on pages 40-41, lines 462-464 on page 44 and lines 474-477 on page 46 in Supplementary Information I - Section E.

Reference:

3. Lax, S., Abreu, C.I. & Gore, J. Higher temperatures generically favour slower-growing bacterial species in multispecies communities. *Nat. Ecol. Evol.* 4, 560–567 (2020).
4. Abreu, C. I., Dal Bello, M., Bunse, C., Pinhassi, J. & Gore, J. Warmer

temperatures favor slower-growing bacteria in natural marine communities. *Sci Adv.* 9, e8352 (2023).

Fig. S24. Multiple linear regression between MCN and environmental factors. a) R² for different models. b) Bayesian Information Criterion (BIC) for different models. c) Relative importance of predictor variables. The red box indicates the selected model.

Table S6 The detail value of the constructed multiple linear regression between temperature, nutrients and positive cohesion.

Table S6.1 The model including 60 samples

60 Samples	Estimate	Std. Error	t value	Pr(> t)	Significant
Intercept	4.8403	0.21806	22.20	2.0e-16	Yes
Temperature	-0.0363	0.00304	-11.92	2.0e-16	Yes
TC	0.0221	0.00316	6.98	3.4e-09	Yes

$MCN = 4.8403 - 0.0363 * \text{Temperature} + 0.0221 * \text{TC} + \epsilon$

ϵ is a constant in the multiple regression model
 TC: Total Carbon

Table S6.2 The model including 50 samples (excluding samples collected from day 0)

50 Samples	Estimate	Std. Error	t value	Pr(> t)	Significant
Intercept	5.3048	0.23823	22.27	2.0e-16	Yes
Temperature	-0.0218	0.00499	-4.36	7.4e-05	Yes
Cellulose	-0.3294	0.09060	-3.64	0.00071	Yes
Lignin	0.2582	0.12234	2.11	0.04042	Yes
TC	0.0222	0.00825	2.70	0.00985	Yes

MCN=5.3048-0.0218*Temperature-0.3294*Cellulose+0.2582*Lignin+0.0222*TC+ε
 ε is a constant in the multiple regression model
 TC: Total Carbon

4. *How is community diversity changing during the composting process? Do the positive cohesion enhance diversity?*

Response: Thank the reviewer for this comment. We calculated the Shannon and Chao1 indices. The results showed no significant difference between all paired groups (Table S4), although the means showed a trend of decreasing and then recovery (Fig. S4). Further correlation analysis showed a weak negative correlation between positive cohesion and diversity ($p > 0.05$).

We added the following text, Fig S4 and Table S4 in Supplementary Information I: “To explore whether there were significant differences between the Shannon and Chao1 indices at different sampling time points, we tested the pairwise Adonis between different groups using the R packages “*vegan*” and “*pairwiseAdonis*”. The results showed no significant difference between all paired groups (Table S3), although the means showed a trend of decrease and then recovery (Fig. S4). Further correlation analysis showed a weak negative correlation between positive cohesion and diversity ($p > 0.05$).” Please see lines 59-65 on page 3, lines 77-82 on page 7 and lines 86-88 on pages 8 and 9 in Supplementary Information I - Section A.

Fig. S4. Variation in α -diversity between sampling times. a) Shannon index. In the boxplots of panels, hinges indicate the 25th, 50th, and 75th percentiles, whiskers indicate $1.5 \times$ interquartile ranges, and dots indicate values of individual samples. b) Chao1 index c) Relationship between Shannon index and positive cohesion. d) Relationship between Chao1 index and positive cohesion. Grey shading denotes the 95% confidence intervals.

Tab. S3.1 Significance test for differences for Shannon index

Group pairs	Sum.sq	R2	F	Pr(>F)	Significant
Day00 vs Day05	0.0162	0.1155	2.3499	0.1630	No
Day00 vs Day12	0.0372	0.1252	2.5755	0.1120	No
Day00 vs Day20	0.0035	0.1223	2.5093	0.1390	No
Day00 vs Day25	0.0003	0.0193	0.3549	0.5520	No
Day00 vs Day30	0.0001	0.0035	0.0640	0.8130	No
Day05 vs Day12	0.0052	0.0137	0.2497	0.7000	No
Day05 vs Day20	0.0051	0.0359	0.6705	0.4140	No
Day05 vs Day25	0.0204	0.1390	2.9061	0.1250	No
Day05 vs Day30	0.0142	0.0987	1.9708	0.1840	No
Day12 vs Day20	0.0199	0.0680	1.3126	0.3250	No
Day12 vs Day25	0.0431	0.1407	2.9468	0.0860	No
Day12 vs Day30	0.0344	0.1143	2.3232	0.1440	No
Day20 vs Day25	0.0058	0.1719	3.7375	0.0680	No

Day20 vs Day30	0.0026	0.078	1.5225	0.2020	No
Day25 vs Day30	0.0006	0.0289	0.5354	0.4810	No

Tab. S3.2 Significance test for differences for Chao1 index

Group pairs	Sum.sq	R2	F	Pr(>F)	Significant
Day00 vs Day05	0.1096	0.1425	2.9915	0.1170	No
Day00 vs Day12	0.1529	0.1529	3.2484	0.0720	No
Day00 vs Day20	0.0313	0.1211	2.4802	0.1310	No
Day00 vs Day25	0.0040	0.0241	0.4439	0.5540	No
Day00 vs Day30	0.0184	0.0577	1.1018	0.3300	No
Day05 vs Day12	0.0057	0.0039	0.0698	0.8650	No
Day05 vs Day20	0.0338	0.0387	0.7251	0.4370	No
Day05 vs Day25	0.1214	0.1357	2.8265	0.0840	No
Day05 vs Day30	0.0396	0.0416	0.7806	0.3570	No
Day12 vs Day20	0.0656	0.0600	1.1498	0.3200	No
Day12 vs Day25	0.1609	0.1434	3.0144	0.0870	No
Day12 vs Day30	0.0696	0.0595	1.1395	0.2740	No
Day20 vs Day25	0.0462	0.1195	2.4422	0.1120	No
Day20 vs Day30	0.0035	0.0073	0.1318	0.8050	No
Day25 vs Day30	0.0268	0.0609	1.1674	0.2920	No

While results appear interesting, the manuscript quite difficult to read because crucial details about why and how a certain analysis is performed are missing. I believe the reader should be able to the get a sense about these details without going to the Methods section. As a general suggestion, starting each results section explaining why an analysis is performed, briefly describing how, and then presenting the results would improve a lot the readability of this manuscript.

Response: We thank the reviewer for this comment. Per your suggestion, we revised our manuscript to highlight why and how we carried out this analysis. We restructured the manuscript and Supplementary Information I, to make it clearer. Please see **Supplementary Information I**.

Detailed comments:

Introduction:

- *Lines 48-55: It might be useful to talk about positive interactions in more general terms to include also commensalism, which has been found to be quite common among soil bacteria (Kehe et al. Sci Adv. 2021)*

Response: We thank the reviewer for this suggestion. Per your suggestion, we present positive interactions in more general terms to include commensalism.

We revised the sentence to “Although positive interactions (primarily as commensalism) are commonly observed between cultivable strains isolated from soil⁵, mutualism is still rare as it would change to commensalism or parasitism depending on the environmental conditions.” Please see lines 44-47 on page 3.

- *Line 69 “proper” might not the right word to convey the need for a system that is closer to natural conditions*

Response: We thank the reviewer for this comment. We revised “proper” to “a quasi-natural habitat under abiotic stress”.

We revised the sentence to “Thus, it is essential to explore how microbial interactions change in a quasi-natural habitat under abiotic stress.” Please see lines 66-67 on page 4.

- *Lines 70-88: In describing compost as a model system, the authors never mention the term that is used in the title and abstract (poikilothermic): I suggest to not use this*

term, because usually it is referred to animals that cannot control their internal temperature. I'd rather say that that compost is system characterized by deterministic temperature variation or something along these lines.

Response: We thank the reviewer for this comment. We have checked and revised the “poikilothermic” throughout the manuscript.

We changed the title to “Inter-bacterial mutualism promoted by public goods in a biosystem characterized by deterministic temperature variation”. We revised the sentence to “Here, we monitored a quasi-natural community (composting) in a biosystem, where temperature (20°C-70°C) was the main abiotic stress.” Please see lines 1-2 on page 1 and lines 25-26 on page 2.

Results:

- I find intriguing that different communities sampled in the same day during the thermophilic phase can display a wide range of temperatures (between 30 to 70C). It would be interesting to look at temperature trajectories of individual communities to assess the variability in the time it takes to switch between phases, and whether interactions or community composition can explain such variability.

Response: We thank the reviewer for this suggestion. Per your suggestion, we focused the temperature trajectories of individual communities and tested the relationship between positive cohesion and temperature in each pile. Considering the risk of pseudo-replication that might be raised by phase division (suggested by Reviewer 3), we based the analysis on sampling points and removed all descriptions

of phases. 50°C and 40°C were used as a divider (Fig. S16a). The variation in temperature indicated that it needed 2.5 ± 0.8 days for the pile to warm up from the initial rise to over 50 °C and 7.4 ± 3.1 days to reduce from 50°C to below 40°C. Remarkably, an unexpected secondary warming occurred in Pile C. Although the exclusion of data collected from pile C would allow for better results (e.g., higher r value in Fig. R2 than Fig 1c), we still retained it to respect the unexpected phenomena and exceptional samples, as Pile C completed the composting process. To confirm the relationship between positive cohesion and temperature in individual communities, a linear model (positive cohesion as the dependent variable, temperature as the independent variable) was performed separately for each pile. Results showed that positive cohesion could explain the change in temperature in nearly all piles ($R^2 = 0.55-0.92$, $p < 0.05$), except for Pile C. Although no significant relationship was observed between positive cohesion and temperature in Pile C, it also achieved an R^2 of 0.3. Therefore, these results showed that the temperature variations in the different piles may be mainly caused by the variations in the positive cohesion.

We added the following text: “**Linear regression analysis in Fig S16** To reveal the relationship between positive cohesion and temperature in each pile, we constructed another linear regression analysis (Positive cohesion = $a + b \times \text{Temperature}$). Each linear model included samples only collected from this pile (6 sampling time points). Other details were similar as previously described in Text S3.1. To further confirm the relationship between positive cohesion and temperature in each pile, we focused the temperature trajectories of individual communities and tested

their relationship by linear regression analysis in each pile. The structure of linear regression analysis was the same as described in Test S3.1, where only the data for each pile were used (6 sample for each analysis). The variation in temperature indicated that it needed 2.5 ± 0.8 days for the pile to warm up from the initial rise to over $50\text{ }^{\circ}\text{C}$ and 7.4 ± 3.1 days to reduce from $50\text{ }^{\circ}\text{C}$ to below $40\text{ }^{\circ}\text{C}$ (Fig. S16a). Results of linear regression analysis showed that positive cohesion could explain the change in temperature in nearly all the piles ($R^2 = 0.55\text{-}0.92$, $p < 0.05$), except for Pile C (Fig. S16b). Although no significant relationship was observed between positive cohesion and temperature in Pile C, it also achieved an R^2 of 0.3. Therefore, these results showed that the temperature variations in the different piles may be mainly caused by the variations in the positive cohesion. Remarkably, an unexpected secondary warming occurred in Pile C. There is no doubt that the exclusion of data collected from pile C allows for better results. however, as Pile C completed the composting process, we did not exclude these data in order to respect the unexpected phenomena and exceptional samples.” Please see lines 309-330 on pages 25 and 26, and lines 346-349 on page 30 in Supplementary Information I -Section C.

Fig. S16. Temperature in each pile. a) The temperature variations within different piles. The black line indicates the time required for the temperature transition. b) Linear model between positive cohesion (dependent variable) and temperature (independent variable).

Fig. R2 Relationship between positive cohesion and temperature. The crosses indicate samples collected from Pile C, which was removed from this analysis. The word next to the cross was the name of sample. The first “C” indicated the sampled pile and the other indicated the sampling time points.

- Given the variability of the temperature of the different communities within the different phase, especially the thermophilic one, I find panel B of Fig. 1 quite misleading.

Response: We thank the reviewer for this comment. We replaced the original Figure 1b with a new one, whose x-axis is sampling time points.

We revised the sentence in manuscript to “As with temperature, positive cohesion also peaked on day 5 (0.28), which was 1.1-1.6 times higher than other periods (Fig. 1b).” Please see lines 110-111 on page 6 and lines 828-839 on pages 35 and 36.

Fig.1 Changes of temperature and positive cohesion. **a)** Temperature variations in the 10 composting piles. 50°C is generally considered to be the dividing line for

composting in terms of distinguishing the high temperature phase from other phases in this system. **b)** Changes in the positive cohesion of different phases. In the boxplots of panels, hinges indicate the 25th, 50th, and 75th percentiles, whiskers indicate $1.5 \times$ interquartile ranges, and dots indicate values of individual samples. **c)** Relationship between positive cohesion and temperature. **d)** Screening for the ACT ASVs (detail of 4 criteria are described in Methods section). **e)** Whole composting network constructed by the random matrix theory (RMT) model. The red line indicates a positive correlation and the blue lines a negative correlation. **f)** The sub-network containing ACT ASVs and their neighbors. Different colors of the outer ring are used to distinguish taxonomic affiliation.

- It would be also curious to see variations in positive (and negative) cohesion in the 10 composting piles (basically the same plot of Fig. 1a but with cohesion on the y axis).

Response: We thank the reviewer for this suggestion. Per your suggestion, we plotted new Fig. 1b (y-axis: positive cohesion, x-axis: sampled day) and Fig. S5 (y-axis: negative cohesion, x-axis: sampled day).

We revised the sentence to “Meanwhile, we calculated the positive and negative cohesion to characterize the potential bacterial interaction, which is the sum of the significant positive (negative) correlations between taxa weighted by the relative abundance of taxa in each sample (Fig. 1b and Fig. S5)²⁰.” Please see lines 103-106 on page 6 in manuscript, and line 215-217 on page 16 in Supplementary Information I - Section B.

Fig. S5. Calculation of negative cohesion. In the boxplots of panels, hinges indicate the 25th, 50th, and 75th percentiles, whiskers indicate $1.5 \times$ interquartile ranges, and dots indicate values of individual samples.

- Line 105-110: This paragraph is quite vague and assumes all reader are familiar with iCAMP analyses. I strongly recommend that the authors briefly explain in the results section what tools of the iCAMP analysis have been used, the reason why they were used, and what is their relevance for the present study. I also strongly suggest to link the main take-aways of this analysis to the supplementary figures. Same for the qPCR on for functional genes.

Response: We thank the reviewer for this comment. We added details for iCAMP and qPCR for functional genes in Supplementary information I-Section B.

“Bacterial communities are influenced by both deterministic and stochastic processes. Deterministic processes include environmental selection and various biological interactions. Therefore, before exploring the effects of microbial interactions and environmental filtering on the composting community, it is necessary to reveal the bacterial community composition and determine whether the composting community is affected by deterministic factors³. To quantify various ecological

processes (particularly mean community assembly processes), The phylogenetic-bin-based analysis (iCAMP) and QPEN analyses were calculated, respectively³. In detail, the obtained ASVs were firstly classified into various bins based on phylogenetic relationships. The author-recommended phylogenetic distance cutoff ($d_s = 0.2$) was used. The bin size limitation (n_{min}) was checked and set to 24. The beta Net Relatedness Index (β NRI) were calculated to determine the main process for each bin. For each bin, the pairwise comparisons with $|\beta$ NRI| > 1.96 were regarded as a percentage of deterministic process, with β NRI < -1.96 governed by homogeneous selection and β NRI > 1.96 by heterogeneous selection. Afterward, the taxonomic diversity (RC_{Bray}) was further calculated to separate the stochastic process ($|\beta$ NRI| \leq 1.96). Fractions with $RC_{bray} > 0.95$ was considered homogenizing dispersal, while those with $RC_{bray} < -0.95$ were considered dispersal limitation. The remaining portion ($|\beta$ NRI| \leq 1.96, $|RC_{bray}| \leq 0.95$) were designated as "drift". Finally, the proportions of the various processes in all bins were abundance weighted and summarized. Meanwhile, the beta mean nearest taxon distance (β MNTD) and beta nearest taxon index (β NTI) were further calculated (QPEN analysis), which are deemed to be more valid for the community-wide-based analysis⁴. All analyses were performed with the iCAMP package in R (v.4.2.0)^{4,5}. Based on iCAMP analysis, dispersal limitation (DL) and homogeneous selection (HoS) were more important than other processes in bacterial community assembly with average relative importance of 34.9% and 64.3% (Fig. S6a), respectively. Thus, according to the iCAMP result, the composting process was regulated by deterministic factors. The results obtained by

QPEN were consistent with iCAMP (Fig. S6b) in that composting process was regulated by deterministic factors. The numerical difference between iCAMP and QPEN might be caused by the fact that iCAMP was the mean after bins and smaller taxa are susceptible to stochastic processes. Overall, these results showed that the composting bacterial community is determined by deterministic processes. Thus, it is valuable to further analyse the effects of biotic and abiotic factors on composting bacterial communities.” Please see lines 91-125 on pages 10 and 11 in Supplementary Information I - Section B

“We further tested the correlations between positive (negative) interaction (characterized by cohesion index) and community assembly and functional potential to reveal their importance for the composting bacterial community. Community assembly was characterized by the proportions of Hos and DL. Functional potential was characterized by the abundance for 54 functional genes (including carbon degradation, carbon fixation and nitrogen cycling) (see Methods for details). Results indicated that positive cohesion was highly associated with both the HoS and DL processes rather than negative cohesion, which showed the importance of positive cohesion to bacterial assembly (Fig. S11). Meanwhile, positive cohesion was also highly associated with the abundance of functional genes (Extended Data Fig. 1). However, no significant differences could be observed between negative cohesion and the abundance of functional genes (Fig. S12). Together, these results indicated that potential positive interactions (characterized by positive cohesion) was one of the main drivers of the bacterial community. Therefore, focusing on positive cohesion

was necessary.” Please see lines 187-201 on pages 13 and 14 in Supplementary Information I - Section B.

The claim that these analyses prompted the authors to focus on the relationship between cohesion and environmental factors is not well motivated.

Response: We thank the reviewer for this comment. Actually, we would like to explain why we focused on positive cohesion rather than negative cohesion here. We rewrote the text to highlight the importance of positive cohesion to both community assembly and function in manuscript.

“Given its importance for both community assembly (calculated by iCAMP analysis) and the functional potential (abundance of 52 functional genes quantified by quantitative PCR (qPCR)), the focus was on positive cohesion (i.e. potential interactions) rather than negative cohesion (Section B in SI file1).” Please see lines 106-110 on page 6.

- From main text it is unclear how the authors estimated the association between each ASVs and positive cohesion. The authors might add a couple of sentences in the paragraph starting at line 119 about this analysis.

Response: We thank the reviewer for this suggestion. We constricted a random forest model to estimate the association between each ASVs and positive cohesion. In this random forest model, the relative abundance of each ASV served as predictor for positive cohesion. The importance was assessed by increase of mean square error and

significance was assessed by the significance of each ASV on positive cohesion.

We added text in Results “Four kinds of ASVs were defined, including overall abundant ASVs (average relative abundance > 0.1%)²¹, ubiquitous ASVs (detected in over 80% of samples)²², frequently abundant ASVs (the relative abundance being top 80% in more than 50% of samples)²³, and PosCoh related ASVs (ASVs with significant impact on the changes of positive cohesion). In detail, the impact was assessed by the percentage of increase of mean square error (increase of MSE), and significance was assessed by the significance of each ASV on positive cohesion (done through a random forest model²⁴) (Fig. S17 in SI file1).” Please see lines 116-123 on page 6.

We added text in Materials and methods “PosCoh related ASVs were identified by random forest model. In this random forest model, the relative abundance of each ASV served as predictor for positive cohesion. To estimate the importance of these indexes, we used percentage increases in the MSE of variables: higher MSE% values imply variables with greater importance²⁴. The significance of each predictor on the response variables was assessed with the “*rfPermute*” R package. 500 decision trees were set, followed by 1000 random permutations and 5 repetitions of 10-fold cross-validation.” Please see lines 523-530 on page 25.

- *Line 134: is rrn copy number important for function because is it a proxy for maximum growth rate? If this is the rationale it should be explained in the results, not only in the discussion.*

Response: We thank the reviewer for this suggestion. Indeed, rrn copy number is important because it is a proxy for maximum growth rate. We added text in the Results. Please refer to Question 1 and 3.

- *Validation of microbial interactions with pure culture section:*

o The description of these experiments is too vague and makes it hard to understand whether the results support any conclusion. In the main text it should be clear in which medium species were cultured.

Response: We thank the reviewer for this suggestion. We added more details about the used medium in the manuscript and added tables in Supplementary information I to clarify the experiment design.

“Firstly, we tested the effect of the supernatants and cobalamin on Firmicutes in LB-Lennox medium (peptones 10 g, yeast extract, 5 g, NaCl 5g per litre). The supernatants were collected from all strains in Key and General groups, after 72 hours of culturing in LB medium with 1ppm cobalt (by adding 2.188 mg / L CoCl₂), which is key to the production of cobalamin⁴⁹.” Please see lines 551-555 on page 26 in manuscript.

We also added the experiment design as tables. Please see lines 618-621 on pages 60 and 61 in Supplementary Information I - Section G.

Table S9 Experiment Design

Table S9.1 The effect of supernatants and cobalamin on the growth of Firmicutes

Medium	Temperature	Groups	Detail
LB	37 °C	Key	10% (v/v) supernatants of Thermobifida fusca *

medium	and 50 °C	group	10% (v/v) supernatants of Saccharomonospora viridis 10% (v/v) supernatants of Sphingomonas paucimobilis
		General group	10% (v/v) supernatants of Lactobacillus brevis 10% (v/v) supernatants of Idiomarina andamanensis
		Cobala min Group	10% (v/v) cobalamin (1.5 ppm)
		Control	additional 10% (v/v) LB medium (with 1 ppm Co)

*** The supernatants were collected from targeted strains after culturing for 72 hours in LB medium with 1 ppm Co (2.188 mg / L CoCl₂).**

Table S9.2 Co-culture between Firmicutes and other species

Medium	Temperature	Groups	Co-culture
LB medium and CL medium *	37 °C and 50 °C	Key group	40 Firmicutes strains + Thermobifida fusca 40 Firmicutes strains + Saccharomonospora viridis
		General group	40 Firmicutes strains + Sphingomonas paucimobilis 40 Firmicutes strains + Lactobacillus brevis 40 Firmicutes strains + Idiomarina andamanensis
		Control	40 Firmicutes strains

*** CL medium was the composting leachate collected from the compost.**

Table S9.3 Absorption (Firmicutes) and synthesis (Key group) of cobalamin

Medium	Temperature	Groups	Co-culture
LB medium + 1.0 ppm Co	37 °C	Key group	Thermobifida fusca Saccharomonospora viridis Sphingomonas paucimobilis
LB medium + 1.0 ppm Co		General group	Lactobacillus brevis Idiomarina andamanensis
LB medium + 1.5 ppm cobalamin		Firmicutes	10 Firmicutes strains

*** 1 ppm Co was added by 2.188 mg / L CoCl₂.**

o I am also confused about the way the growth yield is estimated: In Fig 4 panel a the y axis is the 16S gene operon copy number, but this is a proxy for maximum growth rate.

Response: We thank the reviewer for this comment. Indeed, 16S rRNA gene operon copy number **per cell** is a proxy for maximum growth rate. We quantified the total 16S rRNA gene copy number for each sample by quantitative PCR, which is a widely used method to reflect the abundance of targeted bacteria, for this part (Schirmer et al., 2016; Zheng et al., 2020). We revised the 16S rRNA gene copy number which was obtained by qPCR analysis to “16S rRNA gene abundance” to distinguish them.

We revised the text to “All samples were uniformly collected at 72h and measured the 16S rRNA gene abundance by qPCR to characterize the overall growth yield for all groups. The details of the specific primers and the reaction system are described in Table S10.” Please see lines 565-568 on page 27 in manuscript.

Reference

1. Schirmer, M., Smeekens, S. P., Vlamakis, H., Jaeger, M., Oosting, M. & Franzosa, E. A. et al. Linking the human gut microbiome to inflammatory cytokine production capacity. *Cell*. 167, 1125 (2016).
2. Zheng, Y., Wang, J., Zhou, S., Zhang, Y., Liu, J. & Xue, C. et al. Bacteria are important dimethylsulfoniopropionate producers in marine aphotic and high-pressure environments. *Nat Commun*. 11, (2020).

o Fig. 4 panel b: I might suggest to present interactions type data from experiments at the two temperatures in separate panels in Fig. 4.

o Fig. 4 panel d seems redundant with c

*o I think an interesting control is to perform experiments with the *Lactobacillus* that dominates the communities at day 0.*

Response: We thank the reviewer for this suggestion. Per your suggestion, we performed experiments with the *Lactobacillus brevis* as another general species (control). Briefly, the results obtained by *Lactobacillus brevis* could better support our conclusions. In detail, results showed that the addition of the supernatants collected from *Lactobacillus brevis* could inhibit the growth of the Firmicutes on average, but not significantly (Fig. R3a). While, the co-culture of *Lactobacillus brevis* and Firmicutes show that competition was the dominant interaction type in these bacterial pairs, accounting for 89.0%. The remaining 11.0% was parasitism (Fig. R3b). No mutualism was observed in these bacterial pairs, which might be caused by the ecological suicide described by Ratzke et al (2018). As for other species in general group, we could detect minimal signals of cobalamin (less than 10 ppb) in the *Lactobacillus brevis* group and we replaced the original Extended Data Fig. 4 with a new one, which contained the data collected from *Lactobacillus brevis*. Furthermore, we replaced the original Fig. 4c and d with new figures, which display the interaction type data under 37°C (new Fig. 4c) and 50°C (new Fig. 4d). The results of *Lactobacillus brevis* are included in the new Fig. 4.

We revised the related text in Results as “Results showed that the average strength peaked at 50°C, which was 1.2 times higher than at 37°C (Fig. 4b). The average strength in the key group was also 1.4 times higher than the general group. Mutualistic relationships in the bacterial pairs in the key groups were 33.8% and 44.3% at 37°C and 50°C, respectively, which was 4.7 - 5.5 times higher than that in the general group (Fig. 4c and d). Meanwhile, the key group had significantly lower competitive relationship at 37°C (24.9%) and 50°C (3.2%) than the general group (60.6%, 54.7%) , decreasing by 35.7% and 51.6% respectively.”; “Mutualism occurred more commonly between Key group (*Thermobifida fusca* and *Saccharomonospora viridis*) and Firmicutes (33.8% - 44.3%), which was 4.7 - 5.5 times higher than that of General group (3.2% -35.7%).” Please see lines 272-279 on pages 13 and 14 and lines 318-321 on page 15 in manuscript.

We revised the related text in Materials and Methods as “Moreover, we used De Man Rogosa Sharpe (MRS) medium for the isolation of *Lactobacillus brevis* and 2216E medium for the isolation of *Idiomarina andamanensis* at 25°C and 30°C, respectively (Text S7.3 in SI file1).”; “We divided the 45 strains into 3 groups, including the Key group (*Thermobifida fusca* and *Saccharomonospora viridis*), the General group (*Sphingomonas paucimobilis*, *Lactobacillus brevis*, and *Idiomarina andamanensis*), and the Firmicutes group (40 Firmicutes strains).”; “We setup four groups in this part, including (1) the Blank group: only LB medium (with additional 10% of LB medium (v/v)), (2) the General group: LB medium + supernatants collected from *Sphingomonas paucimobilis*, *Lactobacillus brevis*, and *Idiomarina*

andamanensis (General group), respectively (10% (v/v)), (3) the Key group: LB medium + supernatants collected from *Thermobifida fusca* and *Saccharomonospora viridis* (Key group), respectively (10% (v/v)), (4) the Cobalamin group: LB medium + cobalamin (10% (v/v), 1.5ppm).” and “5 co-culture systems were set up under both 37 °C and 50°C, including (1) 40 Firmicutes strains + *Thermobifida fusca*, (2) 40 Firmicutes strains + *Saccharomonospora viridis*, (3) 40 Firmicutes strains + *Sphingomonas paucimobili*, and (4) 40 Firmicutes strains + *Lactobacillus brevis*, (5) 40 Firmicutes strains + *Idiomarina andamanensis*. Pure cultures of all strains were used as controls. Although the carbon sources had less effect on mutualism⁵, we tested both the LB medium and the composting leachate (CL, collected from composting) to confirm our results. Overall, we set up 5 (strains) * 2 (temperature) * 2 (medium) = 20 groups for this experiment (Table S9.2 in SI file1).”; “Specifically, as *Lactobacillus brevis* is classified as a Firmicutes, the abundance of Firmicutes in bacterial pairs related to *Lactobacillus brevis* were calculated by the difference between Firmicutes and *Lactobacillus brevis*.” Please see lines 539-541 on page 26, lines 547-550 on page 26, lines 557-564 on page 27, lines 569-577 on page 27, and lines 579-581 on page 28.

Fig. R3 Experiments with *Lactobacillus brevis*. a) The growth yield of 40 Firmicutes strains after the addition of supernatants collected from *Lactobacillus brevis* (General species). b) Interactions types between different bacterial pairs. Grey color indicates previous bacterial pairs. Red color indicates competition relationship observed in *Lactobacillus brevis* bacterial pairs. Purple color indicates parasitism relationship observed in *Lactobacillus brevis* bacterial pairs.

Extended Data Fig. 4. Consumption and accumulation of cobalamin. a) The consumption of cobalamin in Firmicutes. b) The accumulation of cobalamin in the Key group species and the General group. Grey shaded areas are 95% confidence intervals.

Fig. 4 Microbial interactions between pure culture strains. **a)** The growth yield of 40 Firmicutes strains after the addition of supernatants and cobalamin. The blank group received only LB medium. The General group contained supernatants collected from *Sphingomonas paucimobilis*, *Lactobacillus brevis*, and *Idiomarina andamanensis* (10% (v/v)) after 72h culture, respectively. The key group contained supernatants collected from *Thermobifida fusca* and *Saccharomonospora viridis* (10% (v/v)), respectively. The cobalamin group received additional LB medium (10% (v/v)) which contained 1.5 ppm cobalamin. In the boxplots of panels, hinges indicate the 25th, 50th, and 75th percentiles, whiskers indicate $1.5 \times$ interquartile ranges, and dots indicate values of individual samples. **b)** The average strength of interactions for different temperatures and groups (***) $p < 0.001$. **c)** The detail of interaction types under 37°C. The stacked bar indicates the interaction types between the Key group and the General group. Bar colors indicate the type of interaction. **d)** The detail of interaction types under 37°C. Bar colors indicate the type of interaction. The legends are the same as for Fig. 4c.

Reference

1. Ratzke, C., Denk, J. & Gore, J. Ecological suicide in microbes. *Nat Ecol Evol* 2, 867–872 (2018).

There are several typos in the figures and figure legends that need to be fixed. In addition, I encourage the authors to define all the acronyms the first time they are used in the text (or in the Figures).

Response: We thank the reviewer for this suggestion. Per your suggestion, we have revised the mistakes and check throughout the manuscript. We also defined all acronyms the first time they are used. We added lists of abbreviations in both manuscript and supplementary Information I.

“Abbreviations used in Supplementary Information I beta Net Relatedness Index: β NRI, beta mean nearest taxon distance: β MNTD, beta nearest taxon index: β NTI, homogeneous selection: HoS, heterogeneous selection: HeS, homogenizing dispersal: HD, dispersal limitation: DL, drift: DR, decision curve analysis: DCA, canonical correlation analysis: CCA, redundancy analysis: RDA, link test for environmental filtering: LTEF, moisture content: MC, electrical conductivity: EC, germination index: GI, mean squared error: MSE, within-module connectivity: Z_i value, among-module connectivity: P_i value, ribosomal RNA operons: rrn, Ribosomal RNA Operon Copy Number Database: rrndb, mean copy number: MCN, multiple linear regression: MLG, bayesian information criterion: BIC, Sequence Read Archive: SRA, Total carbon: TC.” Please see lines 626-636 on pages 63 in Supplementary information I.

Reviewer #2 (Remarks to the Author):

The authors explore how self-generated temperature changes affect a composting community composition over time. By combining 16S sequencing, metagenome-assembled genomes (MAGs), meta-transcriptomics and in vitro interactions, they demonstrate that:

- *Temperature is the main driver of community structure*
- *Heat stress, in accordance with the stress gradient hypothesis, favors facilitative behaviors instead of competition between community members*
- *Cobalamin (Vitamin B12) leakage by Actinobacteria towards Firmicutes contributes to cooperation at high temperature*

We found the manuscript clear and well-structured. We find the use of the semi-natural poikilothermic system of compost well-suited for the in-situ study of community dynamics under temperature variation. We also appreciate that the authors try to go to the molecular details driving the ecological dynamics observed in their poikilothermic biosystem.

Response: We are grateful for these comments. According to your comments and suggestions, we have carefully revised the manuscript. The response text including: a. Black italic type: the exact comment. b. Indent and in normal font: response to the comment. c. Indent and in blue: revisions in manuscript. Below we provide our point-by-point responses to your comments and hope the revised manuscript will meet with your approval.

Major comments

1) Cobalamin cross-feeding is suggested as the mechanism of facilitation happening at high temperature. While the authors demonstrate i) the accumulation of cobalamin in the supernatant after in vitro growth of Actinobacteria, ii) the uptake of cobalamin by Firmicutes during in vitro growth and 3) that mutualistic interactions peak at 50°C in vitro, they don't really show that cobalamin cross-feeding is the driving mechanism.

This brings two points:

a. The wording of the conclusions must be careful. The authors did well by writing that cobalamin “promotes” mutualism and “benefits” the community. However, we feel that it is important to note early on in the manuscript that it is not shown here that cobalamin is the main factor driving cooperation at high temperature (as they mention in the discussion: “It is undeniable that key species may secrete carbon degrading enzymes to promote the growth of facilitated strains.”). We therefore suggest to tone down the conclusion and mention it at the end of the introduction to avoid misinterpretation of the results by the reader.

Response: We thank the reviewer for this suggestion. Per your suggestion, we tone down the conclusion and mention it at the end of the introduction.

“Overall, our results indicated that (i) high temperature stress could enhance mutualism; (ii) slow-growing, but stress-tolerant, species cooperate with other bacteria by leaking metabolites.” Please see lines 92-94 on page 5.

b. The authors justified their targeted analysis of cobalamin, although it was unclear whether doing this analysis differently might have highlighted a different metabolite.

Is cobalamin usage “high”? Relative to what?

Response: We thank the reviewer for this comment. Cobalamin is used in a wide range of concentrations, ranging from 0.0005 to 20 ppm when culturing bacteria (Cooper et al., 2019; Higuchi-Takeuchi et al., 2019; Pira et al., 2022). We would like to confirm whether cobalamin was the main metabolites promoting the growth of Firmicutes. Therefore, the content of cobalamin needs to meet the requirements of Firmicutes growth. We found cobalamin residues only at concentration greater than or equal to 1.5 ppm (Extended Data Fig. 4), suggesting that Firmicutes growth may be limited by cobalamin deficiency at concentration below 1.5 ppm. Thus, we chose 1.5 ppm for tests.

Reference

1. Pira, H., Risdian, C., Muesken, M., Schupp, P. J. & Wink, J. *Pacificimonas pallium* sp. Nov., An Isolated Bacterium from the Mantle of Pacific Oyster *Crassostrea gigas* in Germany, and Prediction of One-Carbon Metabolism. *Diversity*. 14, (2022).
2. Cooper, M. B., Kazamia, E., Helliwell, K. E., Kudahl, U. J., Sayer, A. & Wheeler, G. L. et al. Cross-exchange of B-vitamins underpins a mutualistic interaction between *Ostreococcus tauri* and *Dinoroseobacter shibae*. *Isme J.* 13, 334-345 (2019).

3. Higuchi-Takeuchi, M. & Numata, K. Acetate-Inducing metabolic states enhance polyhydroxyalkanoate production in marine purple non-sulfur bacteria under aerobic conditions. *Frontiers In Bioengineering and Biotechnology*. 7, (2019).

The dynamics they show in vitro (secretion by Actinobacteria and uptake by Firmicutes) could in principle also occur for other metabolites. It might be worth testing one or two other metabolites as baselines, to check whether they show the patterns seen for cobalamin.

Response: We thank the reviewer for this suggestion. Per your suggestion, we tested riboflavin as a baseline as it is a biosynthetic vitamin like cobalamin (Fig. S28). The consumption of riboflavin in Firmicutes showed a trend of decreasing and then increasing, which was a different pattern from that of cobalamin. This result indicated that Firmicutes could secrete riboflavin extracellularly. Different trends could also be observed in the accumulation of riboflavin and cobalamin in the general and key group. The accumulation of riboflavin could be observed in all tested species, rather than only being secreted by a key group. Therefore, cobalamin showed a unique pattern.

We added the following text in manuscript: “To confirm whether other metabolites showed the patterns observed for cobalamin, we utilized riboflavin as a control, as it is a biosynthetic vitamin like cobalamin (Fig. S28). The results presented the exact opposite of those for cobalamin, that riboflavin could be secreted by all species tested (Fig. S31 in SI file1).” Please see lines 261-265 on page 13.

We added the following text and figure in section Supplementary Information: “To confirm whether other metabolites showed the patterns observed for cobalamin, we utilized riboflavin as a control, as it is a biosynthetic vitamin like cobalamin (Fig. S28). **Experimental setup and parameters setting** We cultured the same 10 Firmicutes strains in LB medium containing 1.5 ppm riboflavin. Meanwhile, we cultured Key and General group species in LB medium to reveal the accumulation of riboflavin. Samples were taken at 0, 12, 24, 48, 72h and centrifuged to obtain supernatants. Before detection, the supernatants were filtrated through a polyvinylidene fluoride membrane (0.22 μm) and diluted 50x to avoid signal suppression due to the matrix effect. After that, the supernatants were analyzed with an ultra-performance liquid chromatography-triple quadrupole mass spectrometer (XEVO TQ MS, Waters, UK). In detail, the ACQUITY UPLC HSS T3 column (1.7 μm , 100 mm x 2.0 mm) (Waters, Manchester, UK) was used with a column temperature of 40 °C. Formic acid (0.1 % (v/v)) was used as solvent A and acetonitrile was used as solvent B. Gradient elution was achieved by solvent A: 0~1.0 min (90%~90%), 2.0~4.0 min (90%~50%), 4.0~5.0 min (50%~20%), 5.0~6.0 min (20%~20%) and 6.0~7.0 min (20%~90%). The injection volume was 10.0 μL and the flow rate was 0.2 mL min^{-1} . Only characteristic ions satisfying all the setting parameters were considered as the test cobalamin (Table S11). The consumption of riboflavin in Firmicutes showed a trend of decrease and then increase (Fig. S31a), which was a different pattern from that of cobalamin. This result indicated that Firmicutes could secrete riboflavin extracellularly. Different trends were also

observed in the accumulation of riboflavin and cobalamin. The accumulation of riboflavin was observed for all the tested species, rather than being secreted only by the key group (Fig. S31b). Riboflavin accumulation showed an increasing trend in all groups, unlike cobalamin, which could only be secreted by the key group.” Please see lines 511-538 on pages 51 and 52, and lines 600-602 on page 57 in Supplementary Information I- Section G.

Fig. S31. Consumption and accumulation of riboflavin. a) The consumption of riboflavin in Firmicutes. b) The accumulation of riboflavin in species of the Key and General groups. Grey shaded areas denote the 95% confidence intervals.

2) Many statistical tests are performed to disentangle ecological effects underlying the community dynamics observed. Please clarify the assumptions and expectations of each model in the text when they are mentioned as it is hard to follow for non-experts.

Response: We thank the reviewer for this suggestion. Per your suggestion, we clarify the assumptions and expectations of each model in the text when they are mentioned. We also restructured the manuscript and Supplementary Information I, to make clear why and how we did this analysis.

It is unclear to us, for example, how you can prove that increased co-occurrence at high temperature is not just due to a few species tolerating those harsh environmental conditions. Please explain these points, as they are key to the conclusions of the paper.

Response: We thank the reviewer for this comment. We added the succession of mean copy number (MCN), which is the average rRNA copy number for each sample to suggest that the increased co-occurrence is caused by high temperature selection of slow growing species (i.e., *Thermobifida fusca* and *Saccharomonospora viridis*).

We added the text in manuscript as “The ribosomal RNA operon (rrn) copy number was calculated by matching Ribosomal RNA Operon Copy Number Database (rrndb), which is a phylogenetically conserved trait and a proxy for maximum growth rate²⁶, to inferred maximum growth rates of neighboring ASVs and ACT ASVs (Text S5 in SI file1). The significantly higher copy number (6.9) of rrn in neighbouring ASVs (6.9) (Fig. S23 in SI file1), and the high explanation of changes in functional potential (51.0%) (Extended Data Fig. 2) emphasized that the neighboring ASVs were fast-growing bacteria, and are highly associated with composting function. In contrast, the lower rrn copy number in ACT ASVs indicated their slow-growing characteristics. Due to the significantly positive correlations between ACT ASVs’ abundance (with lower rrn copy number), and temperature (Fig. S20 in SI file1), it also implied that high temperature might favor slow-growing bacterial species. To confirm this, we calculated the mean copy number (MCN), which is the abundance-weighted

rrn copy number for each sample²⁷, to summarize how temperature affected the distribution of fast and slow growers in the composting bacterial community. The minimum value of MCN was observed on day 5 (Fig. S23b in SI file1). Moreover, the correlation analysis (Fig. S23c in SI file1) and multiple linear regression (MLR) analysis (Fig. S24 and Table S6 in SI file1) supported that temperature was the main factor regulating MCN (Text S5 in SI file1). Therefore, high temperature favour slower-growing bacterial species, and these species (i.e., ACT ASVs) might cooperate with other species that grow rapidly and affect composting functions.” Please see lines 137-158 on pages 7 and 8.

We added mean copy number (MCN) and the related text in Section E in Supplementary Information. “**The ribosomal RNA operon copy number and mean copy number** Since the maximum growth rate of a bacterium is proportional to the number of ribosomal RNA operon (rrn) copies it has, we calculated the rRNA copy number of different composting bacteria by matching ASVs to the Ribosomal RNA Operon Copy Number Database (rrndb) to infer the maximum growth rate of different composting bacteria¹⁶. The rrn copy number was matched the lowest available rank (e.g., if the species and genus levels are unclassified, then the family level would be the representative). Mean copy number (MCN) was calculated as the average rRNA copy number for each sample as described by Abreu et al. (2022). In detail, the MCN was calculated by weighting the rRNA copy number of each ASV and its relative abundance¹⁷. MCN for each sample is the average of three replicates.

Multiple linear regression model We conducted 2 multiple linear regression

(MLG) models to reveal how temperature and nutrients affected MCN: (1) with all 60 samples, (2) with 50 samples (excluding samples collected on Day 00) to confirm the results. In both MLG models, temperature and nutrients were independent variables, and MCN was dependent variable. We used R packages “*tidyverse*” and “*leaps*” to construct the MLR and obtain R^2 for each model (Fig. S25a). Given the equal importance of optimal model and minimal model in the construction of MLG models, we further calculated the Bayesian Information Criterion (BIC) for the different models with a lower BIC indicating a better R^2 .

Results of MLG model showed that although the R^2 for the combination of "temperature" and "TC" was not the largest among the 60 sample group, they had the smallest BIC and was the simplest model, therefore this MLR was the most appropriate for the 60 sample group. Also, the combination of "temperature", "TC", “Cellulose”, and “Lignin” was both the simplest and most optimal model for the 50 samples group (Fig. S24 and Table S6). Temperature was the most important factor in both models, therefore, temperature was the key factor influencing the MCN. Considering the significantly negative correlation between temperature and MCN (Fig. S23), high temperature might therefore favor slow-growing bacteria.” Please see lines 424-453 on pages 40 and 41, lines 456-464 on pages 43-44, and lines 474-477 on page 46 in Supplementary information I- Section E.

Fig. S23. Mean copy number (MCN) during the composting process. a) Ribosomal RNA operon copy number between neighbouring ASVs and ACT ASVs. b) MCN for different sampling time points. c) The relationship between MCN and temperature.

Fig. S24. Multiple linear regression between MCN and environmental factors. a) R^2 for different models. b) Bayesian Information Criterion (BIC) for different models. c) Relative importance of predictor variables. The red box indicates the selected model.

Table S6 The detailed values of the constructed multiple linear regressions between temperature, nutrients and positive cohesion.

Table S6.1 The model including 60 samples

60 Samples	Estimate	Std. Error	t value	Pr(> t)	Significant
Intercept	4.8403	0.21806	22.20	2.0e-16	Yes
Temperature	-0.0363	0.00304	-11.92	2.0e-16	Yes
TC	0.0221	0.00316	6.98	3.4e-09	Yes

$$MCN=4.8403-0.0363*Temperature+0.0221*TC+\epsilon$$

ϵ is a constant in the multiple regression model

TC: Total Carbon

Table S6.2 The model including 50 samples (excluding samples collected from day 0)

50 Samples	Estimate	Std. Error	t value	Pr(> t)	Significant
Intercept	5.3048	0.23823	22.27	2.0e-16	Yes
Temperature	-0.0218	0.00499	-4.36	7.4e-05	Yes
Cellulose	-0.3294	0.09060	-3.64	0.00071	Yes
Lignin	0.2582	0.12234	2.11	0.04042	Yes
TC	0.0222	0.00825	2.70	0.00985	Yes

$$MCN=5.3048-0.0218*Temperature-0.3294*Cellulose+0.2582*Lignin+0.0222*TC+\epsilon$$

ϵ is a constant in the multiple regression model

TC: Total Carbon

Minor comments

1) *Of course, it would have been optimal to isolate the Actinobacteria from the compost community rather to purchase them. However, we understand that if these species could not be isolated at the time of the experiment, purchasing similar strains is appropriate.*

a. Can the authors comment on why the Actinobacteria strains could not be isolated in the first place while Firmicutes were?

Response: We thank the reviewer for this comment. Actually, we isolated both Firmicutes and Actinobacteria. To ensure the representation of ACT species, we decided to purchase these strains after confirming the sequence similarity.

b. Please provide comparative genomics information showing that the strains purchased carry the same metabolic capacities (especially for the cobalamin

pathways) as the strains identified in the compost community with the MAGs.

Response: We thank the reviewer for this suggestion. Per your suggestion, we sequenced *Thermobifida fusca* and *Saccharomonospora viridis* and compared their genomes with MAG. 73 and MAG. 406, respectively. Results showed that the average nucleotide identity (ANI) between *Thermobifida fusca* and MAG. 73 was 99.63% and that between *Saccharomonospora viridis* and MAG. 406 was 99.99%. Meanwhile, similar pathways of cobalamin synthesis could be observed between the purchased strains and the MAGs (i.e., aerobic synthesis pathway was complete and the anaerobic was incomplete).

We added text, figure, and table in Supplementary Information “**Extraction of genome DNA** Genomic DNA for *Thermobifida fusca* and *Saccharomonospora viridis* were extracted with the GenElute™ Bacterial Genomic DNA Kit Protocol (NA2100, SIGMA, Japan), following the manufacturer’s instructions. DNA concentration and purity were measured using Qubit 3.0 (Thermo Fisher Scientific, Waltham, USA) and Nanodrop One (Thermo Fisher Scientific, Waltham, USA) at the same time.

Library preparation and sequencing Sequencing libraries were generated using the ALFA-SEQ DNA Library Prep Kit. The library quality was assessed on the Qubit 4.0 Fluorometer (Life Technologies, Grand Island, NY) and Qsep400 High-Throughput Nucleic Acid Protein Analysis system (Houze Biological Technology Co, Hangzhou, China). Finally, the library was sequenced on an Illumina Novaseq6000 platform (MAGIGENE Biological Technology Co. Ltd Guangzhou, China). and 150 bp paired-end reads were generated.

Data preprocessing The raw data were quality controlled, and the high-quality sequences were used for downstream data analysis. Specific processing steps are as follows: (1) Remove low mass bases at both ends of reads (mass value <20), and remove short reads (default setting was 50bp); (2) Remove reads with a specific number of N bases (default setting was 10bp); (3) Remove reads between overlap exceeding a certain threshold (default set as 15bp) and Adapter; (4) Remove duplication pollution.

De novo Genome Assembly The reads with low sequencing quality values were filtered by quality control steps, and the clean data after data filtering and quality control were de novo assembled using SPAdes v3.13.0 to obtain high-quality contigs fragments.

Comparative genomics analysis Comparative genomics analysis were carried out between *Thermobifida fusca* and *Saccharomonospora viridis* and MAG. 73 and MAG. 406, respectively. The details are as follow: (1) Average nucleotide identity (ANI) is one of the most powerful measures to identify the kinship distance of bacterial genomes. When ANI > 95% it indicates that two genomes belong to the same species. (2) Collinearity between the genomes was identified by performing an alignment between the target genome and the reference genome, using MUMmer software (Version 3.23).

Results showed that the ANI between MAG 73 and *Thermobifida fusca* was 99.8%, and MAG 406 and *Saccharomonospora viridis* was 99.6% (Fig. S33 and Table S8). Moreover, KEGG analysis confirmed that biosynthesis pathway of cobalamin in

the two purchased strains was also consistent with that of MAG 73 and MAG 406 (Fig. S33c). Overall, these results indicated that two purchased strains were consistent with MAG 73 and MAG 406. ” Please see lines 539-577 on pages 52-53, lines 607-613 on page 59 and lines 615-617 on page 60 in Supplementary information I-Section G.

Fig. S33. Comparative genomic analysis between purchased strains and MAGs. a) Comparative genomic analysis between MAG. 73 and *Thermobifida fusca*. Upper line represents the genome of MAG. 73 and lower line represents *Thermobifida fusca*. b) Comparative genomic analysis between MAG. 406 and *Saccharomonospora viridis*. Upper line represents the genome of MAG. 406 and lower line represents *Saccharomonospora viridis*. c) The biosynthesis pathway of cobalamin in the purchased strains.

Table S8 Comparative Genomics

Table S8.1 The genomic details of the purchased strains

Strain	Total_length (bp)	Mean_length (bp)	N50 (bp)	N90 (bp)	GC (%)
Thermobifida fusca	3685329	89886.07	204614	75707	67.47
Saccharomonospora viridis	4285697	389608.82	973787	435231	67.35

Table S8.2 Comparative genomes between purchased strains and MAGs

Strain	Reference MAG	ANI (%)	Mapped fragment	Query fragment
Thermobifida fusca	MAG. 73	99.8	1103	1153
Saccharomonospora viridis	MAG. 406	99.6	1201	1257

c. Mention clearly in the main text that the strains used *in vitro* were different from the ones in the community.

Response: We thank the reviewer for this comment. We revised Fig S3 to include the genera of used strains (i.e., *Sphingomonas* and *Idiomarina*). We also checked throughout the manuscript to avoid this mistake.

Please see lines 73-76 on page 6 in Supplementary information I- Section A.

Fig. S3. The overall community structure of the composting bacterial with 12 most abundant taxa at the genus level. **a)** The overview of 12 most abundant genera in 180 samples. **b)** The abundance of 12 most abundant genera in 6 groups set during the sampling period. **c)** The PCoA analysis with 6 groups at the genus level.

2) *Our understanding of the study is that it is mostly about the ecological dynamics of the community throughout the temperature changes of the composting process. Yet, when the idea of the Black Queen Hypothesis is mentioned, we are now talking about evolution of the community. As the heating-cooling cycle of the composting process is not cyclically experienced by the same community (in the sense that this process would typically happen only once), do you think there are enough selection forces to promote the evolution of cooperation among species growing at high temperatures? Please consider adding a few sentences about this in the discussion.*

Response: We thank the reviewer for this suggestion. Per your suggestion, we added the following text in the discussion.

“The imbalance in soil and marine systems could be supported by the BQH that most microbes would live more efficiently by reducing the number of burdensome genes they carry and benefit from the products of other microbes⁴⁰. Although the composting community experienced the heating-cooling cycle one time, in addition to selecting target species, high temperatures might also act as a selection force to promote cooperation evolution among species.” Please see lines 349-355 on page 17.

3) *Throughout the manuscript, authors use the term “key” species, even once mentioning “keystone” species. This “key” term is a bit confusing. Why not use “Actinobacteria”?*

Response: We thank the reviewer for this suggestion. We revised the key species/ ASVs to avoid misleading. Considering that not all Actinobacteria ASVs satisfy the four filtering criteria, we defined the ASVs, satisfying the four filtering criteria, as ACT ASVs and checked throughout the manuscript.

We added the following text: “These three ASVs belonged to the Actinobacteria, with two ASVs (ASV2 and ASV4) classified as *Thermobifida fusca* and one ASV (ASV1) classified as *Saccharomonospora viridis*. To distinguish them from other Actinobacterial ASVs, we named these 3 ASVs as ACT ASVs.” Please see lines 125-128 on page 7.

Furthermore, we revised “keystone species” to “keystone nodes” to refer to the topologically important nodes (ASVs) in the networks (Yuan et al., 2021).

We added definitions of keystone nodes in the supplementary information: “Defining keystone species or taxa, i.e., those with a high impact on the structure and functioning of ecosystems, is a critical, but difficult issue in ecology, especially in microbial communities. Based on the topological properties of networks, keystone nodes could be defined as module memberships. But such keystone nodes are not equivalent to keystone species or taxa typically used in ecology. Thus, in this study, we used keystone nodes instead of keystone species or taxa to refer to the topologically important nodes (ASVs) in the networks. For each node included in the constructed network, its Z_i value (i.e., within-module connectivity) and P_i value (i.e., among-module connectivity) were calculated to classify its topological role, including peripheral nodes ($P_i \leq 0.62$, $Z_i \leq 2.5$), network nodes ($Z_i \geq 2.5$, $P_i \geq 0.62$), connectors ($Z_i < 2.5$, $P_i \geq 0.62$) and module nodes ($Z_i \geq 2.5$, $P_i < 0.62$)^{9,13,14}. All but the peripheral nodes were keystone nodes⁹. In our results, the keystone nodes included module nodes (i.e., nodes highly connected to other members in a module) and connectors (i.e., nodes linking different modules) (Fig. S21). We chose a 50% cutoff because the network reliability degraded significantly for OTUs containing >50% zeroes, and 50% cutoff was the recommended level¹⁵. Indeed, 50% OTU removal might exclude nondominant ASVs, but a false positive is likely more destructive¹⁵. In conclusion, all of the selected ACT ASVs were keystone nodes, which highlighted their importance.” Please see lines 385-403 on page 34 in Supplementary information I- Section D.

Reference

1. Yuan, M. M., Guo, X., Wu, L., Zhang, Y., Xiao, N. & Ning, D. et al. Climate

warming enhances microbial network complexity and stability. Nat Clim Change. 11, 100-343 (2021).

4) *Introduction:*

a. *“Bacteria can adapt to high temperature without collapsing”. We suggest changing “adapt” for “acclimatize” as in this context it is about ecology, not evolution.*

Response: We thank the reviewer for this suggestion. Per your suggestion, we revised “adapt” to “acclimatize”.

Please see lines 79 on page 4.

b. *Please define “poikilothermic”.*

Response: We thank the reviewer for this comment. As poikilothermic refers to animals that cannot control their internal temperature, we revised “poikilothermic” to “a biosystem characterized by deterministic temperature variation” and check throughout the manuscript.

We revised the title to “Inter-bacterial mutualism promoted by public goods in a biosystem characterized by deterministic temperature variation”. We revised the sentence to “Here, we monitored a quasi-natural community (composting) in a biosystem, where temperature (20°C-70°C) was the main abiotic stress.” Please see lines 1-2 on page 1 and lines 25-26 on page 2.

5) *Results:*

a. *Section 2.1: Please explain why the rrn copy number is a good indicator (you can use the same explanation used in the discussion)*

Response: We thank the reviewer for this suggestion. Indeed, rrn copy number is important because it is a proxy for maximum growth rate.

We added the following text: “The ribosomal RNA operon (rrn) copy number was calculated by matching Ribosomal RNA Operon Copy Number Database (rrndb), which is a phylogenetically conserved trait and a proxy for maximum growth rate²⁶, to inferred maximum growth rates of neighboring ASVs and ACT ASVs (Text S5 in SI file1).” Please see lines 137-141 on page 7.

6) *Discussion:*

a. *“This notion could be supported by theoretical predictions such as the Black Queen Hypothesis (BQH) which states that the cross-feeding of “expensive” public goods is evolutionarily selected for”. This is not exactly the BQH, as it is the loss of costly traits that is selected for, not the cross-feeding. We suggest changing the text to: “This notion could be supported by theoretical predictions such as the Black Queen Hypothesis (BQH) which states that losing the production of “expensive” public goods is evolutionarily selected for and can result in obligate cross-feeding.”*

Response: We thank the reviewer for this suggestion. We revised this sentence as your suggestion.

Please see lines 331-334 on page 16.

b. *“slow-growing, but stress-tolerant, species preferred to cooperate with others though the sharing of public goods”*. We suggest changing the wording as *“preferred”* is anthropomorphic and poorly reflects that *“sharing public goods”* is generally a passive mechanism in the form of unvoluntary leakage of metabolites to the environment.

Response: We thank the reviewer for this suggestion. Per your suggestion, we used “leaking metabolites” to highlight the passive mechanism.

We revised this sentence to “slow-growing, but stress-tolerant, species cooperate with other bacteria by leaking metabolites.” Please see lines 93-94 on page 5.

7) Fig. 1a:

a. *The red line doesn't seem to add value to the plot and doesn't fit the data so well. We suggest removing it.*

Response: We thank the reviewer for this suggestion. Per your suggestion, we removed the red line.

Please see lines 828-839 on pages 36 and 37.

b. *The thermophilic phase is described as starting when the temperature reaches 50°C. At the bottom of the plot, the initial phase is shown to last 5 days while the 50°C threshold is reached on day 2. Please adjust it or explain why it is this way.*

Response: We thank the reviewer for this suggestion. To avoid misleading, we have removed the line at the bottom of the plot.

Please see lines 828-839 on pages 36 and 37.

Fig.1 Changes of temperature and positive cohesion. **a)** Temperature variations in the 10 composting piles. 50°C is generally considered to be the dividing line for composting in terms of distinguishing the high temperature phase from other phases in this system. **b)** Changes in the positive cohesion of different phases. In the boxplots of panels, hinges indicate the 25th, 50th, and 75th percentiles, whiskers indicate $1.5 \times$ interquartile ranges, and dots indicate values of individual samples. **c)** Relationship between positive cohesion and temperature. **d)** Screening for the ACT

ASVs (detail of 4 criteria are described in Methods section). **e)** Whole composting network constructed by the random matrix theory (RMT) model. The red line indicates a positive correlation and the blue lines a negative correlation. **f)** The sub-network containing ACT ASVs and their neighbors. Different colors of the outer ring are used to distinguish taxonomic affiliation.

8) *Fig 3a: The legend is not detailed enough to understand the figure, please detail what dots and bars are representing.*

Response: We thank the reviewer for this suggestion. Bars representing the number of sets and dots representing the composition.

We included additional legend text to explain the meaning of the points and bars: “Different sets are represented in different columns, with bars representing the number of sets and dots representing the composition (black: include, gray: exclude). For example, the second column refers that 4 modules (bar) are completed in both Groups V and VII (black point), but not in the other groups (gray point).” Please see lines 850-854 on page 39.

9) *Fig 3c: It is unclear what the ‘total’ category represents. Please add description in the legend.*

Response: We thank the reviewer for this suggestion. Total refers to the number of MAGs containing each of these three genes.

We added the text in legend “Total refers to the number of MAGs containing

each of these three genes.” Please see line 856-857 on page 40.

10) Fig 3c-f: the legends are too small and hard to read, please consider increasing the font size.

Response: We thank the reviewer for this suggestion. Per your suggestion, we revised Fig 3c-f.

Please see lines 846-864 on pages 39 and 40.

Fig. 3 The biosynthesis and demand for cobalamin during composting based on metagenomic and metatranscriptomic sequencing. a) The number of unique and shared modules related to the complete biosynthesis of amino acids, vitamins/co

factors, and lipids/fatty acids between groups I–VII. Group VI, to which MAG73 and MAG406 belong, is marked in orange. Different sets are represented in different columns, with bars representing the number of sets and dots representing the composition (black: include, gray: exclude). For example, the second column refers that 4 modules (bar) are completed in both Groups V and VII (black point), but not in the other groups (gray point). **b)** Three-step synthesis of cobalamin. **c)** Proportion of MAGs encoding three cobalamin-dependent genes (i.e. *mutA*, *metH*, and *rsmB*) and those encoding the complete cobalamin biosynthesis pathway. Total refers to the number of MAGs containing each of these three genes. **d)** Proportion of MAGs transcribing the three cobalamin-dependent genes and those transcribing the complete cobalamin biosynthesis pathway. **e)** RPKM of the three cobalamin-dependent genes at different sampling points. **f)** RPKM values for the three-step synthesis of cobalamin at different sampling points. **g)** Cobalamin requirement of MAGs transcribing nitrogen cycle-related functional genes. **h)** Cobalamin requirement of MAGs transcribing cellulose degradation-related functional genes. The line thickness indicates the number of MAGs transcribing the functional gene.

11) Extended Fig. 1a: typo in the plot labels: ‘Hemicellose/Cellose’ should be ‘Hemicellulose/Cellulose’?

Response: We thank the reviewer for this suggestion. We have revised ‘Hemicellose/Cellose’ to ‘Hemicellulose/Cellulose’. Please see Extended Fig. 1.

12) Extended Fig. 1b-c: Please add in the legend what the arrows color stand for

Response: We thank the reviewer for this comment. We have added the text: “The arrows color indicate the different process in carbon fixation.” and “The arrows color indicate the different process in nitrogen cycling.” Please see Extended Fig. 1b-c.

13) Extended Fig. 1d-f: We understand the legend as “T1/T2/T3 groups” on the x-axis to be the “Initial/Thermophilic/Cooling groups” as stated in the box with the colors. Please consider changing the x-axis labels for simplicity.

Response: We thank the reviewer for this suggestion. To avoid the problem of pseudoreplication, we only focused the sampling time points in the revised manuscript and removed the origin in Extended Fig. 1d-f. We have checked throughout the manuscript to avoid similar mistakes.

Extended Data Fig. 1. Changes in functional genes related to carbon and nitrogen cycling. a) The copies of functional genes related to carbon degradation. b) The copies of functional genes related to carbon fixation. The arrows color indicate the different process in carbon fixation. c) The copies of functional genes related to nitrogen cycling. The arrows color indicate the different process in nitrogen cycling. d) The relationship between positive cohesion and the abundance of genes related to carbon degradation. e) The relationship between positive cohesion and the abundance of genes related to carbon fixation. f) The relationship between positive cohesion and the abundance of genes related to nitrogen cycling.

14) *Extended Fig. 2: Please be more specific in the legend: “based on correlation” between what and what?*

Response: We thank the reviewer for this comment. The correlation were calculated between the relative abundance of neighbouring ASVs and the abundance of the functional genes.

We added the following text: “Contributions of key ASVs to composting ecological functions (carbon degradation, carbon fixation, and nitrogen cycling) based on correlation and best multiple regression model between the relative abundance of neighbouring ASVs and the abundance of the functional genes.” Please see Extended Fig. 2.

Extended Data Fig. 2. Contributions of key ASVs to composting ecological functions (carbon degradation, carbon fixation, and nitrogen cycling) based on correlation and best multiple regression model between the relative abundance of neighbouring ASVs and the abundance of the functional genes. Circle size represents the relative importance of the variable (i.e. the multiple regression modeling and variance decomposition analysis were calculated to reveal the proportion of explained variability). The heatmap indicates two-sided Pearson's correlation coefficients between the relative abundance of ASVs and abundance of functional gene.

15) Extended Fig. 3: Please be more specific in the legend: How did you obtain the red rectangles that differentiate the groups

Response: We thank the reviewer for this suggestion. We got the red rectangle by the difference in the completeness of KEGG modules in each group (by the colour of the heatmap).

We added the text: “Red rectangles are plotted on the heatmap to show

differences between groups based on the completeness of the same module (each row) in different groups after clustering.” Please see Extended Fig. 3.

Extended Data Fig. 3. Metabolic profiling of the bacterial community based on KEGG module completeness. Heatmap showing clustering of genomes (rows) by their KEGG module completeness (columns). Completeness ranges from 1 (green) to zero (white). Red rectangles are plotted on the heatmap to show differences between groups based on the completeness of the same module (each row) in different groups after clustering.

16) Fig. S8b: What are all these indexes representing? Is the point to show that indexes values change over time? Or that some are better than other to describe the system? Please explain in the legend.

Response: We thank the reviewer for this suggestion. These indices are the topology of the network and are used to describe the structure of the network. We wanted to indicate the succession of the networks in response to the sampling time

points in Fig. S8, rather than highlight these indexes. Considering the possibility of misleading and overlapping information from the original Fig. S8b, we removed it and the original Fig. S8a is retained as new Fig. S8. We displayed the detail value of these indexes in Table S13 in SI file II, with an explanation of each indexes.

Please see lines 239-245 on page 19 in Supplementary information I – Section B and Table S13 in SI file II.

The explanation of topological indexes are as following “Total nodes: Total nodes in the constructed networks. (2) Total links: Total links in the constructed networks. (3) R square of power-law: It indicates the degree of fit to the power-law model. (4) Average degree (avgK): Higher avgK means a more complex network. (5) Average clustering coefficient (avgCC): It is used to measure the extent of module structure present in a network. (6) Average path distance (GD): A smaller GD means all the nodes in the network are closer. (7) Geodesic efficiency (E): It is geodesic efficiency. (8) Harmonic geodesic distance (HD): The reciprocal of E, which is similar to GD but more appropriate for disjoint graph. (9) Centralization of degree (CD): It is close to 1 for a network with star topology and in contrast close to 0 for a network where each node has the same connectivity. (10) Centralization of betweenness (CB): It is used to describe the degree of a central node that it is connected to other central nodes. (11) Centralization of stress centrality (CS): It is close to 0 for a network where each node has the same stress centrality, and the bigger the more difference among all stress centrality values. (12) Centralization of eigenvector centrality (CE): It is close to 0 for a network where each node has the same eigenvector centrality, and the bigger

the more difference among all eigenvector centrality values. (13) Maximal closeness centrality: It is close to 0 for a network where each node has the same eigenvector centrality, and the bigger the more difference among all eigenvector centrality values. (14) Centralization of closeness centrality (CCL): It is close to 0 for a network where each node has the same stress centrality, and the bigger the more difference among all stress centrality values. (15) Density (D): It is closely related to the average connectivity. (16) Transitivity (Trans): Sometimes it is also called the entire clustering coefficient. It has been shown to be a key structural property in social networks. (17) PorP: Proportion of positive links. (18) TotalP: Total number of positive links.”

Fig. S8. Succession of the networks in response to the sampling time points. The construction method was same as previous described. Seventeen network topological parameters (calculated on the MENAP interface) were used for a principal co-ordinates analysis (PCoA) analysis to give an overview for the variation in the network at different sampling time points. The detailed values of these network topological parameters are shown in Table S13 in SI file II. Arrows clarify the chronological order of the corresponding network.

17) Fig. S14: Please be more specific in the legend to describe the colors, square size and line size.

Response: We thank the reviewer for this suggestion. Colour gradient and square size denote Pearson’s correlation coefficient. The line size indicates the Mantel’s r statistic.

We added the following text: “Pairwise comparisons of environmental factors are shown in the triangle, with the colour gradient and square size denoting Pearson’s correlation coefficient. The line size indicates the Mantel’s r statistic between each environmental factor and positive cohesion, total positive links, and proportion of positive links. The color indicates positive (red) and negative (blue) correlation.”

Please see lines 337-341 on page 28 in Supplementary information I- Section C.

Fig. S14. Mantel tests. a) Relationship between environmental factors and positive cohesion. b) Relationship between environmental factors and total positive links. c) Relationship between environmental factors and proportion of positive links. Pairwise

comparisons of environmental factors are shown in the triangle, with a colour gradient and square size denoting Pearson’s correlation coefficient. The line size indicated the Mantel’s r statistic between each environmental factors and positive cohesion, total positive links, and proportion of positive links. The color indicated the positive correlation (red) and negative correlation (blue).

18) Fig. S17: Please be more specific in the legend: What does Eligible / Ineligible mean?

Response: We thank the reviewer for this suggestion. Eligible indicates ASVs that meet this screening criteria. Ineligible indicated ASVs that did not meet the screening criteria

We added the text “Eligible indicates ASVs that meet this screening criteria (e.g. only top 0.1% of mean relative abundance ASVs are considered to be “eligible” for overall abundance ASVs). Ineligible indicated ASVs that did not meet the screening criteria.” Please see lines 413-416 on page 37 in Supplementary information I-Section D.

Fig. S19. Relative abundance and percentage of the composting ASVs. a) Overall

abundant ASVs. **b)** Ubiquitous ASVs. **c)** Frequently abundant ASVs. **d)** ASVs associated with positive cohesion (PC). **e)** ACT ASVs (ASVs fulfilling all 4 conditions). The outer circle indicates the total relative abundance of ASVs. The inner circle indicates the percentage of ASVs. Eligible indicates ASVs that meet this screening criteria (e.g. only top 0.1% of mean relative abundance ASVs are considered to be “eligible” for overall abundance ASVs). Ineligible indicated ASVs that did not meet the screening criteria.

19) Please check the figures referencing in the text (especially in supplementary text), as there seems to be many errors.

Response: We thank the reviewer for this suggestion. Per your suggestion, we corrected the mistakes and checked throughout the manuscript.

Reviewer #3 (Remarks to the Author):

Zhao et al has done a thorough study of microbiomes in compost which has employed many complementary approaches including networks, coculturing, functional gene analysis, and chemical testing. I strongly applaud the authors for their efforts to provide a synthetic story with a proposed mechanism for beneficial interactions within the microbiome. This topic of understanding interactions within the microbiomes – especially outside of gut microbiomes – is at the cutting edge of the microbiome field right now and thus is very timely.

While there is much to like about this work, I do have serious reservations that I describe below.

Response: We are grateful for these comments. According to your comments and suggestions, we have carefully revised the manuscript. The response text including: a. Black italic type: the exact comment. b. Indent and in normal font: response to the comment. c. Indent and in blue: revisions in manuscript. Below we provide our point-by-point responses to your comments and hope the revised manuscript will meet with your approval.

1. Pseudoreplication and experimental design/analysis:

It is very hard to tell exactly what the authors have done in their statistical analyses based on what is in the manuscript and supplements, but I believe there are some substantial issues with the analyses based on what I can glean from the text. In general, I suggest the authors be more explicit about how the data are being analyzed

in any revision regardless of whether it goes back to Nature Comm. I understand that it may need to be in the supplement rather than the main text, but I do not see anywhere that specifies the structure of statistical models.

Response: We thank the reviewer for this comment. Per your suggestion, we revised our manuscript to highlight why and how we carried out this analysis. We restructured the manuscript and Supplementary Information I, to make it clearer.

Here are a couple of examples of issues with the design and statistics. What is an independent replicate in the sampling design? There are 180 samples sequenced coming from 10 piles that the authors measure at 6 time points with 3 technical reps/subsamples. There are only 10 piles within each time point and that would be the independent replicate for this experimental design.

Response: We thank the reviewer for this comment. Exactly as you say, 10 piles within each time point were the independent replicates. We revised the “180 samples” to “60 samples” and check throughout the manuscript to clarify our experimental design.

We revised the text in Results to “To reveal changes of potential bacterial interactions under these temperature variations, a total of 60 bacterial communities (including 6 sampled points for each pile) were determined with high-throughput 16S rRNA gene sequencing (V4 region) (Section A in SI file1).”; “The 60 samples were combined according to the different composting phases (10 samples mixed into 1 depending on sampling time) to obtain 6 combined samples, for metagenomic and

metatranscriptomic sequencing.” Please see lines 99-102 on page 5, lines 165-167 on page 8.

We revised the text in Materials and methods to “60 samples of the composting microbiome (including 10 different piles, 6 different sample times) was extracted with the DNeasy PowerSoil Kit (12888-50, Qiagen, Germany) following the manufacturer’s instructions, in triplicate.”; “In conclusion, a total of 180 high-throughput sequencing samples were measured for 60 samples (3 replicates per sample), and the average of the parallel samples was representative of the sampling time point.”; “A total of 60 DNA samples were analyzed by high-throughput qPCR for 54 functional genes (including carbon degradation, carbon fixation and nitrogen cycling) on a Wafergen Smart Chip Real-Time qPCR platform as previously described⁵⁴, with a threshold cycle of 31 as the detection limit⁵⁵, in triplicate.”; “Overall, 18 DNA samples (6 different sampling time points with 3 replicates each) were obtained for further metagenomic sequencing.”; “In conclusion, a total of 18 metagenomic sequencing samples were measured for 6 sampling time points (3 replicates each), and the average of the replicate samples was representative of a sampling time point.”; “In conclusion, a total of 18 metatranscriptomics sequencing samples were measured for 6 sampling time points (3 replicates each), and the average of the replicate samples was representative of a sampling time point.”; “To ensure the reliability of the molecular ecological network (MEN)⁶³, only ASVs present in at least 30 out of the total 60 samples were included for further correlation calculations.” Please see lines 409-411 on page 20, lines 426-429 on pages 20, lines

431-434 on page 21, lines 442-443 on page 21, lines 466-468 on page 21, lines 485-488 on page 23, and lines 490-492 on page 23.

-- If “time point” is the variable of interest, then there would only be 10 independent replicates per time (since subsamples are not independent replicates). In some places time point appears to be the variable of interest; for instance, in Figure S8 the authors show a trajectory of 6 networks. I don’t know if the authors used their subsamples/technical reps as separate input into this network analysis to get up to 30 samples but that would be incorrect since it would inflate replication with non-independent samples (pseudoreplication). Alternatively, they may have used only 10 replicates (1 from each pile by averaging or randomly selecting 1 subsample) which would avoid pseudoreplication.

Response: We thank the reviewer for this comment. “Time point” is the variable of interest, and 10 piles within each sampling time point were the independent replicates. Following your suggestion, we used the mean of three sequencing replicates for further analysis of bacterial communities, and if not specifically stated, all 60 samples were included in the analysis. **We revised all related analysis, texts and figures. The revised figures included: Fig. 1, Extended Data Fig. 1, Extended Data Fig. 2, Fig. S2 - Fig. S24, Table S1-S6.** With the exceptions of Tables S1-S3, all modifications had a consistent result in that the conclusions were the same as before, although the detail values changed. Tables S1-S3 showed different results in that the significance between 6 groups was caused by the homogeneous dispersion (difference

in composition between groups), rather than heterogeneous dispersion (differences in composition within groups) as before. The significant differences between 6 groups obtained by Adonis analysis met its assumption, and were reliable. We believe that this was caused by the original erroneous inclusion of pseudo-replicates in the statistical analysis. We again thank the reviewer for pointing this out.

However, that would also be an issue since this would be low sample number for network construction.

Response: We thank the reviewer for this comment. 10 samples meet the minimum number of samples (>8) to construct a network (Deng et al., 2012). Also, Morriën et al., 2017 (9 repeats), Cameron et al., 2019 (8 repeats) and Anni et al., 2020 (8 repeats) used < 10 samples to construct their networks. Nevertheless, your concerns are valid and using 10 samples per group is probably a low sample number.

We add a sentence to clarify the generalization of the results: “However, due to the limitation of sample numbers, there is a need yet to further verify these findings in other chronosequences or composting processes as well.” Please see lines 391-392 on page 19.

References

1. Deng, Y., Jiang, Y., Yang, Y., He, Z., Luo, F. & Zhou, J. Molecular ecological network analyses. *Bmc Bioinformatics*. 13, (2012).
2. Djurhuus, A., Closek, C. J., Kelly, R. P., Pitz, K. J., Michisaki, R. P. & Starks, H. A. et al. Environmental DNA reveals seasonal shifts and potential interactions in a marine community. *Nat Commun*. 11, (2020).
3. Wagg, C., Schlaeppli, K., Banerjee, S., Kuramae, E. E. & van der Heijden, M. G. A.

Fungal-bacterial diversity and microbiome complexity predict ecosystem functioning. *Nat Commun.* **10**, (2019).

4. Morrien, E., Hannula, S. E., Snoek, L. B., Helmsing, N. R., Zweers, H. & de Hollander, M. et al. Soil networks become more connected and take up more carbon as nature restoration progresses. *Nat Commun.* **8**, (2017).

-- *If the authors are grouping timepoints into three phases (as in figure 1b) to make “phase” the variable of interest, then pseudoreplication is likely an even bigger issue since community 1 at time point 2 is not independent of community 1 at time point 3 or 4, but it seems like data from community 1 at timepoints 2, 3, and 4 are being treated as independent replicates and subsamples within these time points are being treated as independent replicates. Both of these would be serious pseudoreplication issues.*

Response: We thank the reviewer for this comment. “Time point” is the only variable of interest, and 10 piles within each sampling time point were the independent replicates. Indeed, using both sampling time points and phase would cause the pseudoreplication issues and confuse the conclusions. We have removed all the analyses relating to the phases and only highlighted time points. We used the mean of three sequencing replicates for further analysis of bacterial communities, and if not specifically stated, all 60 samples were included in the analysis. **We revised all related analysis, texts and figures. The revised figures included: Fig. 1, Extended Data Fig. 1, Extended Data Fig. 2, Fig. S2 - Fig. S24, Table S1-S6.** Detail could be observed as above.

2. Missing information/difficult to find information -- Another issue is there are a lot of places where not enough information is given to understand why the authors are doing something or what it means. I understand space is limited in the main text and short format papers are really hard to write, but I found I had a lot of questions about basic things reading the paper that impeded my understanding and it took a lot of work on my part to try to get answers. Even after looking at all materials in the supplement etc, I cannot tell what the authors did in many cases (e.g., how many samples went into some analyses, what was the structure of their statistic models, etc).

Response: We thank the reviewer for this comment. Per your suggestion, we revised our manuscript to highlight why and how we carried out this analysis. We restructured the manuscript and Supplementary Information I to make it clearer.

To help the authors improve their manuscript, I will highlight a few examples where the manuscript is unclear and not enough information is provided. In these cases, I spent a lot of time trying to piece information together to understand what is going on, but that level of time investment is not realistic for the average reader and did not answer my questions in some cases. This was a repeated problem throughout the manuscript and the paper could be improved by a careful evaluation of the entire text for areas where this happened. I think this is partially due to the breadth of analyses and approaches included but still needs to be resolved.

Response: We thank the reviewer for this comment. As you mention, the original description of the experimental designs and statistic methods were not enough for the

average reader. Per your suggestion, we added more detailed information and rewrite the Supplementary Information I to make it more clear.

Table S1 – what is this a test of? The table title is “Tab. S1 Significance test for differences between 6 groups and 3 groups” but that doesn’t tell me what the response variable is or why this analysis is being done. I looked in the main text for where these analyses are being called and it says “Thermophilic phase (days 5, 12, and 20) and Cooling phase (days 25 and 30) (Fig 1a), according to the composting phase (Table S1-S3 and Text S1 in SI file1).” However, this doesn’t clarify what the response variable is either. When I look at Fig 1a, it is a graph of temperature. So maybe the response variable is temperature? However, that graph shows many box plots not grouping on 6 vs 3 which is what the table professes to be about.

Response: We thank the reviewer for this comment. We initially wanted to illustrate the feasibility of splitting into 3 groups by comparison. The response variable was the grouping method (6 group: sampling points, 3 group: different periods splitting by temperature). To avoid the pseudoreplication, we removed all the analyses which were split into 3 group. Thus, we only showed the differences in composting microbes between different sampling points. Adonis analysis was carried out to explore whether there existed significance differences in composting microbes between different sampling points on the basis of Bray–Curtis distance in Table S1 via the R package “vegan” (Anderson, et al., 2001; Oksanen et al., 2012). We further calculated Multivariate homogeneity of groups dispersions in Table S2 to test whether

the significance was caused by the artifact of heterogeneous dispersion (differences in composition within groups). Results showed that the significance was caused by homogeneous dispersion ($p > 0.05$ in variances test), which met the “one assumption” for Adonis.

We added the following text: “We used Adonis analysis to test the differences of bacterial communities between groups, and the grouping method is described in Text S1.1 (i.e., based on 6 sampling time points). 60 samples were included in this analysis. The Adonis analysis was carried out on the basis of Bray–Curtis distance using the R package “*vegan*”^{1,2}. Since the number of taxonomically unassigned reads per sample dramatically increased at the species level, the community composition was recorded at Phylum, Class, Order, Family, Genus, and ASV levels. Results showed that the significant differences could be observed in all tested levels (Table S1). The significance in Adonis analysis might be caused by homogeneous dispersion or heterogeneous dispersion. Only homogeneous dispersion meets the assumption for Adonis analysis, therefore, we further calculated multivariate homogeneity of group dispersions (variances) to test the homogeneity of dispersion among groups using the R package “*vegan*”. No significance ($p > 0.05$) was observed at any of the levels tested, indicating that the result was influenced by the difference in composition between groups (homogeneous dispersion) rather than within groups (heterogeneous dispersion) (Table S2). Thus, the significant differences obtained by Adonis analysis met its assumption, and were reliable. Thus, bacterial communities were significantly different at all tested levels when grouped by sampling time points.” Please see lines

39-57 on pages 2 and 3 and lines 83-85 on page 8 in Supplementary information I-

Section A.

Tab. S1 Adonis analysis of bacterial community

Group pairs	Sum.sq	R2	F	Pr(>F)	Significant
Phylum	1.2589	0.2093	2.8595	0.01	Yes
Class	1.3042	0.2130	2.9231	0.01	Yes
Order	3.1258	0.4276	8.0693	0.001	Yes
Family	2.3773	0.3849	6.7573	0.001	Yes
Genus	1.8565	0.3623	6.1355	0.001	Yes
ASV	63841823.3	0.2194	3.0361	0.001	Yes

Table S2 Multivariate homogeneity of groups dispersions (variances) of bacterial community

Group pairs	Sum.sq	Mean.sq	F	Pr(>F)	Significant
Phylum	0.1098	0.0220	1.2314	0.2750	NO
Class	0.1051	0.0210	1.2319	0.3120	NO
Order	0.1082	0.0216	1.3991	0.2380	NO
Family	0.0944	0.0189	1.2074	0.3290	NO
Genus	0.0801	0.0160	1.1798	0.3340	NO
ASV	5140395	1028079	1.2093	0.3320	NO

Reference

1. Anderson, M. J. A new method for non-parametric multivariate analysis of variance. *Austral Ecol.* 26, 32-46 (2001).

2. Oksanen, J., Blanchet, F. G., Kindt, R., Legendre, P. & Wagner, H. *Vegan: Community Ecology Package*. R package version 2.0-3. R Foundation for Statistical Computing, Vienna, Austria., (2012).

Also, some of the other supplementary material is hard to understand due to lack of information (e.g. Fig S5 doesn't define Hes, DL, Hos, etc in caption).

Response: We thank the reviewer for this comment. HoS is the abbreviation of

homogeneous selection. HeS is heterogeneous selection. DL is dispersal limitation. DR is drift. HD is homogenizing dispersal.

We added the details of “Hes” , etc in the legend of Fig S6 “The bacterial community assembly process includes deterministic processes (homogeneous selection (HoS), heterogeneous selection (HeS)) and stochastic processes (homogenizing dispersal (HD), dispersal limitation (DL), and drift (DR)).” Please see lines 219-222 on page 17 in Supplementary information I - Section B.

We check throughout the manuscript and added a list of abbreviations in Supplementary information I. Please see lines 626-636 on page 63 in Supplementary information I.

Fig. S6. The overview of composting bacterial assembly. a) Results calculated by iCAMP analysis. b) Results calculated by QPEN analysis. Bacterial assembly process includes deterministic processes (homogeneous selection (HoS), heterogeneous selection (HeS)) and stochastic processes (homogenizing dispersal (HD), dispersal limitation (DL), and drift (DR)).

Abbreviations used in Supplementary Information I

beta Net Relatedness Index: β NRI, beta mean nearest taxon distance: β MNTD,

beta nearest taxon index: β NTI, homogeneous selection: HoS, heterogeneous selection: HeS, homogenizing dispersal: HD, dispersal limitation: DL, drift: DR, decision curve analysis: DCA, canonical correlation analysis: CCA, redundancy analysis: RDA, link test for environmental filtering: LTEF, moisture content: MC, electrical conductivity: EC, germination index: GI, mean squared error: MSE, within-module connectivity: Z_i value, among-module connectivity: P_i value, ribosomal RNA operons: rrn, Ribosomal RNA Operon Copy Number Database: rrndb, mean copy number: MCN, multiple linear regression: MLG, bayesian information criterion: BIC, Sequence Read Archive: SRA, Total carbon: TC.

Line 97 – The first sentence of the paragraph says there are 10 compositing setups and the second sentence says there are 180 bacterial communities evaluated. I immediately wondered how you could get 180 samples from 10 replicates. This really isn't clear until line 367. This would not be a big deal to move up in the text, but it does highlight a bigger problem that most of the data is not independent of one another (see comments on pseudoreplication above).

Response: We thank the reviewer for this comment. Per your suggestion, we revised the misleading figures and their related text. We also check throughout the manuscript. Detail could be observed as above.

Line 119 – Text reads “We selected four measures based on relative abundance, occurrence frequency, and significance to positive cohesion to reveal the key amplicon

sequence variants (ASVs) influencing potential mutualism. Only those ASVs that fulfilled all four criteria could be defined as key ASVs (Fig. 1d and SI file2: Sheet 2).“ This is really hard to follow. As written, I only see 3 not 4 measures listed (abundance, frequency, and positive cohesion) and what it takes to fulfill these criteria is not clear since nothing says whether high or low values are important. In looking later in the manuscript, I see that the authors give four criteria on line 468, but more information at least on the direction of effects needs to come sooner.

Response: We thank the reviewer for this comment. Per your suggestion, we added this information at the beginning of the results.

We added the following text: “Four kinds of ASVs were defined, including overall abundant ASVs (average relative abundance > 0.1%)²¹, ubiquitous ASVs (detected in over 80% of samples)²², frequently abundant ASVs (the relative abundance being top 80% in more than 50% of samples)²³, and PosCoh related ASVs (ASVs with significant impact on the changes of positive cohesion). In detail, the impact was assessed by the percentage of increase of mean square error (increase of MSE), and significance was assessed by the significance of each ASV on positive cohesion (done through a random forest model²⁴) (Fig. S17 in SI file1). Only 0.0002% (3 ASVs) fulfilled all four criteria, and these accounted for 12.3% of the total sequences (Fig. S18-S20 and Text S4.1 in SI file1). These three ASVs belonged to the Actinobacteria, with two ASVs (ASV2 and ASV4) classified as *Thermobifida fusca* and one ASV (ASV1) classified as *Saccharomonospora viridis*. To distinguish them from other Actinobacterial ASVs, we named these 3 ASVs as ACT ASVs. Based on

their within-module connectivity (Z_i) and among-module connectivity (P_i) in the molecular ecological network²⁵ (Fig 1e and SI file2: Sheet 3), all ACT ASVs were also identified as keystone nodes (Fig. S21 and Text S4.2 in SI file1). ” Please see lines 116-131 on pages 6 and 7.

3. The way the authors talk about mutualisms is a bit odd in places (but could easily be corrected). For instance, in the abstract the authors write “Mutualism is supposed to be an ideal interactive paradigm of multi-species coexistence, but it is rarely observed. Abiotic stress is usually considered to promote mutualism, while there is still a paucity of research regarding how mutualism responds to it”.

First, mutualism is commonly observed in nature and there is a ton of research on how abiotic stress shapes mutualisms. The authors likely mean mutualisms in microbial communities, but as written the first two sentences of the abstract are factually incorrect.

Response: We thank the reviewer for this suggestion. In fact, we want to describe the mutualisms in bacterial communities.

Per your suggestion, we revised the sentence to “Although mutualism is commonly observed in nature, it is rarely observed in bacterial communities. While abiotic stress is usually considered to promote microbial mutualism, there is still only a paucity of research regarding how mutualism responds to it.” Please see lines 22-25 on page 2.

Second, in this writing and in other places in the text, there is a connotation of mutualisms as idealized beneficial interactions where everything is altruistic (e.g., calling it an “ideal interactive paradigm of multi-species coexistence”). Realistically, it has been well established for decades that mutualisms are complex, context-dependent interactions that often shift from beneficial to commensal or parasitic depending on the environmental conditions and there is a lot of evidence of conflict among partners in mutualisms. I would just say the wording needs reworking.

Response: We thank the reviewer for this suggestion. As you mentioned, environmental conditions can lead to a shift from beneficial to commensal or parasitic.

Per your suggestion, we revised the sentence in introduction to “Although positive interactions (primarily as commensalism) are commonly observed between cultivable strains isolated from soil⁵, mutualism is still rare as it would change to commensalism or parasitism depending on the environmental conditions.” Please see lines 44-47 on page 3.

Third, I think cross-feeding needs to be acknowledged in the intro because this is a well-established, common type of mutualistic interaction that occurs in natural microbiomes.

Response: We thank the reviewer for this suggestion. As you say, cross-feeding is a well-established, common type of mutualistic interaction.

We added a sentence to introduce the cross-feeding: “Mutualism is a win-win

situation and is represented by cross-feeding, a well-established and common type of mutualistic interaction⁴.” Please see lines 43-44 on page 3.

4. Keystone ASVs

Why are ASVs required to be in 90 out of 180 samples (line 440) to be included in the network? That is a very high threshold and may be excluding keystone ASVs which are not always dominant taxa. The four criteria around abundance and frequency for identifying key ASV (line 486) also excludes nondominant keystone ASVs. Outside of the microbial world keystone species are typically low biomass/abundance taxa that have a disproportionately large effect on their communities. Those taxa are being filtered out here. In microbiome work, researchers usually do not require taxa be rare to count as a keystone but have shown that rare and specialist taxa can be important keystones. Explicitly excluding those microbes via filtering and criteria maybe giving the false impression that there are only a few keystone ASVs.

Response: We thank the reviewer for these comments. As you say, keystone ASVs are typically low biomass/abundance taxa. While, we aimed to screen for the representative taxa (average relative abundance > 0.1% and could be detected in over 80% samples), rather than rare taxa. To avoid possible misinformation, we revised “key species / ASVs / taxa” to “ACT ASVs”, and checked throughout whole manuscript. We chose a 50% cutoff because the network reliability degraded significantly for OTUs containing >50% zeroes, and 50% cutoff was the recommendation (Weiss et al., 2016). Indeed, 50% OTU removal might exclude

nondominant ASVs, but a false positive is likely more destructive. Meanwhile, 50% cutoff is also commonly used to screen for the keystone nodes (Yuan et al., 2020). Considering the keystone nodes, calculated in the network, are not equivalent to keystone species or taxa typically used in ecology (Barberán et al., 2012). We used keystone nodes instead of keystone species or ASVs to refer to the topologically important ASVs in the network, and checked throughout whole manuscript.

We added the text in Supplementary information I- Section D to clarify keystone nodes “Defining keystone species or taxa, i.e., those with a high impact on the structure and functioning of ecosystems, is a critical, but difficult issue in ecology, especially in microbial communities. Based on the topological properties of networks, keystone nodes could be defined as module memberships. But such keystone nodes are not equivalent to keystone species or taxa typically used in ecology. Thus, in this study, we used keystone nodes instead of keystone species or taxa to refer to the topologically important nodes (ASVs) in the networks. For each node included in the constructed network, its Z_i value (i.e., within-module connectivity) and P_i value (i.e., among-module connectivity) were calculated to classify its topological role, including peripheral nodes ($P_i \leq 0.62$, $Z_i \leq 2.5$), network nodes ($Z_i \geq 2.5$, $P_i \geq 0.62$), connectors ($Z_i < 2.5$, $P_i \geq 0.62$) and module nodes ($Z_i \geq 2.5$, $P_i < 0.62$)^{9,13,14}. All but the peripheral nodes were keystone nodes⁹. In our results, the keystone nodes included module nodes (i.e., nodes highly connected to other members in a module) and connectors (i.e., nodes linking different modules) (Fig. S21). We chose a 50% cutoff because the network reliability degraded significantly for OTUs containing >50% zeroes, and

50% cutoff was the recommended level¹⁵. Indeed, 50% OTU removal might exclude nondominant ASVs, but a false positive is likely more destructive¹⁵. In conclusion, all of the selected ACT ASVs were keystone nodes, which highlighted their importance.”

Please see lines 385-403 on page 34 in Supplementary information I- Section D.

Reference

1. Yuan, M. M., Guo, X., Wu, L., Zhang, Y., Xiao, N. & Ning, D. et al. Climate warming enhances microbial network complexity and stability. *Nat Clim Change*. 11, 100-343 (2021).
2. Weiss, S., Van Treuren, W., Lozupone, C., Faust, K., Friedman, J. & Deng, Y. et al. Correlation detection strategies in microbial data sets vary widely in sensitivity and precision. *Isme J*. 10, 1669-1681 (2016).
3. Barberan, A., Bates, S. T., Casamayor, E. O. & Fierer, N. Using network analysis to explore co-occurrence patterns in soil microbial communities. *Isme J*. 6, 343-351 (2012).

Reviewer #1 (Remarks to the Author):

The manuscript has improved a lot after the authors addressed the comments provided by myself, the other reviewers and the editor. I appreciate the effort the authors took to consider my suggestions and I find that the readability of the manuscript has improved a lot.

I have one main suggestion:

While I find intriguing that temperature does favor the slower-growing taxa, the paragraph about this point appears a bit unrelated to the rest. My suggestion is to motivate exploring 16S rRNA copy number in the context of trying to explain why the relative abundance of the ACT ASVs (and the related MAGs) peaks at day 5 and then decreases as the composting process proceeds (and temperature decreases).

Minor details:

1. Lines 49-53: I would highlight the paper by Kehe et al Sci Adv. as counterexample for the dominance of competition in pairwise co-culture experiments.
2. Line 156: "high temperature favors..."
3. Line 220: please define RPKM
4. Line 252: Substitute "While," with "By contrast,"
5. Line 266: write "promoting" instead of "to promote"

Reviewer #2 (Remarks to the Author):

We greatly appreciate the effort put into revising the manuscript. The vast majority of our comments have been well addressed, but we have a few minor requests for clarification that are described in the attached document.

Reviewer #2 Attachment on the following page

Inter-bacterial mutualism promoted by public goods in a biosystem characterized by deterministic temperature variation

We thank the authors for carefully addressing each of our comments and making the appropriate changes in the figures and the text. We generally find that the answers and modifications provided by the authors are adequate, but we still would like to clarify a few points.

Here, we only list the comments for which we would like further details: in black our **original comment**, in blue the **authors' response** and in red our **new comments**.

Major comments

1) Cobalamin cross-feeding is suggested as the mechanism of facilitation happening at high temperature. While the authors demonstrate i) the accumulation of cobalamin in the supernatant after *in vitro* growth of Actinobacteria, ii) the uptake of cobalamin by Firmicutes during *in vitro* growth and 3) that mutualistic interactions peak at 50°C *in vitro*, they don't really show that cobalamin cross-feeding is the driving mechanism.

b. The authors justified their targeted analysis of cobalamin, although it was unclear whether doing this analysis differently might have highlighted a different metabolite. Is cobalamin usage "high"? Relative to what?

Response: We thank the reviewer for this comment. Cobalamin is used in a wide range of concentrations, ranging from 0.0005 to 20 ppm when culturing bacteria (Cooper et al., 2019; Higuchi-Takeuchi et al., 2019; Pira et al., 2022). We would like to confirm whether cobalamin was the main metabolites promoting the growth of Firmicutes. Therefore, the content of cobalamin needs to meet the requirements of Firmicutes growth. We found cobalamin residues only at concentration greater than or equal to 1.5 ppm (Extended Data Fig. 4), suggesting that Firmicutes growth may be limited by cobalamin deficiency at concentration below 1.5 ppm. Thus, we chose 1.5 ppm for tests.

Reference

1. Pira, H., Risdian, C., Muesken, M., Schupp, P. J. & Wink, J. *Pacificimonas pallium* sp. Nov., An Isolated Bacterium from the Mantle of Pacific Oyster *Crassostrea gigas* in Germany, and Prediction of One-Carbon Metabolism. *Diversity*. 14, (2022).
2. Cooper, M. B., Kazamia, E., Helliwell, K. E., Kudahl, U. J., Sayer, A. & Wheeler, G. L. et al. Cross-exchange of B-vitamins underpins a mutualistic interaction between *Ostreococcus tauri* and *Dinoroseobacter shibae*. *ISME J.* 13, 334-345 (2019).

Please consider adding this explanation in the Methods section and include the citations you mention in the manuscript.

2) Many statistical tests are performed to disentangle ecological effects underlying

the community dynamics observed. It is unclear to us, for example, how you can prove that increased co-occurrence at high temperature is not just due to a few species tolerating those harsh environmental conditions. Please explain these points, as they are key to the conclusions of the paper.

Response: We thank the reviewer for this comment. We added the succession of mean copy number (MCN), which is the average rRNA copy number for each sample to suggest that the increased co-occurrence is caused by high temperature selection of slow growing species (i.e., Thermobifida fusca and Saccharomonospora viridis).

Thank you for clarifying this point and making Fig. S23. Then if temperature has such a strong role in selecting for slow-growing species, the vitamin cross-feeding seems secondary (meaning a weak effect on co-occurrence). We therefore suggest writing explicitly that temperature selects for slow-growing stress-tolerant strains, and that in turn, mutualistic interactions emerge between the remaining strains. It appears important to us to disentangle explicitly the two effects: saying that “the strengthening of mutualism was largely ascribed to the high-temperature stress” (Lines 27-28) is too direct.

Minor comments

1) Of course, it would have been optimal to isolate the Actinobacteria from the compost community rather than purchase them. However, we understand that if these species could not be isolated at the time of the experiment, purchasing similar strains is appropriate.

a. Can the authors comment on why the Actinobacteria strains could not be isolated in the first place while Firmicutes were?

Response: We thank the reviewer for this comment. Actually, we isolated both Firmicutes and Actinobacteria. To ensure the representation of ACT species, we decided to purchase these strains after confirming the sequence similarity.

We don't understand the argument here: you isolated the strains from your actual community but decided they were not representative enough so you ordered others? What do you mean by “ensuring the representation of ACT species”? To cover larger phylogenetic diversity? It is unclear to us what is the benefit of using different strains to describe a system they were not a part of in the first place.

c. Mention clearly in the main text that the strains used in vitro were different from the ones in the community.

Response: We thank the reviewer for this comment. We revised Fig S3 to include the genera of used strains (i.e., Sphingomonas and Idiomarina). We also checked throughout the manuscript to avoid this mistake.

Please see lines 73-76 on page 6 in Supplementary information I- Section A.

This is not sufficient to address our comment. Please change the text in the results section to mention it explicitly (Lines 243-245): “The effects of *Thermobifida fusca* and *Saccharomonospora viridis* (Key group; **strains purchased to be more representative**) on Firmicutes (40 strains) were tested, with *Sphingomonas paucimobilis*, *Lactobacillus brevis* and *Idiomarina andamanensis* as controls (General group) (Fig. S29 in SI file1).”

14) Extended Fig. 2: Please be more specific in the legend: "based on correlation" between what and what?

Response: We thank the reviewer for this comment. The correlation were calculated between the relative abundance of neighbouring ASVs and the abundance of the functional genes.

We added the following text: "Contributions of key ASVs to composting ecological functions (carbon degradation, carbon fixation, and nitrogen cycling) based on correlation and best multiple regression model between the relative abundance of neighbouring ASVs and the abundance of the functional genes." Please see Extended Fig. 2.

We still don't understand how the figure was obtained and what the color represent.

Reviewer #3 (Remarks to the Author):

The authors have worked hard to address concerns and the manuscript is much improved.

However, I do want to make sure there is no pseudoreplication remaining in the paper. In general, this point has been addressed, but I am still not completely sure about the possibility of pseudoreplication remaining in the network analysis based on how the text is written. Can the authors please confirm that they made the network with 10 input samples and did not treat all 60 samples as independent inputs into the analysis since samples taken from the same pile at different time points are not independent and thus cannot be folded into the same network analysis. Instead, they can either make a network at each time point based on the 10 piles at that time point or 1 network based on 10 sample inputs that represent the average of all 6 time points for a given pile.

While I am aware that some researcher have published networks with 10 or less samples, I do not feel confident in a network built with only ten samples and many other microbiome researcher would agree that more samples are needed for a robust network analysis. Nonetheless, networks built with 10 input samples would at least avoid pseudoreplication. While this is still a concern, I agree it is a point of discussion in the microbiome network world right now and I am okay with overlooking this small sample size limitation for network analysis, assuming that the authors can clarify that the pseudoreplication issue is fully resolved (described in the previous paragraph).

Thank you for the diligent efforts.

Reviewer #1

The manuscript has improved a lot after the authors addressed the comments provided by myself, the other reviewers and the editor. I appreciate the effort the authors took to consider my suggestions and I find that the readability of the manuscript has improved a lot.

Response: Thank you for your recognition of our work. We are grateful for the many constructive suggestions. According to your suggestions, we have carefully revised the manuscript. We have given our point-by-point responses to your suggestions and hope the revised manuscript will meet with your approval.

I have one main suggestion:

While I find intriguing that temperature does favor the slower-growing taxa, the paragraph about this point appears a bit unrelated to the rest. My suggestion is to motivate exploring 16S rRNA copy number in the context of trying to explain why the relative abundance of the ACT ASVs (and the related MAGs) peaks at day 5 and then decreases as the composting process proceeds (and temperature decreases).

Response: We thank the reviewer for this suggestion. Per your suggestion, we rewrote the text in Results to use 16S rRNA copy number (fast-growing copiotrophs or low-growing but efficient oligotrophs) to explain why the relative abundance of ACT ASVs peaked at day 5. We also added a sentence in the Discussion to link with the above paragraph and highlight the importance of MCN to bacterial community structure.

We revised the text in the Results section to: “The relative abundance of all ACT ASVs peaked at day 5 and then decreased as temperature decreased. Moreover, the significant positive correlations between temperature and the relative abundance of the ACT ASVs (Fig. S20 in SI file1) also confirmed that high temperature favored these species. To further determine the ability of species to survive under high temperature (fast-growing copiotrophs or slow-growing but efficient oligotrophs), maximum growth rate was assessed²⁶. Considering that the ribosomal RNA operon (rrn) copy number is a phylogenetically conserved trait and a proxy for maximum growth rate²⁷, we matched Ribosomal RNA Operon Copy Number Database (rrndb) to infer maximum growth rates of neighboring ASVs and ACT ASVs (Text S5 in SI file1).” Please see lines 139-149 on pages 7 and 8.

We added the text in Discussion to emphasize the importance of maximum growth rates for bacterial community structure to link with the above paragraph: “The distribution of fast and slow growers is a powerful trait-based description of the structure of a bacterial community^{26,32}.” Please see lines 315- 316 on page 15.

Reference

26. Abreu, C. I., Dal Bello, M., Bunse, C., Pinhassi, J. & Gore, J. Warmer temperatures favor slower-growing bacteria in natural marine communities. *Sci Adv.* **9**, e8352 (2023).
33. Martiny, J. B. H., Jones, S. E., Lennon, J. T. & Martiny, A. C. Microbiomes in light of traits: A phylogenetic perspective. *Science.* **350**, (2015).

Minor details:

1. *Lines 49-53: I would highlight the paper by Kehe et al Sci Adv. as counterexample for the dominance of competition in pairwise co-culture experiments.*

Response: We thank the reviewer for this suggestion. We added the text “Kehe *et al.* reported a counterexample that positive interactions (primarily as parasitisms) are commonly observed between cultivable strains isolated from soil⁹.” Please see lines 51-53 on page 3.

2. *Line 156: “high temperature favors...”*

Response: We thank the reviewer for this suggestion. Per your suggestion, we have revised the sentence.

Please see line 164 on page 8.

3. *Line 220: please define RPKM*

Response: We thank the reviewer for this suggestion. RPKM is the abbreviation for reads per kilobase of exon model per million mapped reads.

We have modified the sentence as “The sum of reads per kilobase of exon model per million mapped reads (RPKM) for genes encoding cobalamin-dependent enzymes peaked on day 05 (2074.9), followed by day 12 (1122.8) and day 20 (774.1), when the average temperature exceeded 50°C (Fig. 3e).” Please see line 228-231 on page 11.

4. Line 252: Substitute “While,” with “By contrast,”

Response: We thank the reviewer for this suggestion. Per your suggestion, we have revised the sentence.

Please see line 261 on page 13.

5. Line 266: write “promoting” instead of “to promote”

Response: We thank the reviewer for this suggestion. Per your suggestion, we have revised the sentence.

Please see line 276 on page 13.

Reviewer #2

We greatly appreciate the effort put into revising the manuscript. The vast majority of our comments have been well addressed, but we have a few minor requests for clarification that are described in the attached document.

Response: Thank you for your recognition of our work. We are grateful for the many constructive suggestions. According to your suggestions, we have carefully revised the manuscript. We have given our point-by-point responses to your suggestions and hope the revised manuscript will meet with your approval.

Major comments

1. Please consider adding this explanation in the Methods section and include the citations you mention in the manuscript.

Response: We thank the reviewer for this suggestion. Per your suggestion, we added this explanation in the Methods section and include the citations in the manuscript.

We added sentence: “Cobalamin is used in a wide range of concentrations, ranging from 0.0005 to 20 ppm when culturing bacteria⁶⁸⁻⁷⁰. We used 1.5 ppm because we found that cobalamin residues only at concentration greater than or equal to 1.5 ppm (Extended Data Fig. 4), which means that it meets the requirements for Firmicutes growth.” Please see lines 600-604 on page 29.

2. Thank you for clarifying this point and making Fig. S23. Then if temperature has such a strong role in selecting for slow-growing species, the vitamin cross-feeding

seems secondary (meaning a weak effect on co-occurrence). We therefore suggest writing explicitly that temperature selects for slow- growing stress-tolerant strains, and that in turn, mutualistic interactions emerge between the remaining strains. It appears important to us to disentangle explicitly the two effects: saying that “the strengthening of mutualism was largely ascribed to the high-temperature stress” (Lines 27-28) is too direct.

Response: We thank the reviewer for this suggestion. Actually, we desire to show that high temperature favors slow- growing and stress-tolerant strains (i.e., *Thermobifida fusca* and *Saccharomonospora viridis*), and that these selected species would cooperate with others through the sharing of cobalamin. We agree that the original text described was too direct and misleading.

Per your suggestion, we revised this sentence to make it clearer: “Using genomic analyses and *in vitro* tests, we suggest that temperature selects for slow-growing and stress-tolerant strains (i.e., *Thermobifida fusca* and *Saccharomonospora viridis*), and mutualistic interactions emerge between them and the remaining strains through the sharing of cobalamin.” Please see lines 27-31 on page 2.

Minor comments

1. *We don't understand the argument here: you isolated the strains from your actual community but decided they were not representative enough so you ordered others? What do you mean by ensuring the representation of ACT species”? To cover larger phylogenetic diversity? It is unclear to us what is the benefit of using different strains to describe a system they were not a part of in the first place.*

Response: We thank the reviewer for this comment. We misused the word “representative”. We desired to confirm that the ability to synthesize cobalamin in *Thermobifida fusca* and *Saccharomonospora viridis* strains is universal in these two species and not unique to the strains isolated from composting. Thus, to confirm the generality of cobalamin synthesis in these species, we decided to purchase and use type strains rather than the isolated strains. Following your suggestion, we sequenced the purchased strains and compared their genomes to the MAGs in terms of sequences and metabolic capacity, which helped us to clarify the similarity of sequences and functions between the purchased strains and the MAGs. Again, we thank you for these valuable suggestions.

We added text in manuscript to clarify our purpose: “The type strains of *Thermobifida fusca* and *Saccharomonospora viridis* were purchased from the Deutsche Sammlung von Mikroorganismen und Zellkulturen (DSMZ, German) to confirm and demonstrate the generality of the ability to synthesize cobalamin within these species and not unique to the strains isolated from composting.” Please see lines 556-560 on page 27.

2. This is not sufficient to address our comment. Please change the text in the results section to mention it explicitly (Lines 243-245): “The effects of *Thermobifida fusca* and *Saccharomonospora viridis* (Key group; **strains purchased to be more representative**) on Firmicutes (40 strains) were tested, with *Sphingomonas paucimobilis*, *Lactobacillus brevis* and *Idiomarina andamanensis* as

controls (General group) (Fig. S29 in SI file1)."

Response: We thank the reviewer for this suggestion. Per your suggestion and the response to minor comment question 1, we revised the sentence "The effects of *Thermobifida fusca* and *Saccharomonospora viridis* (Key group; type strains purchased to be more generality) on Firmicutes (40 strains) were tested, with *Sphingomonas paucimobilis*, *Lactobacillus brevis* and *Idiomarina andamanensis* as controls (General group) (Fig. S29 in SI file1)." Please see lines 252-255 on page 12.

3. *We still don't understand how the figure was obtained and what the color represent.*

Response: We thank the reviewer for this comment. This figure was obtained by Pearson's correlation and random forest model. In detail, the bar and circle size were obtained from the random forest model and the heatmap was obtained from Pearson's correlation. In the random forest model, the relative abundance of neighboring ASVs served as predictors for the copy numbers of functional genes. The height of the bar represents the explained variation (i.e., %Var explained), indicating the degree to which all neighboring ASVs explain the variation in functional gene copy number. The circle size represents the variable importance of each neighboring ASV for the variation in functional gene copy number (i.e., percentage of increase of mean square error). The heatmap was constructed via the function "cor()".

We revised the legend to make it clearer: "Extended Data Fig. 2. Contributions of neighboring ASVs to composting ecological functions (carbon degradation, carbon

fixation, and nitrogen cycling) based on correlation and random forest model. In this random forest model, the relative abundance of each neighboring ASV served as predictors for the copy number of each functional gene. The bar was the explained variation (i.e., %Var explained) calculated *via* R package “*randomForest*”, indicating the degree to which all neighboring ASVs explained the variation in functional gene copy number. The circle size represents the variable importance of each neighboring ASV for the variation in functional gene copy number (i.e., percentage of increase of mean square error), which was calculated *via* R package “*rfPermute*”. The heatmap is a Pearson’s correlation between the relative abundance of each neighboring ASV and the copy number of each functional gene. The heatmap was constructed via the function “*cor()*”. Colors represent Pearson’s correlation coefficients.” Please see section Extended Data Fig.

Reviewer #3

The authors have worked hard to address concerns and the manuscript is much improved.

Response: Thank you for your recognition of our work. We are grateful for the many constructive suggestions. According to your suggestions, we have carefully revised the manuscript. We have given our point-by-point responses to your suggestions and hope the revised manuscript will meet with your approval.

However, I do want to make sure there is no pseudoreplication remaining in the paper. In general, this point has been addressed, but I am still not completely sure about the possibility of pseudoreplication remaining in the network analysis based on how the text is written. Can the authors please confirm that they made the network with 10 input samples and did not treat all 60 samples as independent inputs into the analysis since samples taken from the same pile at different time points are not independent and thus cannot be folded into the same network analysis. Instead, they can either make a network at each time point based on the 10 piles at that time point or 1 network based on 10 sample inputs that represent the average of all 6 time points for a given pile.

Response: We thank the reviewer for this suggestion. Per your suggestion, we reconstructed a network based on 10 sample inputs that represent the average of all 6 time points for a given pile. We revised the related texts, figures and tables, including: Fig 1e, Fig 1f, Extended Data Fig. 2, Fig S21, Fig S22, Fig S23a, SI file2: Sheet 3, and SI file2: Sheet 4. **Results showed that all modifications had a consistent result**

in that the conclusions were the same as before, although the detailed values changed.

We added sentence in the Method section to clarify the number of samples used in network construction: “A molecular ecological network (MEN) was constructed based on 10 sample inputs that represented the average of all 6 time points for a given pile.” Please see lines 502-503 on page 24.

We revised the detailed values in the Results section “The sub-network between ACT ASVs and neighboring ASVs (those directly connected to the ACT ASVs) indicated that 23 neighboring ASVs (71.9%), were positively correlated with the ACT ASVs (Fig. 1f and SI file2: Sheet 4). More than 68.8% of them were affiliated with Firmicutes (Fig. 1f and Fig. S22 in SI file1), which is the dominant phylum in composting systems¹⁸.”; “The significantly higher copy number of *rrn* in neighbouring ASVs (6.4) (Fig. S23 in SI file1), and the high explanation of changes in functional potential (55.0%) (Extended Data Fig. 2) emphasized that the neighboring ASVs were fast-growing bacteria, and were highly associated with composting function.” Please see lines 133-138 on page 7 and lines 149-153 on page 8.

Fig.1 Changes of temperature and positive cohesion. **a)** Temperature variations in the 10 composting piles. 50°C is generally considered to be the dividing line for composting in terms of distinguishing the high temperature phase from other phases in this system. **b)** Changes in the positive cohesion of different phases. In the boxplots of panels, hinges indicate the 25th, 50th, and 75th percentiles, whiskers indicate $1.5 \times$ interquartile ranges, and dots indicate values of individual samples. **c)** Relationship between positive cohesion and temperature. **d)** Screening for the ACT ASVs (detail of 4 criteria are described in Methods section). **e)** Whole composting network constructed by the random matrix theory (RMT) model. The red line indicates a positive correlation and the blue lines a negative correlation. **f)** The

sub-network containing ACT ASVs and their neighbors. Different colors of the outer ring are used to distinguish taxonomic affiliation.

Fig. S21. Zi-Pi analysis for screening the keystone nodes.

Fig. S22. Phylogenetic analysis of the 16S rRNA gene (V4 region) of neighboring ASVs

Fig. S23. Mean copy number (MCN) during the composting process. a) Ribosomal RNA operon copy number between neighbouring ASVs and ACT ASVs. In the boxplots of panels, hinges indicate the 25th, 50th, and 75th percentiles, whiskers indicate $1.5 \times$ interquartile ranges, and dots indicate values of individual samples. b) MCN for different sampling time points. c) The relationship between MCN and temperature.

While I am aware that some researcher have published networks with 10 or less samples, I do not feel confident in a network built with only ten samples and many other microbiome researcher would agree that more samples are needed for a robust network analysis. Nonetheless, networks built with 10 input samples would at least avoid pseudoreplication. While this is still a concern, I agree it is a point of discussion in the microbiome network world right now and I am okay with overlooking this small sample size limitation for network analysis, assuming that the authors can clarify that the pseudoreplication issue is fully resolved (described in the previous paragraph). Thank you for the diligent efforts.

Response: We thank the reviewer for this comment. As you mentioned, more samples are needed for a robust network analysis. We were limited by the size of the research facility (10 piles could be operated at same time), thus we only included 10 samples in our analysis. Following your suggestions, we will include more biological repetitions in future studies. Please watch for our follow-up research. Again, we thank the reviewer for the constructive suggestions.

Reviewer #3 (Remarks to the Author):

Thank you for the hard work editing this paper. I am satisfied with the revised manuscript and congratulate the authors on a job well done.